# Loss-of-function mutations in *PLD4* lead to systemic lupus erythematosus

Qintao Wang[1,6], Honghao Zhu[1,6], Xiangwei Sun[1,6], Changming Zhang[2,6], Shuangyue Ma[1,6], Ying Jin[2], Jinjian Fu[3], Chenlu Liu[3], Jiahui Peng[1], Ruoran Wang[1], Lin Liu[4], Yi Zeng[1], Cheng Gong[1], Qing Zhou[1,3,5 ✉], Xiaomin Yu[1 ✉] & Zhihong Liu[1,2 ✉]

Monogenic lupus offers valuable insights into the underlying mechanisms and therapeutic approaches for systemic lupus erythematosus (SLE)[1–3]. Here we report on five patients with SLE carrying recessive mutations in phospholipase D family member 4 (*PLD4*). Deleterious variants in *PLD4* resulted in impaired single-stranded nucleic acid exonuclease activity in in vitro and ex vivo assays. *PLD4* loss-of-function mutations led to excessive activation of Toll-like receptor 7 (TLR7) and TLR9. Downstream inflammatory signalling pathways, especially type I interferon signalling, were hyperactivated in patient dendritic cells. *Pld4*-deficient mice presented with autoimmunity and cell-intrinsic expansion of plasmacytoid dendritic cells and plasma cells. *Pld4*-deficient mice responded to the JAK inhibitor baricitinib, suggesting that targeting type I interferon may be a potential therapy for patients with PLD4 deficiency.

SLE is a complex multiorgan condition of variable severity[4,5]. Monogenic lupus represents a subset of autoimmune disorders caused by mutations in single genes and encompasses a spectrum of diseases with lupus-like phenotypes[1]. To date, more than 30 disease-causing genes leading to lupus have been reported[1]. Identifying more disease-causing genes of lupus could improve diagnosis, deepen pathological understanding and develop targeted therapeutics for this complex disease.

Intracellular nucleic-acid-sensing pathways have a crucial role in defending against external pathogens, tissue damage and repair[6,7]. TLR7 and TLR9 located in the endosome are pivotal for sensing RNA and DNA and are crucial for the development of SLE[2,8–10]. They initiate downstream inflammatory signalling pathways such as type I interferon (IFN), nuclear factor kappa B (NF-κB) and mitogen-activated protein kinase (MAPK) by recognizing endogenous or exogenous nucleic acid[11,12]. In plasmacytoid dendritic cells (pDCs), activation of the TLR7 and TLR9 pathways leads to the release of large amounts of IFNs, promoting the presentation of autoantigens and the occurrence of inflammatory responses[13]. In B cells, activation of both pathways leads to the production of substantial autoantibodies against nucleic acids, contributing to SLE[14].

PLD4[15–18] is highly expressed in DCs, monocytes and B cells. It is a 5′ exonuclease that localizes in the endolysosomes and can cleave single-stranded RNA (ssRNA) and single-stranded DNA (ssDNA), thereby restricting the overactivation of TLR7 and TLR9[18–23]. *Pld4*-knockout (KO) mice demonstrate a range of autoimmune phenotypes, including reduced body weight, enlarged spleen size, increased autoantibodies and immune complex deposition[24]. Furthermore, mice lacking both *Pld4* and its family member *Pld3* die early in life[19].

Although autoimmune phenotypes have been established in *Pld4*-deficient mice, PLD4 deficiency has not yet been implicated in human diseases. Here we report *PLD4* recessive mutations in five patients with SLE.

## *PLD4* variants identified in patients with SLE

The five patients were diagnosed with SLE. All of the patients presented with a renal phenotype. Kidney biopsy showed proliferative lupus nephritis (Fig. 1a and Supplementary table 1). Haematological involvement was observed in all of the patients, including leukopaenia, anaemia and thrombocytopenia. Skin rash, such as urticaria, malar rash or patchy rash, was identified in four patients. Arthralgia/arthritis and serositis were each noted in three patients (Supplementary table 1). Moreover, all of the patients were positive for antinuclear antibodies (ANA) and hypocomplementaemia. Using whole-exome sequencing (WES), we found that all of the patients carry biallelic mutations in the *PLD4* gene (Fig. 1b and Extended Data Fig. 1a). All of the mutations were localized in the catalytic domain of PLD4 and were predicted to be deleterious (Fig. 1c and Supplementary table 2). Although different mutations of PLD4 are not spatially clustered within a single structural domain, the structure predicts that they may affect exonuclease activity through different mechanisms, such as affecting the formation of hydrogen bonds or affecting binding with substrates (Extended Data Fig. 1b–e).

## Enhanced type I IFN pathway in DCs

To comprehensively assess inflammatory levels in the patients, we performed RNA-sequencing (RNA-seq) analysis of peripheral blood mononuclear cells (PBMCs) from patient P1, patient P2 and healthy control individuals. Gene set enrichment analysis (GSEA) revealed that IFNα response, IFNγ response and TNF signalling through NF-κB were the most significantly enriched pathways in patients P1 and P2 (Fig. 2a and Extended Data Fig. 2a). The heat map showed genes associated with inflammatory signalling pathways, particularly type I IFN

[1]Liangzhu Laboratory, Zhejiang University, Hangzhou, China. [2]National Clinical Research Center of Kidney Diseases, Jinling Hospital, Affiliated Hospital of Medical School, Nanjing University, Nanjing, China. [3]Life Sciences Institute, Zhejiang University, Hangzhou, China. [4]Urology and Nephrology Center, Department of Nephrology, Zhejiang Provincial People's Hospital, Affiliated People's Hospital, Hangzhou Medical College, Hangzhou, China. [5]Department of Rheumatology, Sir Run Run Shaw Hospital, Zhejiang University School of Medicine, Hangzhou, China. [6]These authors contributed equally: Qintao Wang, Honghao Zhu, Xiangwei Sun, Changming Zhang, Shuangyue Ma. ✉e-mail: zhouqingnwu@gmail.com; yuxiaomin@zju.edu.cn; liuzhihong@zju.edu.cn

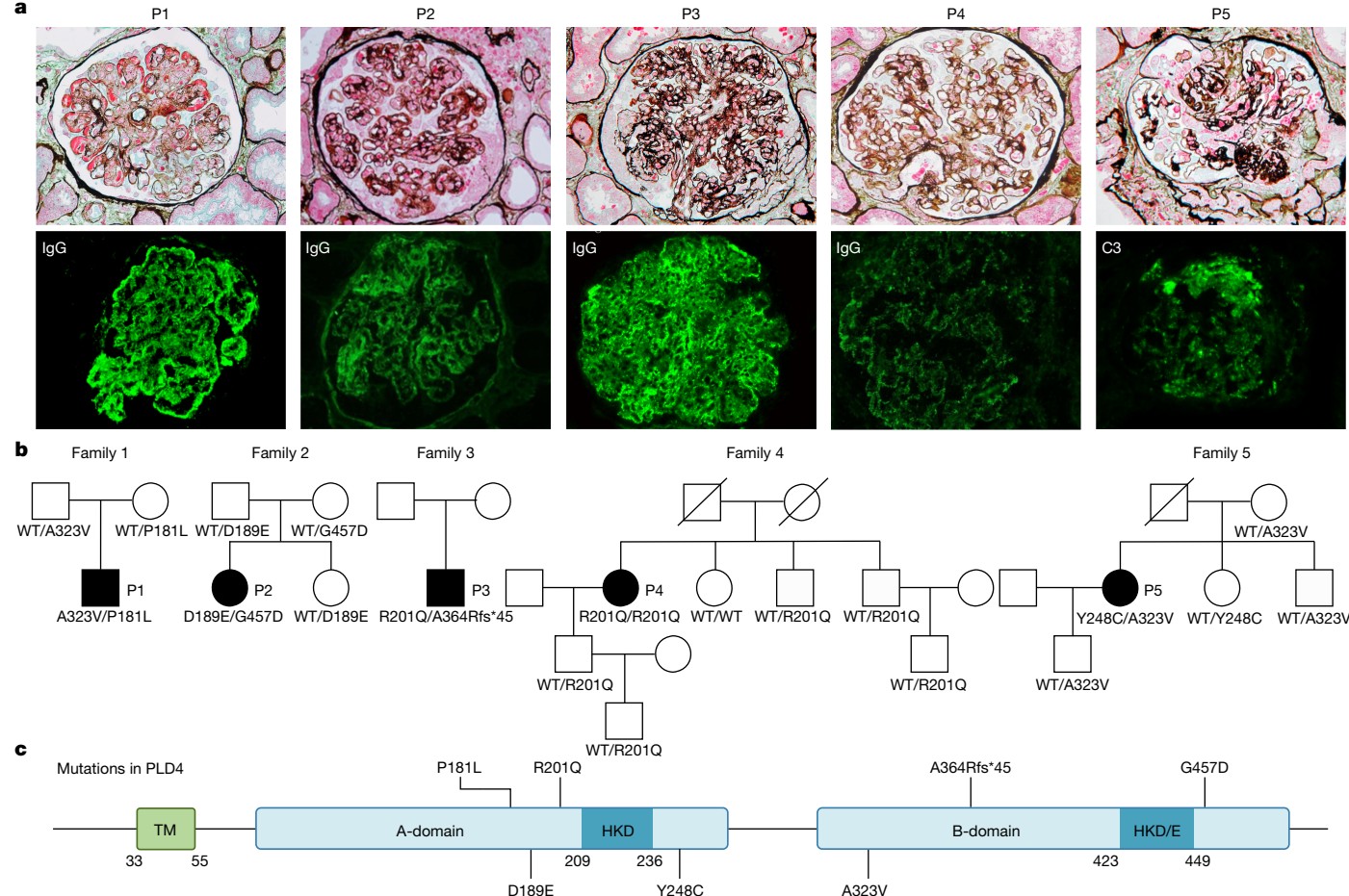

**Fig. 1 | Identification of biallelic *PLD4* variants in five patients with SLE. a**, PASM staining and immunofluorescence staining of glomeruli in kidney biopsies of five patients (P1–P5). **b**, Pedigree of five patients carrying *PLD4* mutations. The open circles and boxes indicate no clinical phenotype, and the solid black circles and boxes indicate the presence of a clinical phenotype. WT, wild type. **c**, Schematic of the location of mutations on the PLD4 protein structure. TM, transmembrane domain.

pathway, were significantly upregulated in PBMCs from patients P1 and P2 (Fig. 2b and Extended Data Fig. 2b). Flow cytometry results showed an upregulation in the phosphorylation of STAT1, further demonstrating activation of type I IFN pathway in PBMCs of patients P1 and P2 (Fig. 2c). Moreover, intracellular staining results showed that IFNα, IL-1β, IL-6 and IL-8 were significantly increased in the PBMCs of patient P1 (Extended Data Fig. 2c,d). Furthermore, the proportion of CD14[+] monocytes increased, whereas the proportion of CD19[+] B cells decreased in PBMCs of patients P1 and P2 (Extended Data Fig. 2e). These findings indicate that the type I IFN signalling pathway is significantly activated in the cells of patients P1 and P2.

We performed single-cell RNA-seq (scRNA-seq) analysis of the PBMCs from patient P1, patient P2 and healthy control individuals to identify differences in the expression profiles among different cell types (Fig. 2d). Patient cells with high expression of NF-κB-pathway-related genes were distributed across most cell types. Notably, cells with high expression of genes related to the type I IFN pathway in patients P1 and P2, such as *IFIT2*, *OAS2*, *IFI44* and *IFI44L*, were primarily found in PLD4-expressing cells, such as DCs and monocytes (Fig. 2e and 2f (top) and Extended Data Figs. 3 and 4a). Genes encoding TLR7/9 and TLR signalling pathway-related molecules, including *TLR7*, *TLR9*, *TRAF6*, *IRAK1* and *IRAK4*, were also significantly elevated in DCs of patients P1 and P2 compared with in the healthy controls (Fig. 2f (bottom) and Extended Data Fig. 4b). The flow cytometry results demonstrate that TLR9 is upregulated in pDCs of patient P1, consistent with that in patients with SLE without *PLD4* mutations (Extended Data Fig. 4c,d).

Furthermore, genes involved in the type I IFN pathway were also abnormally activated in the B cells of patients P1 and P2 (Extended Data Fig. 4e). Flow cytometry analysis of PBMCs from P1 and P4 showed normal IgG levels and B cell subset distributions during remission, aligning with healthy controls, whereas the flare phase of P4 mirrored those of patients with SLE without *PLD4* mutations (Extended Data Fig. 5a–f).

Cytometry by time of flight (CyTOF) results showed that the levels of IFNα, IFNγ, TNF, IL-1β, CXCL2, CCL4, IL-23 and GM-CSF in PBMCs of patient P2, especially DCs, were higher than in the healthy controls (Fig. 2g and Extended Data Fig. 5g). Through measurements of inflammatory gene expression levels and cytokine levels in patients P1 and P2, we found that the TLR7/9 and downstream type I IFN pathway were most significantly activated in DCs, suggesting that PLD4 deficiency in DCs triggers systemic inflammation and autoimmunity in patients.

## Mutations impair PLD4 function

To validate the effects of these mutations on PLD4 function, we used purified wild-type and mutant PLD4 proteins to assess the single-stranded nucleic acid exonuclease activity of PLD4. Our results demonstrated that these missense mutations (Pro181Leu, Asp189Glu, Arg201Gln, Tyr248Cys, Ala323Val and Gly457Asp) impaired the exonuclease activity of PLD4 (Fig. 3a and Extended Data Fig. 6a). To further elucidate the impact of these mutations on PLD4 exonuclease activity,

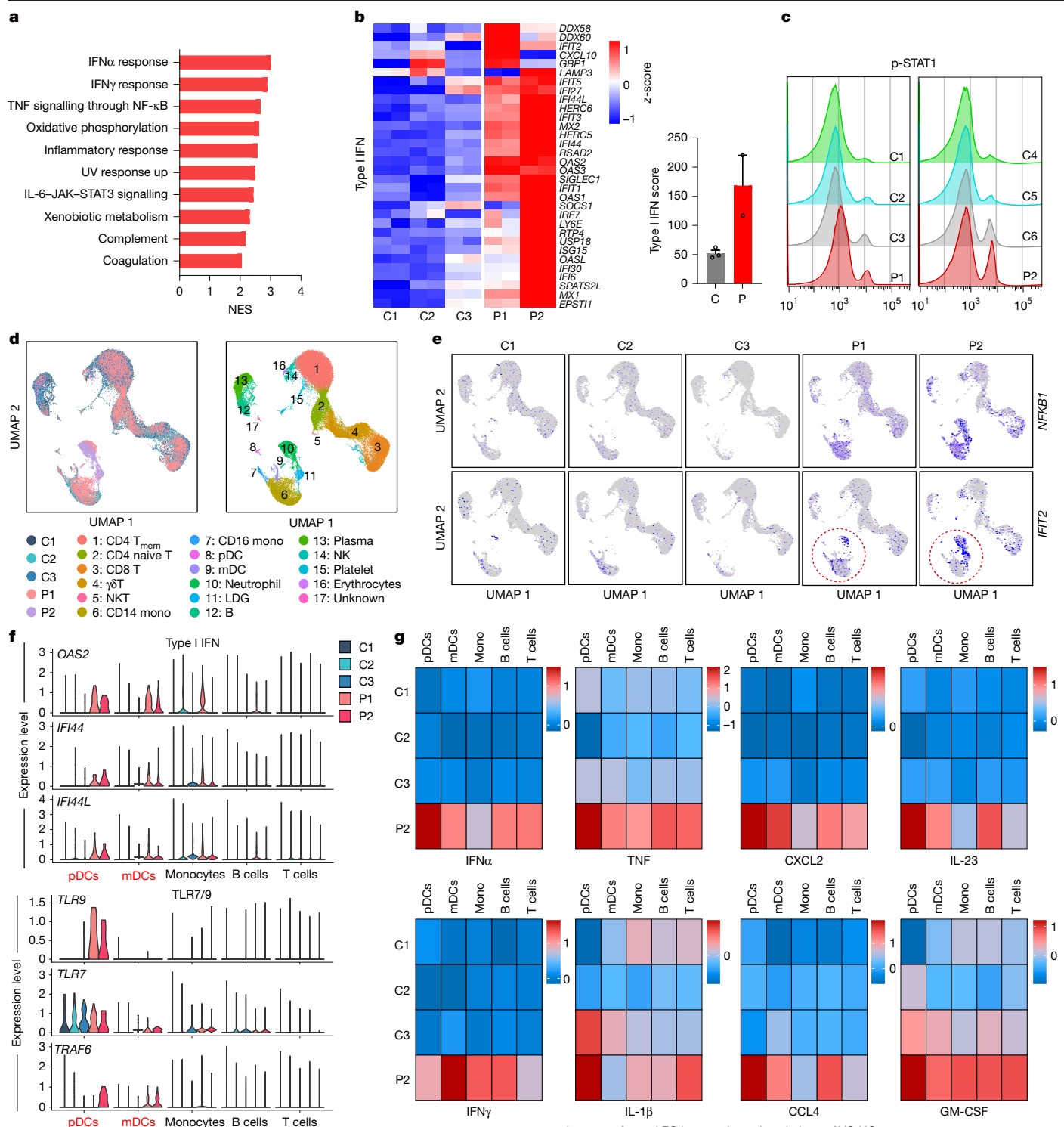

**Fig. 2 | Aberrant activation of TLR signalling and type I IFN pathway in patient DCs. a,b,** RNA-seq analysis of PBMCs from patients P1 and P2 and healthy controls. Enriched hallmark gene sets are shown. **a,** The top 10 enriched pathways in GSEA of PBMCs from patients P1 and P2. NES, normalized enrichment score. **b,** The type I IFN pathway involved genes and type I IFN scores in control (C) and patient (P) samples. Data are mean ± s.e.m. **c,** Flow cytometry analysis of the phosphorylation levels of STAT1 in patients P1 and P2 and healthy control (C1–C5) PBMCs. **d–f,** scRNA-seq analysis of PBMCs from patients P1 and P2 and healthy controls. NK, natural killer cells; NKT, natural killer T cells; T$_{mem}$, memory T cells. **d,** Uniform manifold approximation and projection (UMAP) plot showing the differences in various cell types between patients P1 and P2 and healthy controls. **e,** UMAP plot of NF-κB and type I IFN signalling pathways genes. The red circle indicates *PLD4*-expressing cells (DCs and monocytes (mono)) and inflammation-induced low-density granulocytes (LDGs). **f,** The upregulated expression of key genes in type I IFN and TLR7/9 signalling pathways in major cell populations. TLR7/9, TLR7/9 signalling pathways. **g,** CyTOF analysis of PBMCs from patient P2 and healthy controls. The average expression of inflammatory cytokines in major cell populations is shown. The results in **c** are representative of two independent experiments. FC, fold change.

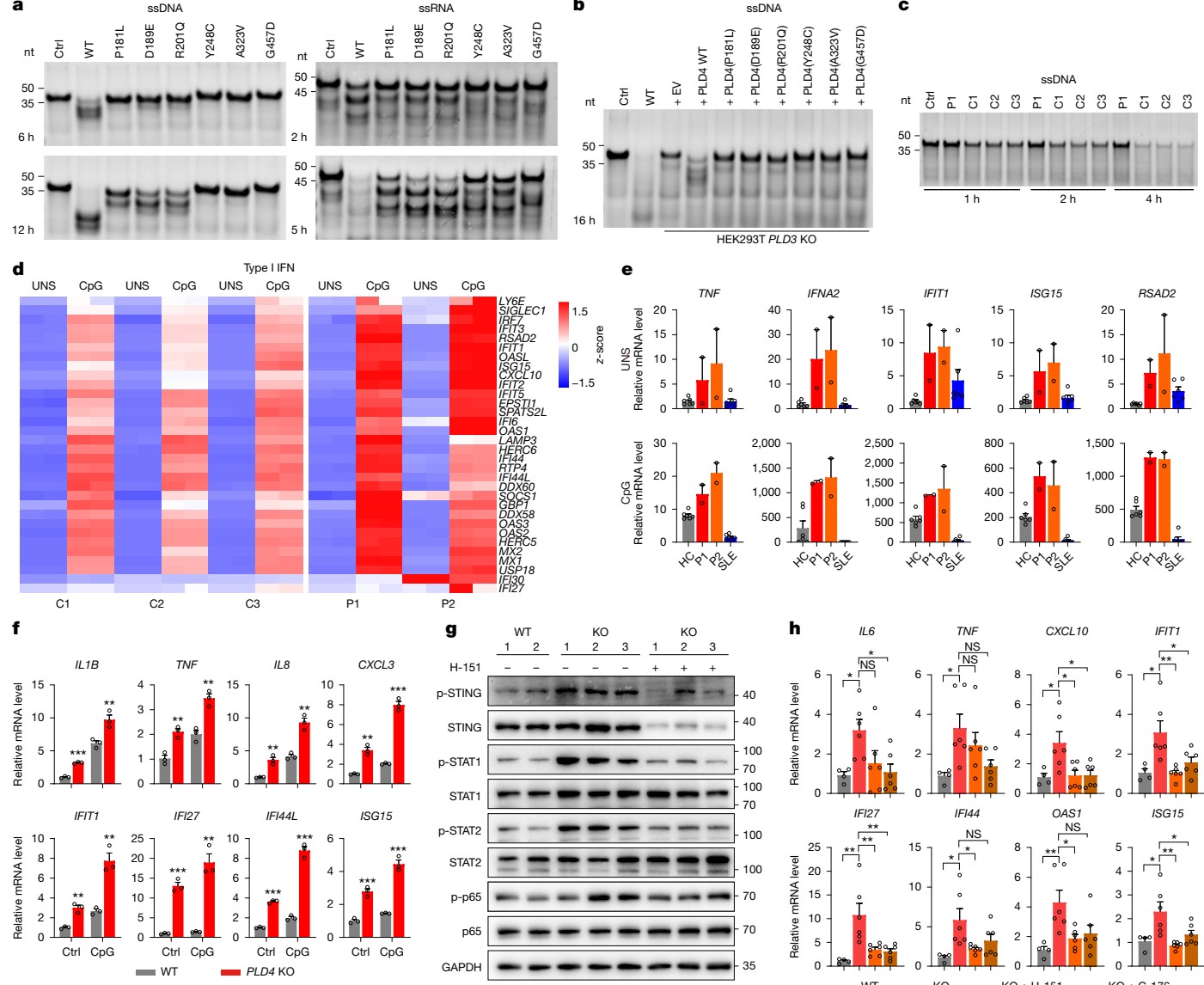

**Fig. 3 | Mutations impair PLD4 exonuclease activity and result in aberrant type I IFN signalling activation. a–c,** The impact of the mutations on PLD4 exonuclease activity. Ctrl, control; nt, nucleotides. **a,** Single-stranded nucleic acid exonuclease activity of purified wild-type and mutant PLD4 at different times. **b,** ssDNA exonuclease activity of HEK293T *PLD3*-KO cells (eliminating the endogenous *PLD3* interference) reconstituted with wild-type or mutant *PLD4*. EV, empty vector. **c,** ssDNA exonuclease activity of PLD4 in patient P1 and healthy control PBMCs. **d,** RNA-seq analysis of type I IFN pathway genes in patients P1 and P2 and healthy controls at the basal level and after CpG-DNA stimulation. UNS, unstimulated. **e,** NF-κB and type I IFN pathway gene expression in PBMCs of patients P1 and P2, healthy controls and patients with SLE. *n* = 6 healthy controls (HC) and *n* = 6 patients with SLE without *PLD4* mutations (SLE); the dots in the patient group represent samples taken at a different time for a patient. **f,** NF-κB and type I IFN pathway gene expression at the basal level and after CpG-DNA treatment in *PLD4*-KO THP-1 cells. *n* = 3 (WT) and *n* = 3 (*PLD4* KO). **g,h,** The effects of STING inhibitors H-151/C-176 on inflammatory pathways in THP-1 *PLD4*-KO monoclonal cells. **g,** Western blot analysis of the signalling pathways changed after H-151 treatment. **h,** NF-κB and type I IFN pathway gene expression after H-151 and C-176 treatment. *n* = 4 (WT) and *n* = 6 (KO, KO + H-151, KO + C-176). Data are mean ± s.e.m. The results are representative of at least three independent experiments (**a**, **b** and **f–h**), two independent experiments (**c**) and a summary of two independent experiments (**e**). Statistical analysis was performed using one-way analysis of variance (ANOVA) with Tukey's post hoc analysis (**h**) and unpaired *t*-tests followed by false-discovery rate (FDR) correction (**f**); NS, not significant; \**P* < 0.05, \*\**P* < 0.01, \*\*\**P* < 0.001.

we used cell lysates from HEK293T cells overexpressing these mutations to conduct exonuclease activity assays. The results consistently showed that the ssDNA in all mutant-PLD4 groups remained uncleaved (Fig. 3b and Extended Data Fig. 6b). Moreover, we evaluated the exonuclease activity of endogenous PLD4 in PBMCs from patient P1 and healthy controls. The results showed that, after 1, 2 and 4 h of the exonuclease activity assay, patient P1 exhibited more residual substrate over time compared with the healthy controls, suggesting that the exonuclease activity of patient P1's endogenous PLD4 is markedly impaired (Fig. 3c).

## Activated TLR7/9 pathways in patients

PLD4 enzymatically cleaves ssRNA and ssDNA to prevent excessive activation of TLR7 and TLR9[18–20]. We demonstrated the upregulation of both full-length and activated forms of TLR7 in the PBMCs of patient P1 compared with in the healthy controls, with concomitant activation of the downstream inflammatory signalling, such as the type I IFN and MAPK pathways (Extended Data Fig. 6c).

Flow cytometry results showed an upregulation in the phosphorylation of STAT2, p65 and ERK, indicating activation of type I IFN,

NF-κB and MAPK pathways in PBMCs of patients P1 and P2 at the basal level and after stimulation with the TLR9 agonist unmethylated cytosine-phosphate-guanine DNA (CpG-DNA; Extended Data Fig. 6d,e). Moreover, RNA-seq analysis of PBMCs from patients P1 and P2 revealed significantly elevated type I IFN pathway gene expression in patients compared with in the healthy controls, both at basal level and after CpG-DNA stimulation (Fig. 3d). Quantitative PCR (qPCR) analysis confirmed that the transcriptional levels of key inflammatory genes, such as *TNF*, *IFNA2*, *IFIT1*, *ISG15* and *RSAD2*, were also upregulated in PBMCs from patients P1 and P2 compared with in the healthy controls at the basal level and after CpG-DNA stimulation, whereas CpG-DNA-stimulated PBMCs from patients with SLE without *PLD4* mutations exhibited minimal responses (Fig. 3e). This is consistent with previous studies suggesting that the high IFNα levels and immune microenvironment in typical patients with SLE may lead to increased tolerance to TLR stimulation. Notably, after CpG-DNA stimulation of isolated monocytes from patient P1, several genes elevated fold upregulation relative to the healthy controls (Extended Data Fig. 6f,g).

Moreover, we generated a THP-1 *PLD4*-KO cell line, which exhibited upregulation of phosphorylated STAT1 (p-STAT1), p-p65 and p-ERK, indicating that TLR7/9 downstream pathways such as type I IFN, NF-κB and MAPK were activated (Extended Data Fig. 6h). Furthermore, transcription levels of inflammatory cytokines, chemokines and IFN-stimulated genes (ISGs) in KO cells were upregulated at the basal level. After CpG-DNA stimulation, *IFIT1*, *IFI44L* and *CXCL3* exhibited a more pronounced increase in expression in KO cells compared with in the wild-type cells (Fig. 3f). In KO cells, the levels of pro-inflammatory cytokines IL-1β and IL-6 were also significantly higher than in the wild-type cells (Extended Data Fig. 6i).

## Activated STING in *PLD4*-KO cell

Previous investigations have established that STING-dependent signalling, particularly type I IFN responses, activated in PLD3/PLD4-deficient mice models[19]. Besides, PLD3 ablation induces lysosomal accumulation of mitochondrial DNA followed by cytoplasmic leakage, thereby triggering cGAS–STING pathway activation and subsequent autophagy[25]. To dissect STING's involvement in PLD4-deficiency-driven inflammation, we examined the activation of STING in THP-1 *PLD4*-KO monoclonal cells (Extended Data Fig. 6j). Immunoblotting demonstrated the phosphorylation of STING and inflammatory pathway enhanced in THP-1 *PLD4*-KO monoclonal cell lines (Fig. 3g).

Treatment with specific inhibitors of STING H-151[26] effectively attenuated the type I IFN signalling activation induced by PLD4 deficiency (Fig. 3g). Notably, downstream inflammatory genes, especially type I IFN involved genes such as *IFIT1*, *IFI27*, *IFI44*, *OAS1* and *ISG15*, were significantly downregulated when treated with H-151 and C-176[26] (Fig. 3h). Furthermore, *PLD4* and *STING1* double-KO cell lines corroborated these findings. Both qPCR and western blot analysis demonstrated the rescue of type I IFN pathway activation after STING ablation, whereas NF-κB signalling showed only partial restoration (Extended Data Fig. 6k,l). These findings align with previous studies[19] and collectively establish an important role of the cGAS–STING signalling axis in mediating the immune dysregulation resulting from PLD4 deficiency.

## *Pld4*$^{-/-}$ mice manifest autoimmunity

*Pld4* homozygous KO (*Pld4*$^{-/-}$) mice exhibited autoimmune phenotypes, such as slower body weight gain, significantly elevated levels of anti-double-stranded-DNA (dsDNA) antibodies, anti-dsRNA antibodies and IgG in the plasma and splenomegaly compared with in the wild-type and heterozygous KO (*Pld4*$^{+/-}$) mice (Fig. 4a,b and Extended Data Fig. 7a,b). Moreover, *Pld4*$^{-/-}$ mice displayed severe nephritic phenotypes, including thickening of the glomerular basement membrane

(Extended Data Fig. 7c) and increased deposition of IgG and C3 (Extended Data Fig. 7d,e). Among various organs accumulating autoimmune damage, inflammatory genes (*Tnf*, *Cxcl10*, *Mx2* and *Ifng*) were most prominently elevated in the kidneys of *Pld4*$^{-/-}$ mice compared with in the wild-type and *Pld4*$^{+/-}$ mice, consistent with the nephritis manifestation in patients (Extended Data Fig. 7f).

scRNA-seq analysis of the kidneys showed a twofold to threefold increase in the overall and various proportions of infiltrating immune cells, including macrophages, DCs, T and B cells, in *Pld4*$^{-/-}$ mice compared with in wild-type mice (Extended Data Fig. 7g–i). Moreover, genes involved in the type I IFN pathway, such as *Ifi27*, *Isg15* and *Ddx58*, were significantly upregulated in both immune cells and renal tissue cells (podocytes, endothelial cells, principal cells and proximal tubule cells) in *Pld4*$^{-/-}$ mice compared with in wild-type mice (Fig. 4c and Extended Data Fig. 7j).

Flow cytometry analysis of mouse kidneys corroborated scRNA-seq data: immune cell populations exhibited marked expansion in *Pld4*$^{-/-}$ mice compared with in wild-type mice (Fig. 4d). Key cell populations implicated in the pathogenesis of SLE, including pDCs and plasma cells, exhibited significant elevations (Fig. 4e,f). Furthermore, CD4$^+$ effector T cells and CD8$^+$ effector T cells were markedly increased, whereas age-associated B cells and myeloid DCs (mDCs) remained unchanged (Fig. 4g and Extended Data Fig. 7k–m). Within the spleen, only pDCs and plasma cells demonstrated pronounced increases in *Pld4*$^{-/-}$ mice compared with in the wild-type mice (Extended Data Fig. 8a–h). These observations suggest distinct tissue-specific consequences arising from PLD4 deficiency, with particularly pronounced renal tissue damage.

To determine which immune cell expansion was cell intrinsic, mixed bone marrow chimeras were generated by transplants 1:1 mixes of bone marrow from WT-CD45.1–WT-CD45.2 or WT-CD45.1–*Pld4*$^{-/-}$-CD45.2 into lethally irradiated WT-CD45.1 mice. The autoantibody analysis revealed that chimeric mice reconstituted with *Pld4*$^{-/-}$ bone marrow exhibited elevated levels of anti-dsDNA and anti-dsRNA antibodies (Fig. 4h). Flow cytometry results in the kidneys revealed that the expansion of pDCs and plasma cells was cell intrinsic, whereas the expansion of T cells was cell extrinsic (Fig. 4i and Extended Data Fig. 9a–f). Similarly, splenic analyses showed equivalent results, with pDCs and plasma cells also exhibiting a cell-intrinsic effect (Extended Data Fig. 9g–m).

Together, the manifestations observed in mice support the pivotal roles of PLD4 in the development of SLE and effect of inflammatory responses in nephritis pathogenesis.

## JAKi rescues *Pld4*$^{-/-}$ mouse phenotypes

Given that type I IFN pathway was significantly upregulated in *Pld4*$^{-/-}$ mice and patients, we hypothesized that treatment with the JAK inhibitor (JAKi) baricitinib might act as an effective therapy. After 8 weeks of simulated oral administration through gavage (Extended Data Fig. 10a), the baricitinib-treated mice showed significant improvements in body-weight gain, plasma levels of anti-dsDNA and anti-dsRNA antibodies, and spleen size compared with the untreated mice (Fig. 5a–c). Moreover, renal tissue inflammatory genes expression, such as *Il1b*, *Il6*, *Tnf*, *Ifng*, *Mx2* and *Ifit1* (Fig. 5d), and glomerular immune complex deposition like IgG and C3 in the baricitinib-treated mice were significantly reduced compared with in the untreated mice (Fig. 5e and Extended Data Fig. 10b).

Based on the positive responses in the mice, we treated patient PBMCs with baricitinib. The results showed that treatment with baricitinib markedly inhibited the elevated type I IFN pathway in patient PBMCs and partially inhibited the NF-κB pathway (Fig. 5f). In summary, the favourable effects of baricitinib suggest that the type I IFN is a critical pathway in the autoimmune and inflammatory phenotype after PLD4 deficiency (Extended Data Fig. 10c).

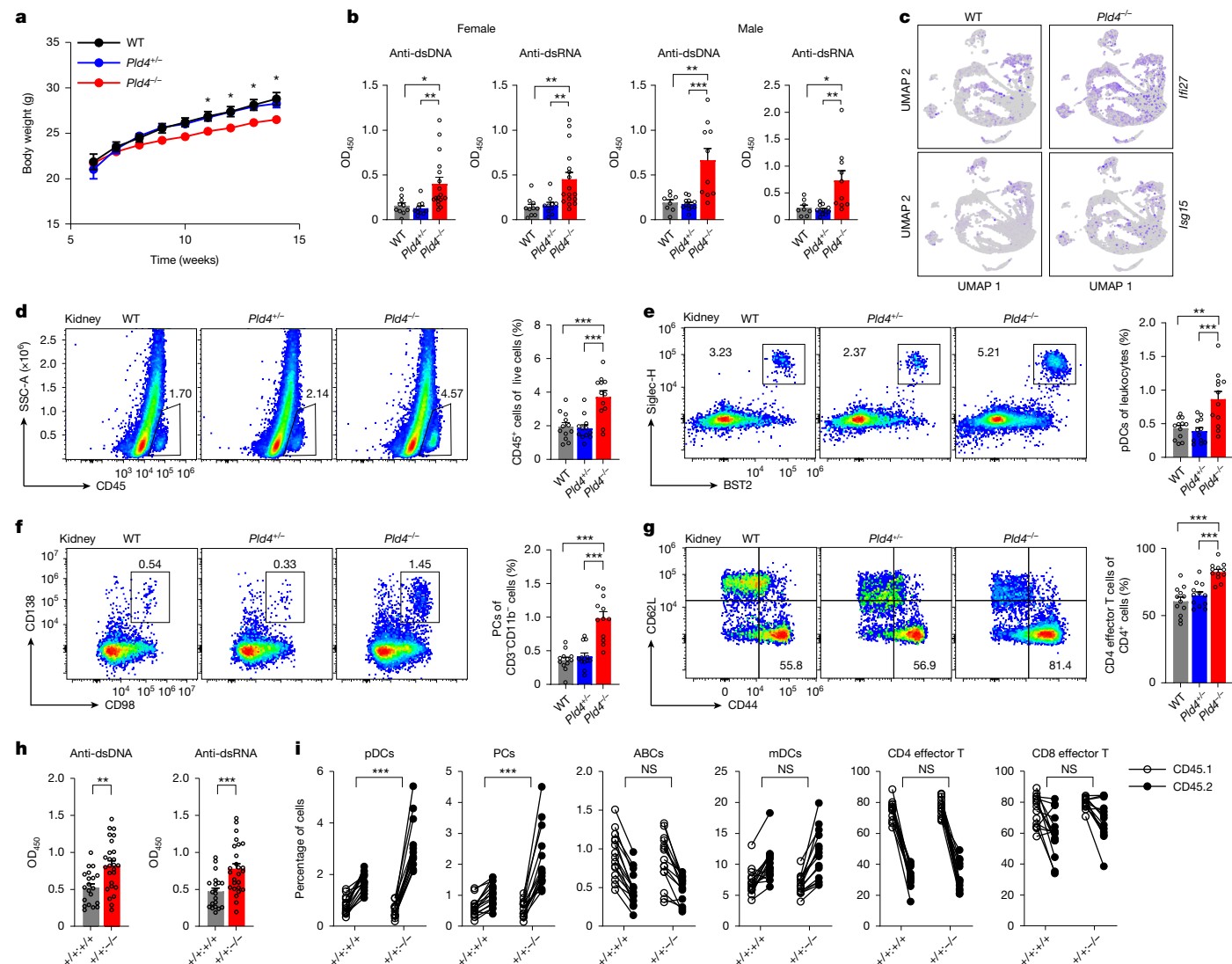

**Fig. 4 | Pld4 deficiency in mice results in autoimmunity and cell-intrinsic expansion of pDCs and plasma cells. a,b**, The autoimmune phenotypes in *Pld4*-deficient mice. **a**, Body-weight change between *Pld4*⁻/⁻ and wild-type mice. *n* = 11 (WT), *n* = 11 (*Pld4*⁺/⁻) and *n* = 11 (*Pld4*⁻/⁻) mice. The asterisks represent statistical comparisons between *Pld4*⁻/⁻ and wild-type mice. **b**, Plasma anti-dsDNA and anti-dsRNA antibodies levels of *Pld4*⁻/⁻ and wild-type mice. Female: *n* = 10 (WT), *n* = 10 (*Pld4*⁺/⁻), *n* = 17 (*Pld4*⁻/⁻); male: *n* = 8 (WT), *n* = 12 (*Pld4*⁺/⁻) and *n* = 10 (*Pld4*⁻/⁻) mice. OD₄₅₀, optical density at 450 nm. **c**, UMAP visualization of type I IFN pathway genes in the kidney scRNA-seq data of *Pld4*⁻/⁻ and wild-type mice. **d–g**, The kidney flow cytometry results of *Pld4*⁻/⁻ and wild-type mice: immune cells (CD45⁺) (**d**), pDCs (CD45⁺CD3⁻CD19⁻CD11b⁻B220⁺CD11c⁺Siglec-H⁺BST2⁺) (**e**), plasma cells (PCs) (CD45⁺CD3⁻CD11b⁻B220^low/⁻CD138⁺CD98⁺) (**f**) and CD4

effector T cells (CD45⁺CD3⁺CD4⁺CD44⁺CD62L⁻) (**g**). *n* = 12 (WT), *n* = 12 (*Pld4*⁺/⁻) and *n* = 12 (*Pld4*⁻/⁻) mice. **h,i**, The results of mixed bone marrow chimeric mice reconstituted with 1:1 ratio of CD45.1-WT–CD45.2-WT (+/+:+/+) or CD45.1-WT–CD45.2-*Pld4*⁻/⁻ (+/+:−/−) bone marrow. **h**, The plasma anti-dsDNA and anti-dsRNA antibodies levels of chimeric mice. *n* = 20 (+/+:+/+) and *n* = 26 (+/+:−/−). **i**, Kidney immune cell phenotyping of chimeric mice. *n* = 15 (+/+:+/+) and *n* = 15 (+/+:−/−). Data are mean ± s.e.m. The results are representative of at least three independent experiments (**b** and **d–g**), two independent experiments (**a**) and a summary of two independent experiments (**h** and **i**). Statistical analysis was performed using two-way ANOVA (**a** and **i**), one-way ANOVA with Tukey's post hoc analysis (**b** and **d–g**) and unpaired *t*-tests (**h**).

## Discussion

SLE is a complex autoimmune disease, characterized by a range of clinical manifestations and substantial heterogeneity in treatment response and prognosis[27]. The study of monogenic lupus provides insights into the pathogenesis and targeted treatment of SLE[2,28,29]. Among these findings, TLR7, TLR9 and proteins in their pathways have been shown to be crucial for the development of SLE[2,8,20,29]. PLD4 acts as a limiting factor upstream of TLR7/9 and modulates the activation of these pathways[18–20]. Here we identified biallelic loss-of-function mutations in *PLD4* in five patients with SLE, highlighting the pivotal role of endosomal nucleic acid homeostasis dysregulation in monogenic SLE. The identification of *PLD4* as a disease-causing gene in

SLE offers a deeper understanding of the molecular underpinnings of the SLE.

The role of TLR7 in lupus is well recognized[2,8,30–32], but the role of TLR9 remains controversial[9,33–35]. A recent study identified MYD88-independent protective roles and MYD88-dependent proinflammatory role of TLR9, which offers a molecular explanation for understanding its context-dependent complexity[36]. In mouse models of PLD4 deficiency, TLR9 exhibits context-dependent roles across genetic backgrounds. Within the C57BL/6 strain, TLR9-mediated autoinflammation cooperates with TLR7 and cGAS–STING signalling pathways to drive disease pathogenesis in PLD3/PLD4 deficiency mice. Besides, in BALB/c mice, TLR9-driven autoimmunity after PLD4 deficiency is the cause of disease in this background[37]. These findings, combined with

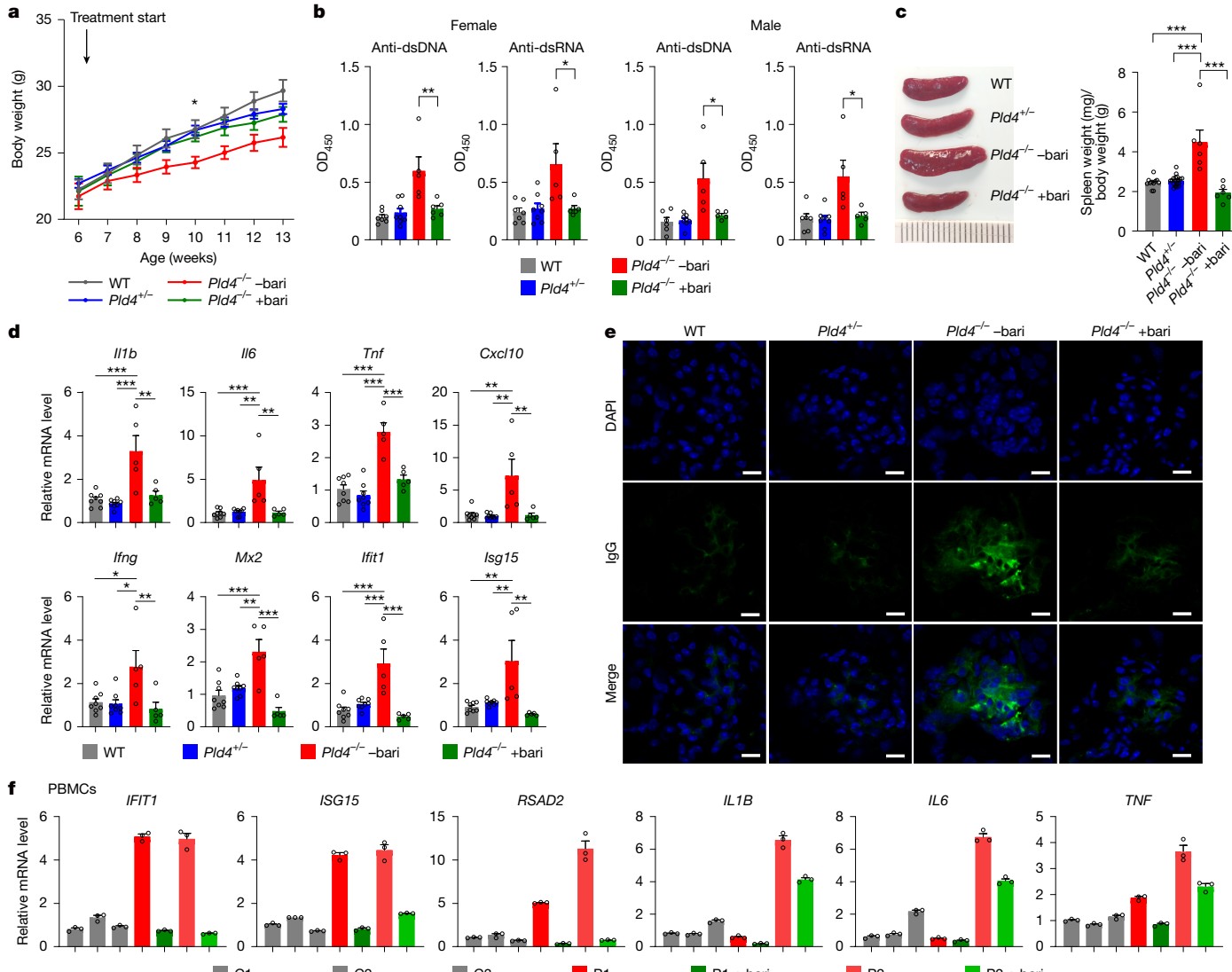

**Fig. 5 | Baricitinib rescues phenotypes in *Pld4*-deficient mice. a–e,** The changes in autoimmune phenotypes in *Pld4*⁻/⁻ mice after baricitinib (bari) treatment. **a,** Body-weight growth curves of different groups of mice. *n* = 6 (WT), *n* = 15 (*Pld4*⁺/⁻), *n* = 4 (*Pld4*⁻/⁻, no baricitinib) and *n* = 4 (*Pld4*⁻/⁻, +baricitinib). The asterisks represent the statistical comparison between *Pld4*⁻/⁻ mice with baricitinib treatment and *Pld4*⁻/⁻ mice without baricitinib treatment. **b,** Plasma anti-dsDNA and anti-dsRNA antibodies levels in *Pld4*⁻/⁻ mice after 8 weeks of baricitinib treatment. Female: *n* = 7 (WT), *n* = 8 (*Pld4*⁺/⁻), *n* = 5 (*Pld4*⁻/⁻, no baricitinib), *n* = 6 (*Pld4*⁻/⁻, +baricitinib); male: *n* = 6 (WT), *n* = 8 (*Pld4*⁺/⁻), *n* = 5 (*Pld4*⁻/⁻, no baricitinib), *n* = 5 (*Pld4*⁻/⁻, +baricitinib). **c,** The spleen size and weight changes in *Pld4*⁻/⁻ mice after 8 weeks of baricitinib treatment. *n* = 11 (WT), *n* = 20

(*Pld4*⁺/⁻), *n* = 6 (*Pld4*⁻/⁻, no baricitinib) and *n* = 6 (*Pld4*⁻/⁻, +baricitinib). **d,** qPCR analysis of renal tissue inflammation changes in *Pld4*⁻/⁻ mice after 8 weeks of baricitinib treatment. *n* = 8 (WT), *n* = 8 (*Pld4*⁺/⁻), *n* = 5 (*Pld4*⁻/⁻, no baricitinib) and *n* = 5 (*Pld4*⁻/⁻, +baricitinib). **e,** IgG staining of kidney glomeruli in *Pld4*⁻/⁻ mice after 8 weeks of baricitinib treatment. Scale bars, 10 μm. **f,** qPCR analysis of type I IFN and NF-κB signalling pathway genes expression of patients P1 and P2 and healthy control PBMCs after 16 h baricitinib treatment. Data are mean ± s.e.m. For **a–f**, the results are representative of two independent experiments. Statistical analysis was performed using two-way ANOVA (**a**) and one-way ANOVA with Tukey's post hoc analysis (**b–d**).

our CpG-DNA stimulation experiments in PLD4-deficient cell lines and patient-derived cells, suggest that substrate accumulation caused by PLD4 deficiency shifts the balance between TLR9 protective effects and TLR9 proinflammatory activity toward the latter.

pDCs are a unique cell population central to antiviral responses through nucleic acid sensing and robust type I IFN production[13,38]. PLD4 exhibits evolutionarily conserved high expression in pDCs[39] (versus PLD3), implicating its non-redundant role in pDC nucleic acid homeostasis. Patients with PLD4 deficiency uniformly develop lupus nephritis, with scRNA-seq and CyTOF analyses suggesting that pDCs are the predominant cellular drivers of upregulated type I IFN and TLR signalling in the patient PBMCs. Notably, plasma cells, a well-established pathogenic cell population in SLE, critically drive disease progression

through the sustained production of autoantibodies. Flow cytometry further reveals expanded plasma cells in patient PBMCs. Mirroring human pathology, PLD4-deficient mice show preferential renal involvement, whereas mixed bone marrow chimeras demonstrate cell-intrinsic expansion of pDCs and plasma cells. This cross-species convergence underscores the central regulatory role of pDCs and plasma cells in PLD4-deficiency-mediated immune dysregulation.

Our study illustrates that although the PLD4–TLR7/9 axis contributes significantly to SLE, different patients have different severity. Besides, monogenic lupus driven by the PLD4–TLR7/9 axis shares similarities with interferonopathies in autoinflammatory diseases[1,2,20,40] and genome-wide association studies have also shown that, in addition to SLE, PLD4 is associated with several other diseases, such as

systemic sclerosis[41] and rheumatoid arthritis[42]. Given the important role of PLD4 in the immune system, it is possible that other phenotypes may be observed in individuals with PLD4 deficiency. The genetic and mechanistic insights from our study may inform the understanding and treatment of a wider spectrum of immune disorders.

Conventional treatments for SLE often lack specificity[43]. According to the significant activation of the type I IFN signalling pathway caused by PLD4 deficiency, we chose to inhibit inflammation by targeting this pathway. Our study demonstrates that baricitinib significantly ameliorates the immune phenotype in PLD4-deficient mice and patient cells. As the initial symptoms of the three patients were arthritis/arthralgia and skin rashes, targeted therapy with baricitinib could be a potential effective intervention to prevent irreversible sequelae during the progression from mild manifestations to SLE outcome. Conserved type I IFN signalling in PLD4-deficient models supports the translational applicability of JAK inhibitor, although interspecies discrepancies necessitate clinical validation in patients with PLD4 deficiency. Overall, *PLD4* genetic screening could become a standard measure for patients with undiagnosed SLE to identify those who may benefit from PLD4-targeted therapies.

In summary, our study identifies a link between PLD4 deficiency and SLE in humans and we denoted the disorder PLD4 deficiency disorder, or PLDD. Our study expands the genetic landscape of SLE and provides compelling evidence for the role of PLD4 in disease pathogenesis. The promising results with baricitinib in PLD4-deficient models highlight the potential of targeted therapy in patients with SLE with PLD4 deficiency, paving the way for more personalized and effective treatment strategies.

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

## Methods

### Case reports

The first patient (P1) was male, with disease-onset at age 12 years. The patient presented with periorbital oedema after tonsillitis, accompanied by reduced urine output and recurrent fevers. Laboratory investigations showed proteinuria, haematuria, hypoalbuminaemia, leukopaenia, mild anaemia and hypocomplementaemia. ANA and anti-dsDNA autoantibodies were positive. Kidney biopsy showed diffuse proliferative lupus nephritis with crescent formation.

The second patient (P2) was female, with disease-onset at age 16 years. The patient had urticaria, alopecia, arthralgia, oedema and recurrent infection. Laboratory investigations showed significant proteinuria, haematuria, acute kidney injury, leukopaenia, thrombocytopenia, autoimmune haemolytic anaemia and hypocomplementaemia. ANA and anti-dsDNA autoantibodies were positive. Kidney biopsy revealed diffuse proliferative lupus nephritis with crescent formation and acute tubular injury.

The third patient (P3) was male, with disease-onset at age 19 years. The patient presented with arthritis, morning stiffness, malar rash, hair loss, photosensitivity, periorbital oedema, gross haematuria and abdominal distension. Laboratory investigations showed hypertension, elevated serum creatinine, massive proteinuria, haematuria, pancytopenia and hypocomplementaemia. ANA and anti-dsDNA autoantibodies, and rheumatoid factor were positive. Kidney biopsy revealed diffuse proliferative lupus nephritis with crescent formation and fibrinoid necrosis. Maintenance haemodialysis began at the age of 40 years.

The fourth patient (P4) was female, with disease-onset at age 49 years. The patient presented with generalized arthralgia, patchy facial erythema with itching and dry mouth. Laboratory evaluations revealed massive proteinuria, haematuria, pancytopenia and hypocomplementaemia. ANA, anti-nRNP/Smith (Sm), anti-Sm and anticardiolipin antibodies were positive. Kidney biopsy revealed membranoproliferative lupus nephritis.

The fifth patient (P5) was female, with disease-onset at age 37 years. The patient presented with patchy rashes with itching, muscle soreness and haemoptysis. Laboratory analysis indicated haematuria, proteinuria, elevated serum creatinine, anaemia and thrombocytopenia, and hypocomplementaemia. ANA and myeloperoxidase–antineutrophil cytoplasmic antibody (MPO–ANCA) were positive. Kidney biopsy showed membranoproliferative lupus nephritis with crescent formation. She began maintenance haemodialysis at the age of 43 years.

Detailed case presentations are provided in the Supplementary information.

### Patients

All of the patients who met the diagnostic criteria for SLE were evaluated at Jinling Hospital. All patients enrolled in the study were evaluated under a protocol approved by the Institutional Review Boards evaluated at Jinling Hospital (2022DZKY-061-01). All patients and family members signed written informed consent.

### PLD4 exonuclease activity

In HEK293T cells, plasmids encoding wild-type and various mutant PLD4 were transfected, followed by collection of protein lysates and purification through Flag-tag magnetic beads (Sigma-Aldrich, M8823). The purified protein or total cell lysates were then subjected to PLD4 enzymatic activity assay[18,19] (50 mM MES pH 5.5, 150 mM NaCl, 2.5 μM substrates, 10 nM or 20 nM purified PLD4), incubated at 37 °C for different time and subsequently analysed by TBE–PAGE, with nucleic acid staining performed for 15 min before imaging.

### Mice and mice treatment

*Pld4*-KO mice (NM-KO-200682), on the C57BL/6 background, were purchased from the Shanghai Model Organisms Center. CD45.1 mice (T054816) were purchased from GemPharmatech. All of the mice were maintained under a specific-pathogen-free environment in the Laboratory Animal Center of Zhejiang University and experimentation was approved by the Institutional Animal Care and Use Committee of Zhejiang University (ZJU20250573). No statistical method was used to calculate sample size. Sample sizes with mice were determined by the availability of animals with the correct genotypes or based on numbers used in previous publications[44] where comparable sample sizes produced statistically significant results. Age- and sex-matched mice were used in each experiment (littermates). Experimenters were blinded to genotypes in the kidney histology pathology analysis. Other data collection and analyses were not performed in a blinded manner to the conditions of the experiment.

*Pld4*-KO mice, aged 6 weeks and with similar body weights, underwent tail vein blood sampling to measure anti-DNA and anti-RNA autoantibodies. On the basis of autoantibody levels and body weights, mice were evenly divided into two groups. One group received daily oral gavage of baricitinib (Selleck, S2851) at a dose of 30 mg per kg per day, while the other group received a solvent gavage to exclude solvent effects. The baricitinib working solution was prepared with 5% baricitinib, 50% PEG 300, 5% Tween-80 and 40% double distilled $H_2O$.

### WES and Sanger sequencing

DNA was extracted from peripheral blood using the Maxwell RSC Whole Blood DNA Extraction Kit (Promega, AS1520), with 1 μg of DNA used for WES. Data alignment was performed using the BWA, and variants were annotated using ANNOVAR (https://annovar.open-bioinformatics.org/en/latest/). Variants were filtered using online databases, including gnomAD (https://gnomad.broadinstitute.org/), dbSNP (https://www.ncbi.nlm.nih.gov/snp/) and Kaviar (https://db.Systemsbiology.net/kaviar/). Subsequently, potential pathogenic variants were selected based on inheritance patterns and the biological functions of the genes. Finally, candidate mutations were confirmed by Sanger sequencing.

### RNA-seq and scRNA-seq

Total RNA was extracted from PBMCs designated for RNA-seq using the QIAGEN RNeasy kit (74104). The RNA quality and purity were assessed using an Agilent Bioanalyzer RNA chip with 1 μg of total RNA. After assessment, the RNA was purified and fragmented, followed by the construction of an mRNA library using the Illumina TruSeq RNA Sample Preparation Kit V2. The RNA was then reverse-transcribed into cDNA, with an average fragment size of approximately 200 bp. Subsequent steps included end-repair, adding an A base to the 3′ ends, adaptor ligation and PCR amplification. RNA-seq was conducted on the Illumina NovaSeq platform. Sequencing data were aligned using HISAT2 in human reference genome (GRCh38), with reads counting performed by featureCounts. Differential gene expression analysis was conducted using DESeq2, and downstream heat-map visualization was performed using the *R* package pheatmap.

scRNA-seq used in this study was performed with samples from two sources: human PBMCs and mouse kidney cells. Human PBMCs were counted directly for library construction after separation. The mice kidney tissue was digested with collagenase IV for 120 min. After single-cell counting, the procedure was as follows: cells were uniformly mixed with gel beads using the 10x Genomics single-cell sequencer to prepare oil droplets encapsulating the cells, causing cell lysis, RNA release and reverse transcription. Adaptors were added to the cDNA from each cell, followed by PCR amplification and single-cell libraries were prepared for sequencing. The scRNA-seq analysis workflow was as follows: raw data were processed using Cell Ranger to count reads and generate expression matrices for different transcripts in each cell. The expression matrices were then quality-controlled, dimensionally reduced, annotated and visualized using the Seurat package[45] in *R*.

## CyTOF analysis

The PBMCs from both patients and healthy controls were pretreated with brefeldin A (BFA) for 4 h to block cytokine secretion and stained with 250 nM cisplatin for 5 min to label dead cells. Subsequently, cells were incubated with an Fc receptor blocking agent and a mixture of surface antibodies for 30 min at room temperature, followed by fixation and permeabilization, and further stained with intracellular detection antibodies on ice for 30 min before acquisition.

CyTOF data were initially processed using FlowJo to remove dead cells, duplicate cells and background noise. Cells were subdivided into distinct subgroups based on the expression levels of marker genes. High-dimensional data were transformed into two-dimensional representations using $t$-SNE dimensional reduction analysis. The data were then annotated based on marker gene expression and compared for the expression levels of target proteins.

## Cell preparation, culture and stimulation

HEK293T and THP-1 cell lines, obtained from the American Type Culture Collection, were cultured and stimulated as follows: HEK293T cells were maintained in Dulbecco's modified Eagle medium (Thermo Fisher Scientific, C11995500CP) supplemented with 10% FBS (Noverse, NFBS-2500A) and 1% penicillin−streptomycin (Thermo Fisher Scientific, 15140163). THP-1 cells and PBMCs were maintained in RPMI-1640 (Thermo Fisher Scientific, C11875500CP) supplemented with 10% FBS and 1% penicillin−streptomycin. PBMCs were isolated from whole blood using lymphocyte-separation medium (LSM, MPbio, 0850494) through density-gradient centrifugation according to the manufacturer's instructions. The THP-1 *PLD4*-KO cell line was generated using the CRISPR−Cas9 system by infecting THP-1 wild-type cells with sgPLD4 virus packaged in HEK293T cells, followed by selection with puromycin to establish stable KO cell lines.

In the phosphorylation flow cytometry experiments on PBMCs, cells were stimulated with 5 µM CpG-DNA (ODN 2216, InvivoGen, tlrl-2216) for 20 min. For qPCR and RNA-seq experiments in PBMCs, stimulation was performed with 4 µM CpG-DNA for 24 h. For intracellular cytokine staining experiments, PBMCs were stimulated with 2 µM CpG-DNA for 24 h, with BFA added 6 h before sampling to block cytokine secretion. In the CyTOF experiments on PBMCs, cells were treated with BFA for 4 h. In the qPCR experiments on PBMCs treated with baricitinib, the baricitinib concentration was 0.5 µM for 16 h treatment. In qPCR experiments on the THP-1 *PLD4* polyclonal KO cell line, cells were stimulated with 5 µM CpG-DNA for 6 h. For STING inhibition experiment, THP-1 *PLD4* monoclonal KO cells were treated with 5 µM H-151 or C-176 for 48 h, then subjected to western blotting or qPCR.

## Plasmids and antibodies

PCR amplification of *PLD4* cDNA from healthy control PBMCs was used to construct the human *PLD4* plasmid. Site-directed mutagenesis was used to generate mutant plasmids. Western blotting, flow cytometry and immunofluorescence were performed using a variety of antibodies: NF-κB p65 (Cell Signaling Technology, 8242), p-NF-κB p65 (Cell Signaling Technology, 3033), p44/42 MAPK (Cell Signaling Technology, 4696), p-p44/42 MAPK (Cell Signaling Technology, 4370), STAT1 (Cell Signaling Technology, 14994), p-STAT1 (Cell Signaling Technology, 9167), p-STAT2 (Tyr690) (Cell Signaling Technology, 88410), STAT2 (Cell Signaling Technology, 72604), β-actin (Cell Signaling Technology, 4970), GAPDH (Cell Signaling Technology, 2118), TLR7 (Cell Signaling Technology, 5632), p-STING (Ser366) (Cell Signaling Technology, 50907), STING (Cell Signaling Technology, 13647), HRP-conjugated anti-DYKDDDDK tag (Flag) (Huabio, 0912-3), CD3-APC-H7 (BD Biosciences, 560176), CD4-FITC (BD Biosciences, 555346), CD8-APC (BD Biosciences, 561952), CD14-PE-CY7 (BD Biosciences, 557742), CD19-BB700 (BD Biosciences, 566396), PE mouse anti-human IFNα (BD Biosciences, 560097), BD Pharmingen p38 MAPK (pT180/pY182) PE (BD Biosciences, 612565), BD Phosflow

BV421 anti-human NF-κB p65 (pS529) (BD Biosciences, 565446), Alexa Fluor 647 anti-p-ERK1/2 (BioLegend, 369504), PLD4 (Thermo Fisher Scientific, PA5-98680), anti-C3 (Abcam, ab11862), FITC anti-human CD21 (BioLegend, 354910), FITC anti-human CD24 (BioLegend, 311104), PerCP/Cyanine5.5 anti-mouse/rat/human CD27 (BioLegend, 124214), Brilliant Violet 605 anti-human CD38 (BioLegend, 303532), Brilliant Violet 421 anti-human IgG Fc (BioLegend, 410704), APC anti-human HLA-DR (BioLegend, 307610), Brilliant Violet 570 anti-mouse CD11c (BioLegend, 117331), V500 mouse anti-mouse CD45.2(104) (BD Biosciences, 562129), Brilliant Violet 650 anti-mouse CD45.1 (BioLegend,110736), BV421 hamster anti-mouse CD3e (BD Biosciences, 562600), FITC rat anti-mouse CD4 (BD Biosciences, 553046), APC-Cy7 rat anti-mouse CD8a (BD Biosciences, 557654), APC rat anti-mouse CD19 (BD Biosciences, 550992), PE rat anti-mouse CD138 (BD Biosciences, 553714), BV605 CD317 (BD Biosciences, 747606), PE-Cy7 rat anti-mouse CD45R/B220 (BD Biosciences, 552772), BUV496 CD11b (BD Biosciences, 749864), BV750 F4/80 (BD Biosciences, 747295), RB780 Ly-6C (BD Biosciences, 755871), BUV395 I-A, I-E (BD Biosciences, 569244), BUV805 CD44 (BD Biosciences, 741921), BUV563 CD62L (BD Biosciences, 741230), BV786 rat anti-mouse CD25 (BD Biosciences, 564023), UV737 rat anti-mouse CD21/CD35 (BD Biosciences, 612810), RB545 CD23 (BD Biosciences, 756344), BV480 CD95 (BD Biosciences, 746755), BUV615 CD49b (BD Biosciences, 751052), BV711 CD279 (BD Biosciences, 744547), PE-CF594 rat anti-mouse CD185 (CXCR5) (BD Biosciences, 562856), R718 Ly-6G (BD Biosciences, 567039), PERCPEF710 BCL-6 (Invitrogen, 46-5453-82), RB744 rat anti-mouse Siglec-H (BD Biosciences, 757466) and BUV661 rat anti-mouse CD98 (BD Biosciences, 752893).

## Flow cytometry analysis

In experiments assessing the changes in phosphorylation levels of key proteins in inflammatory signalling pathways and intracellular inflammatory cytokines in PBMCs, cells were plated at a density of $1.5 \times 10^6$ cells per ml and stimulated with CpG-DNA for either 20 min or 24 h. After the removal of the stimulus, surface antibodies were added, and the cells were stained at room temperature for 30 min. After PFA fixation and permeabilization, intracellular antibodies were added, and the cells were stained at room temperature for 1 h before proceeding to flow cytometry analysis.

Single-cell suspensions from mouse spleens were prepared by gently triturating the tissue with the plunger end of a syringe and filtering the resultant mixture through 40 µm filters with 2% FBS in PBS (2% FBS−PBS). The filtrate was then centrifuged at 350$g$ for 5 min. The cell pellet was resuspended in 2 ml of RBC lysis buffer and incubated at 25 °C for 3 min. The suspension was then diluted with 10 ml 2% FBS−PBS. After a second centrifugation at 350$g$ for 5 min, the spleen cells were resuspended in PBS.

For mouse kidney single-cell suspension preparation, half of the kidney was minced into 1–2 mm³ pieces using surgical curved scissors in a dish and transferred to a 15 ml centrifuge tube. To this, 2 ml of digestion solution (1 mg ml⁻¹ collagen IV, 200 µg ml⁻¹ DNase I) was added, and the tissue was incubated at 37 °C for 30 min. Digestion was then halted by the addition of 10% FBS−PBS and the resultant mixture was filtered through 40 µm filters. The mixture was subsequently centrifuged and resuspended in 2 ml of RBC lysis buffer according to the spleen single-cell suspension protocol.

## Immunofluorescence

For immunofluorescence staining of mouse kidney tissue cryosections, the sections were first fixed and permeabilized, followed by blocking with freshly prepared 10% normal goat serum (NGS, Beyotime, C0265)/ PBS for 1 h. Mouse antibodies diluted in 10% NGS/PBS (IgG 1:200; C3 1:200) were applied to the sections and incubated overnight at 4 °C. Secondary antibodies diluted at 1:500 in 10% NGS/PBS were then incubated with the sections at room temperature for 1 h. Finally, the sections were imaged using a Zeiss 710 inverted microscope.

## Western blotting and immunoprecipitation

For western blotting, cells were lysed on ice for 20 min in NP-40 lysis buffer containing protease and phosphatase inhibitors (Thermo Fisher Scientific, 78442), followed by centrifugation at 12,000$g$ for 10 min at 4 °C. The supernatant was mixed with SDS sample buffer, heated at 95 °C for 5 min, then subjected to separation by SDS–PAGE.

For immunoprecipitation of endogenous PLD4 in PBMCs, homemade PLD4 antibody whole immune serum was used at a 1:50 ratio to immunoprecipitate 500 μg of PBMC protein. The immunocomplexes were then eluted with 0.2 M glycine at pH 2.5 and subsequently neutralized using 1.0 M Tris-HCl at pH 8.0.

## RT–qPCR

qPCR with reverse transcription (RT–qPCR) was performed using ABclonal's 2× Universal SYBR Green Fast qPCR Mix (RK21203), Vazyme HiScript IV All-in-One Ultra RT SuperMix for qPCR(R433), ChamQ Blue Universal SYBR qPCR Master Mix (Q312-02) and Roche's Light-Cycler480 qPCR system to measure mRNA expression levels in various cell or tissue samples. Fluorescence signal intensities from the collected samples and primers were used to obtain $C_t$ values. Relative expression levels were normalized to the reference gene *GAPDH/Gapdh* and calculated using the $\Delta\Delta C_t$ method.

## ELISA

This study primarily conducted two types of enzyme-linked immunosorbent assay (ELISA): one for the detection of cytokines in cell lines and another for coating DNA/RNA to detect autoantibodies in mouse plasma, with similar steps for both. Initially, an appropriate amount of coating buffer was used to dilute coating antibodies or antigens (1 μg DNA or RNA), which were then added to the wells of an ELISA plate and incubated overnight at 4 °C. Subsequently, the plate was blocked with diluent buffer to reduce non-specific binding and incubated at room temperature for 1 h. Samples and standards were then diluted in diluent buffer and incubated at room temperature for 2 h. Detection antibodies were diluted in diluent buffer and further incubated for 1 h at room temperature. This was followed by dilution of HRP conjugate (Beyotime, A0216) in diluent buffer and a further 0.5 h incubation at room temperature. TMB substrate (Beyotime, P0209) was added and incubated for 30 min. Finally, stop solution (Beyotime, P0215) was added before measuring the absorbance at 450 nm using a microplate reader.

## Bone marrow chimeras

B6-CD45.1(Ptprc-p.K302E) were taken lethally irradiated (7 Gly), then reconstituted with mixed bone marrow ($5 \times 10^6$ cells, 1:1 ratio of B6-CD45.1 and either wild-type or Pld4-dificient (B6-CD45.2)). After 8 weeks of reconstitution, mice were euthanized, and plasma was collected for ELISA analysis, while kidneys and spleens were collected for flow cytometry analysis.

## Statistical analysis

Data analysis and graphing were performed using GraphPad Prism 8, R v.3.5.2 and PyMOL (v.3.1)[46]. Data for statistical tests were derived from three or more independent experiments. Mouse, cell and human-derived data are presented as mean ± s.e.m. For comparisons between two groups, unpaired *t*-tests were used for significance analysis; for comparisons involving more than two groups, ANOVA was used; $P < 0.05$ was considered to be statistically significant.

## Reporting summary

Further information on research design is available in the Nature Portfolio Reporting Summary linked to this article.

## Data availability

The raw RNA-seq data reported in this paper have been deposited in the Genome Sequence Archive[47] in National Genomics Data Center[48], China National Center for Bioinformation/Beijing Institute of Genomics, Chinese Academy of Sciences (GSA-Human: HRA012523). Gel source data are shown in Supplementary Fig. 1. All data supporting the findings of this study are available in the Article and its Supplementary Information, or from the corresponding authors on reasonable request. Source data are provided with this paper.

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

**Acknowledgements** The work was supported by the National Natural Science Foundation of China (32141004, 82225022, 32321002, 82394424, 82170739, 82471844, 82394420); the Hundred-Talent Program of Zhejiang University, Leading Innovative and Entrepreneur Team Introduction Program of Zhejiang (grant no. 2021R01012); the National Key Research and Development Program of China (2024YFC2511002); the Basic Research Program of Jiangsu Province (BK20243061); and the Jiangsu Provincial Science and Technology Resources Coordination Service Platform (BM2015004-1).

**Author contributions** Z.L., X.Y. and Q.Z. directed and supervised the research. Q.W., H.Z., X.S., C.Z. and S.M. contributed equally. Q.W., H.Z., X.S., J.P., L.L., Y.Z. and C.G. performed experiments. H.Z. and Q.Z. identified *PLD4* mutations in patients with SLE. Q.W. and X.S. performed the exonuclease activity of PLD4 assays, animal experiments with baricitinib treatment and the mixed bone marrow chimera experiments. R.W. provided suggestions on confocal imaging. Q.W., H.Z., S.M., J.F. and C.L. analysed the data. Z.L., C.Z., and Y.J. enrolled the patients, and collected and interpreted clinical information. Q.W., Q.Z., X.S. and C.Z. wrote the manuscript, with input from the other authors. All of the authors contributed to the review and approval of the manuscript.

**Competing interests** The authors declare no competing interests.

**Additional information**
**Correspondence and requests for materials** should be addressed to Qing Zhou, Xiaomin Yu or Zhihong Liu.

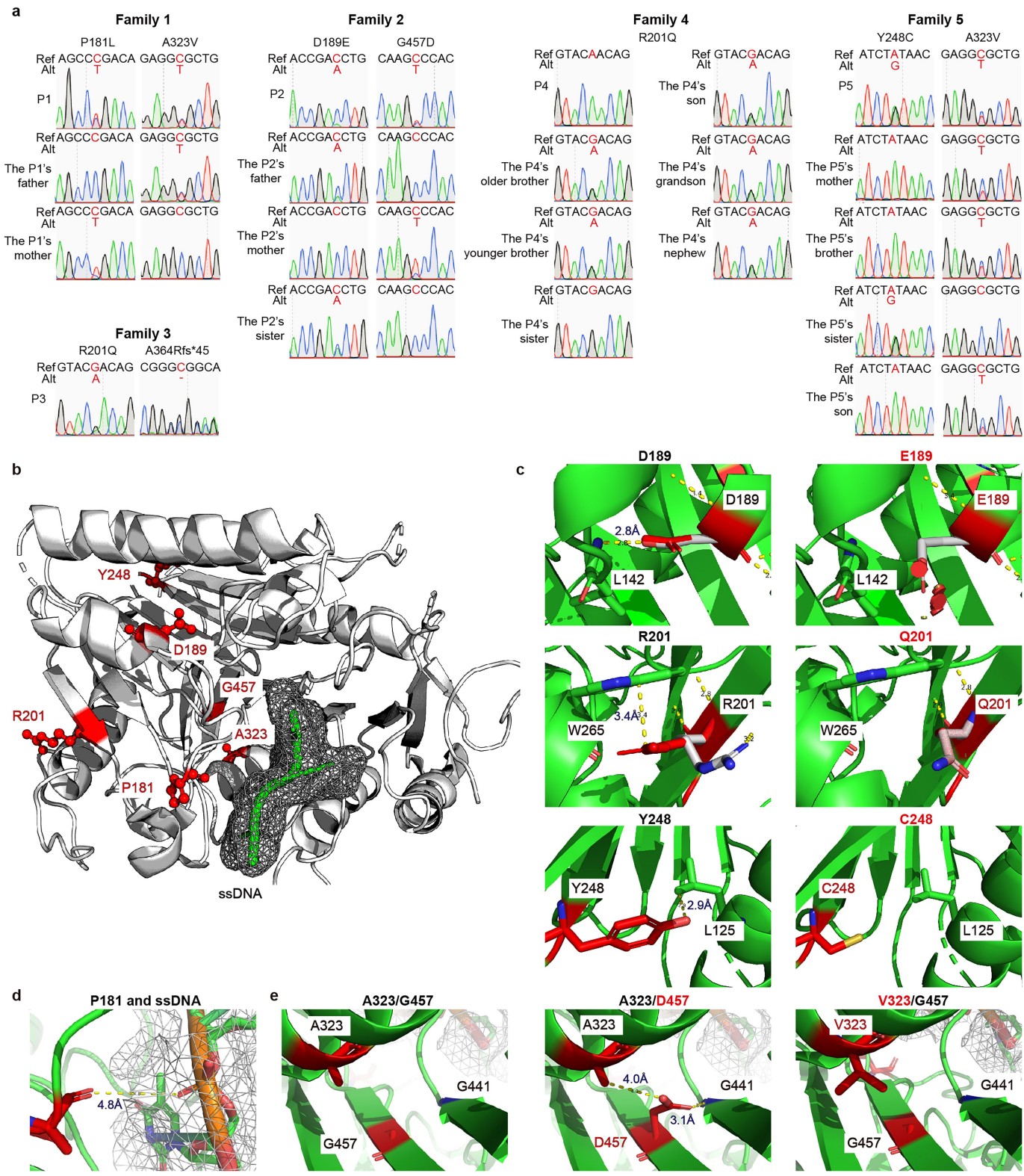

**Extended Data Fig. 1 | Sanger sequencing verification and structure prediction of PLD4 mutations. a**, Sanger sequencing verification of *PLD4* mutations in five families. **b**, Schematic diagram of the position of the mutations site in the PLD4 protein structure (PDB: 8V08). **c-e**, Potential effects of spatial structural simulation mutations on PLD4 protein function, including the disruption of hydrogen bonding with neighbouring amino acid residues (**c, e**) and altered DNA-binding capacity (**d**).

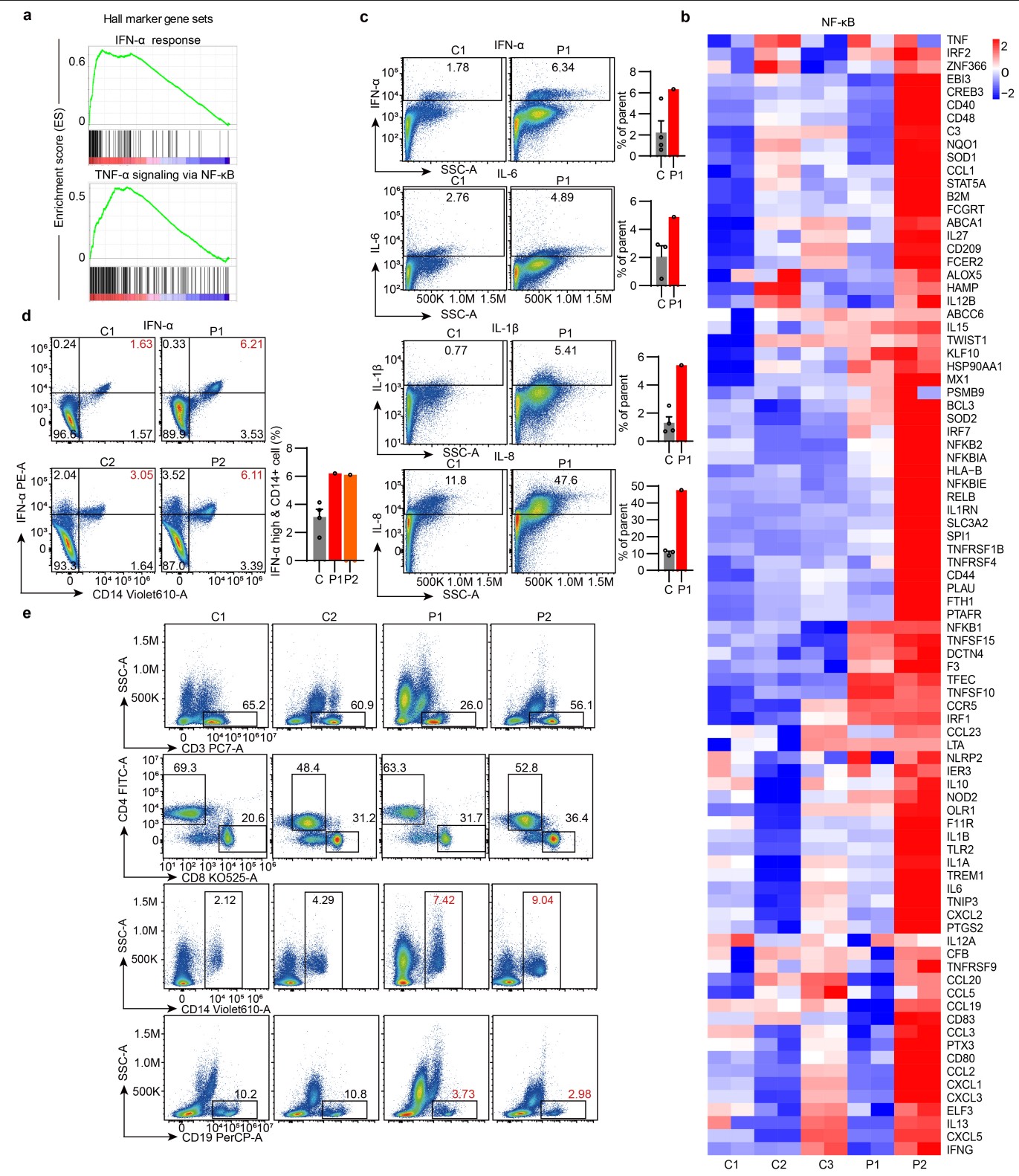

**Extended Data Fig. 2 | Inflammatory pathway upregulated in patients' PBMCs. a, b**, Representative the RNA-Seq results of PBMCs from patients and healthy controls. **a**, The most significantly enriched inflammatory signalling pathways in GSEA of patients' PBMCs. **b**, Heatmap of NF-κB pathway involved genes in the RNA-Seq data of the PBMCs from patients P1, P2 and healthy controls. **c-e**, Representative Flow cytometry analysis of the differences in inflammatory cytokine levels and major cell population proportions in PBMCs from patients and healthy controls. **c**, Flow cytometry of the increased expression levels of IFN-α, IL-6, IL-1β and IL-8 in the PBMCs from patient P1 and healthy controls. C: n = 4, in IFN-α and IL-1β staining, C: n = 3 in IL-6 and IL-8 staining. **d**, Flow cytometry analysis of upregulated IFN-α in CD14 positive cells from patients P1, P2 and healthy controls. C: n = 4. Data are mean ± s.e.m. **e**, Flow cytometry analysis of CD3, CD4, CD8, CD14 and CD19 staining of the PBMCs from patients and healthy controls. The cells selected in the black frame are positive cells, and the number is the proportion of positive cells in the parent group of cells. These results show a representative of three independent experiments in (**c, d, e**).

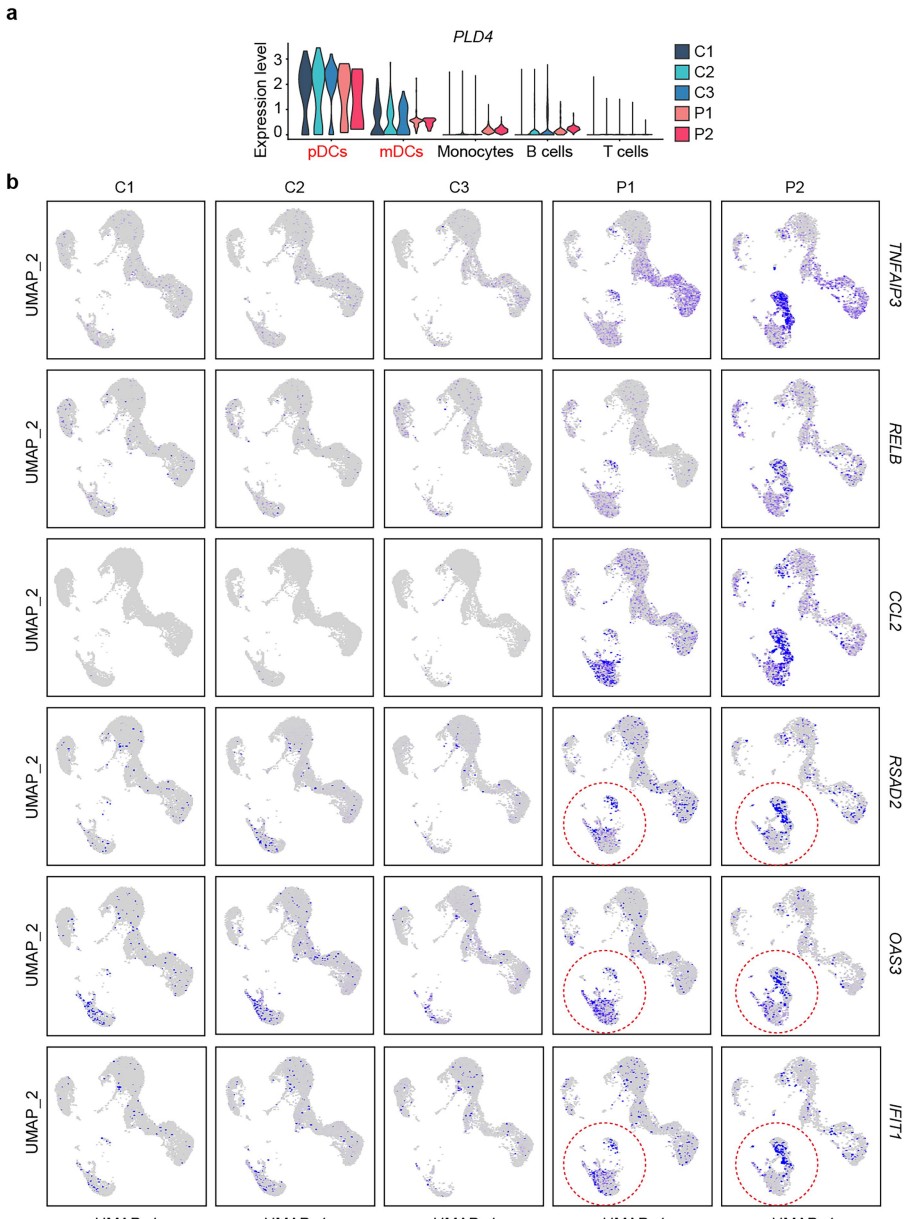

**Extended Data Fig. 3 | The expression of *PLD4*, NF-κB and type I interferon pathways genes in the PBMCs from patients and healthy controls. a**, Violin plot of the expression pattern of *PLD4* in the PBMCs from patients P1, P2 and healthy controls. pDC: plasmacytoid dendritic cell; mDC: myeloid dendritic cell. **b**, UMAP plot of gene expression levels in NF-κB and type I interferon signalling pathways of the PBMCs from patients and healthy controls. The red circle indicates PLD4 specifically expressing cells (DCs and monocytes) and inflammation-induced LDGs.

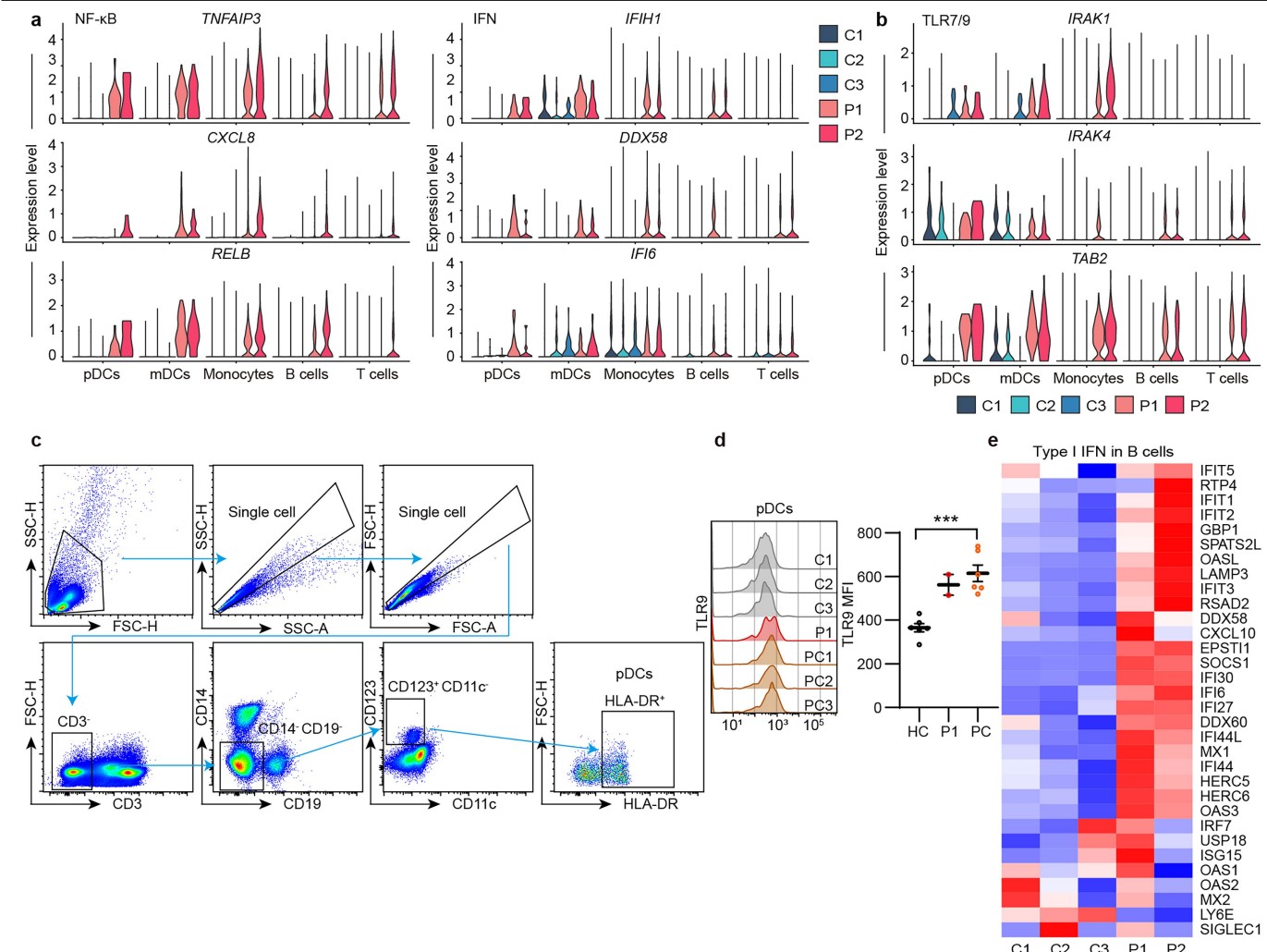

**Extended Data Fig. 4 | scRNA-Seq shows significant activation of TLRs and downstream inflammatory pathways in patients' DCs. a,b,e** Representative the scRNA-Seq analysis of type I interferon and TLR signalling pathway across different cell populations in PBMCs from patients P1, P2 and healthy controls. **a**, Violin plot of the expression of genes in NF-κB and type I interferon signalling pathways in major cell populations of the PBMCs from patients P1, P2 and healthy controls. pDC: plasmacytoid dendritic cell; mDC: myeloid dendritic cell. **b**, Violin plot of the upregulated expression of genes in TLR7/9 signalling pathway in major cell populations of the PBMCs from patients P1, P2 and healthy controls. TLR7/9: TLR7/9 signalling pathway related genes. **c**, Representative pDCs gating strategies from human PBMCs. **d**, Flow cytometry plot and quantification of TLR9 expression in pDCs from patient P1, healthy controls and SLE patients controls. PC: SLE patients control without *PLD4* mutations. HC: n = 6, PC: n = 6, dots in patient group represent different time samples for a patient. Data are mean ± s.e.m. **e**, Heatmap of upregulated type I interferon pathway involved genes in patients and healthy controls' B cells in scRNA-Seq. IFN: interferon. The results show a summary of two independent experiments in (**d**). ***P < 0.001. One-way ANOVA with Tukey's post hoc analysis was used in (**d**).

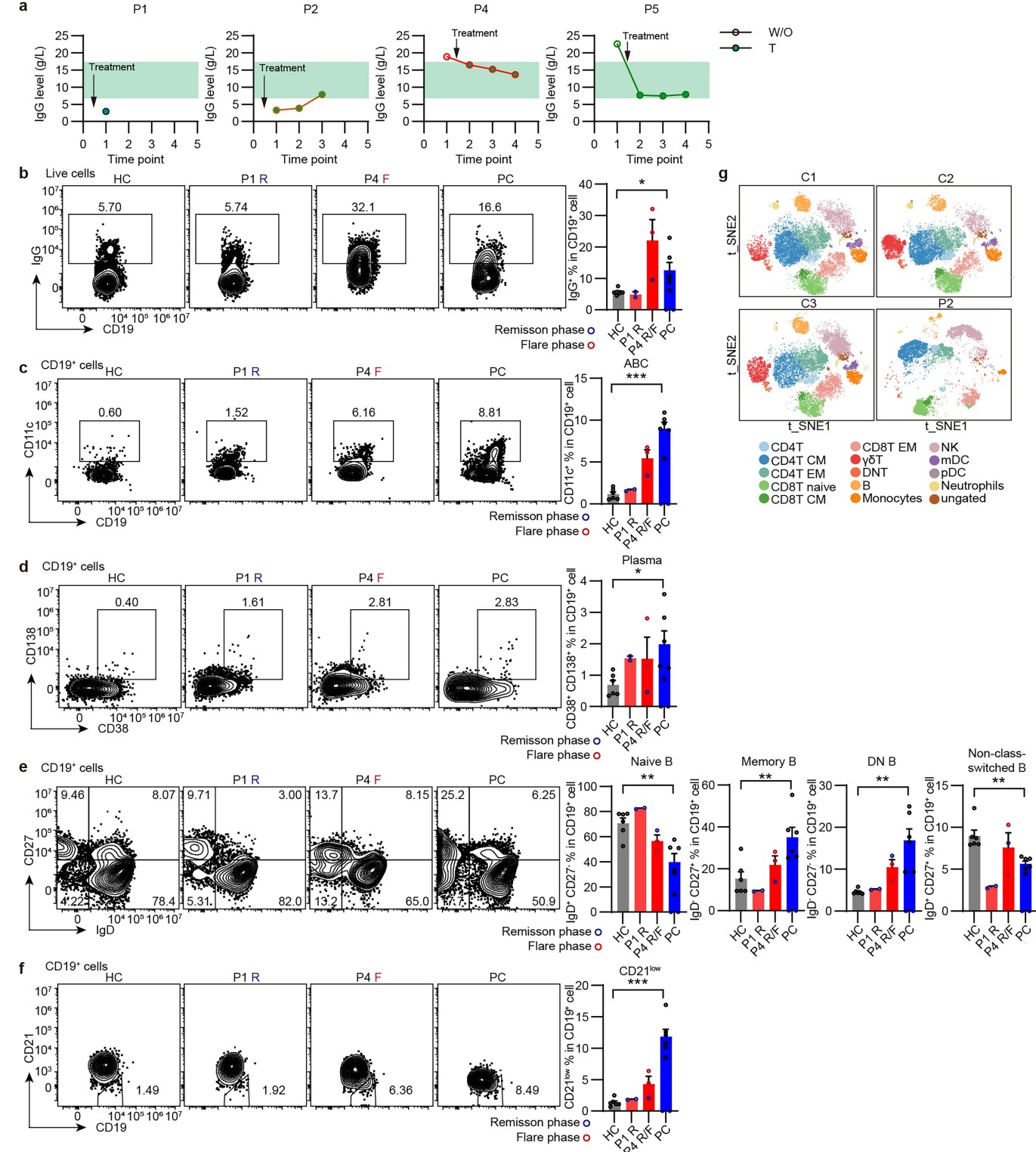

**Extended Data Fig. 5 | B-cell aberrations in patients with PLD4 deficiency.**
**a**, IgG levels of four patients at different stages. The green box represents the
normal level of IgG. W/O: without treatment. T: treatment with steroids or
multiple immunosuppressants. **b-f**, Representative the flow cytometry plot
and quantification of IgG level (CD19⁺IgG⁺, **b**), ABCs (CD19⁺CD11c⁺, **c**), PCs
(CD19⁺CD38⁺CD138⁺, **d**), Naïve B cells (CD19⁺CD27⁻IgD⁺, **e**), Memory B cells
(CD19⁺CD27⁺IgD⁻, **e**), Double negative B cell (CD19⁺CD27⁻IgD⁻, **e**), non-class-
switched B cells (CD19⁺CD27⁺IgD⁺, **e**) and CD21^low B cells (CD19⁺CD21^low, **f**) in
patients P1, P4, healthy controls and SLE patients controls. HC: Healthy controls;

P1 R: patient P1 remission phase; P4 R/F: patient P4 remission phase /
flare phase; PC: SLE patients control without *PLD4* mutations. Blue circles in
patients P1 or P4: remission phase in patients P1 or P4; Red circles in patient P4:
flare phase in patient P4. HC: n = 6, PC: n = 6, dots in patient group represent
different time samples for a patient. Data are mean ± s.e.m. **g**, t-SNE plot of
CyTOF depicting differences of various cell types between patient P2 and
healthy controls. These results show a representative of two independent
experiments in (**b,c,d,e,f**). *P < 0.05; **P < 0.01; ***P < 0.001. unpaired *t*-test was
used in (**b,c,d,e,f**).

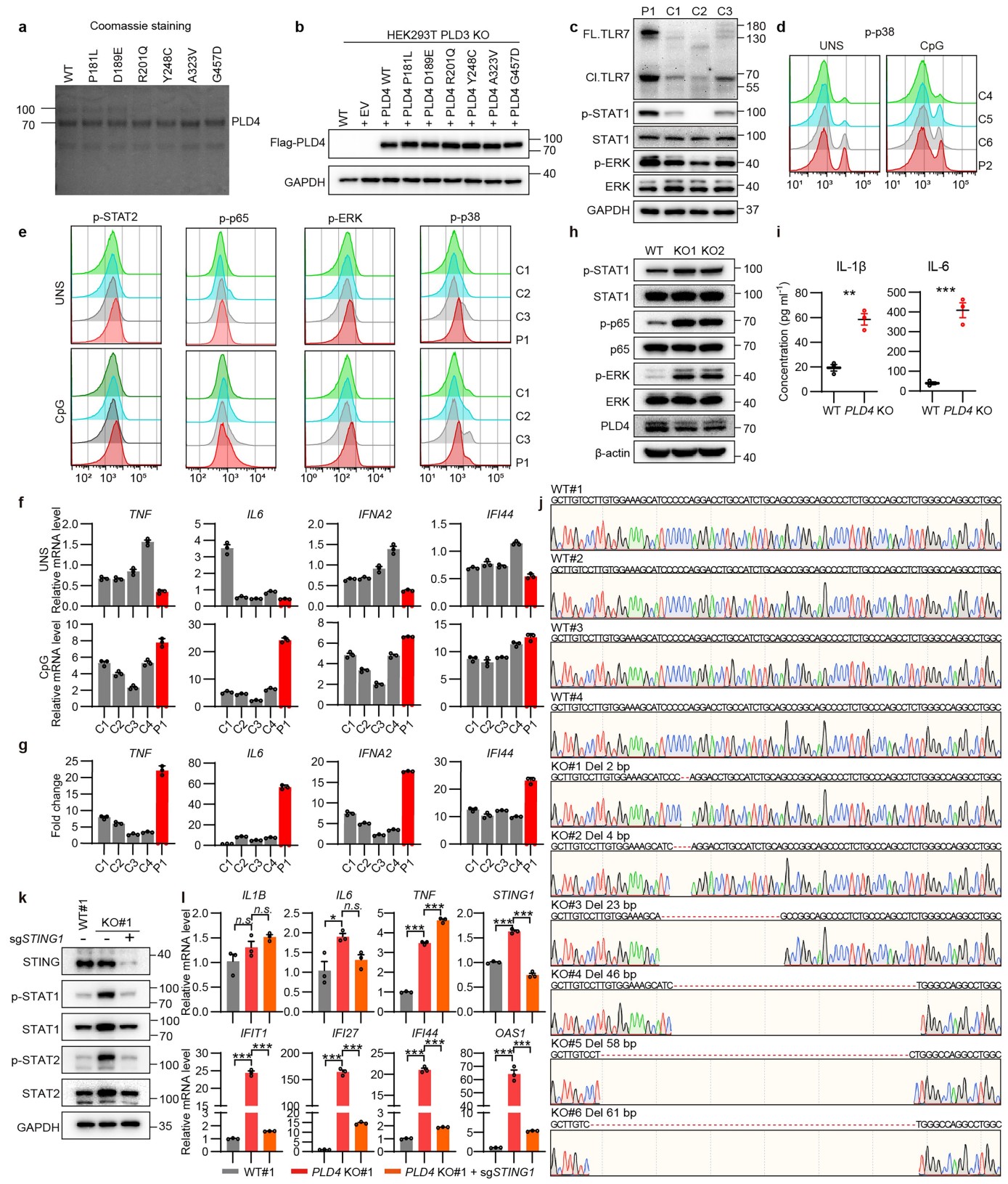

**Extended Data Fig. 6** | See next page for caption.

**Extended Data Fig. 6 | PLD4 deficiency leads to aberrant activation of downstream inflammatory signalling pathways. a**, Coomassie staining of purified WT and mutant PLD4 from HEK293T cell. **b**, Western blotting of HEK293T *PLD3* KO cells overexpressing WT and mutant PLD4. **c**, Western blotting of TLR7 and downstream signalling pathways in the PBMCs from patient P1 and healthy controls. Fl.TLR7: full-length TLR7; Cl.TLR7: cleavage TLR7. **d,e**, Flow cytometry analysis in the phosphorylation of key proteins in the inflammatory pathway in PBMCs of patients P1 (**e**), P2 (**d**) compared to healthy controls. UNS: unstimulated; CpG: CpG-DNA stimulation. **f, g**, NF-κB and type I interferon pathways genes expression in the monocytes from patient P1 and healthy controls under untreated (**f, upper**) and CpG-DNA stimulation (**f, lower**). **g**, Fold induction of genes in (**f**) following CpG-DNA stimulation. **h**, Western blotting of inflammatory pathways activation in THP-1 *PLD4* KO cells. **i**, The inflammatory cytokines levels in supernatant from THP-1 *PLD4* KO cells cultured for 24 h. WT and *PLD4* KO: n = 3. **j**, Sanger sequencing verification of THP-1 *PLD4* monoclonal KO cells. **k,l**, Representative the change of signalling pathways and downstream inflammatory genes in THP-1 *PLD4* monoclonal KO cell after STING KO. **k**, Western blotting of signalling pathways change after STING KO. **l**, NF-κB and type I interferon pathway genes expression after STING KO. WT, *PLD4* KO and *PLD4* KO + sg*STING1*, n = 3. Data are mean ± s.e.m. These results show a representative of three independent experiments in (**a,b,h,i,k,l**), two independent experiments in (**d,e**). Experiment in (**c,f,g**) was performed once. *n.s.*, no significant difference; *$P < 0.05$; **$P < 0.01$; ***$P < 0.001$. One-way ANOVA with Tukey's post hoc analysis was used in (**l**), unpaired *t*-test was used in (**i**).

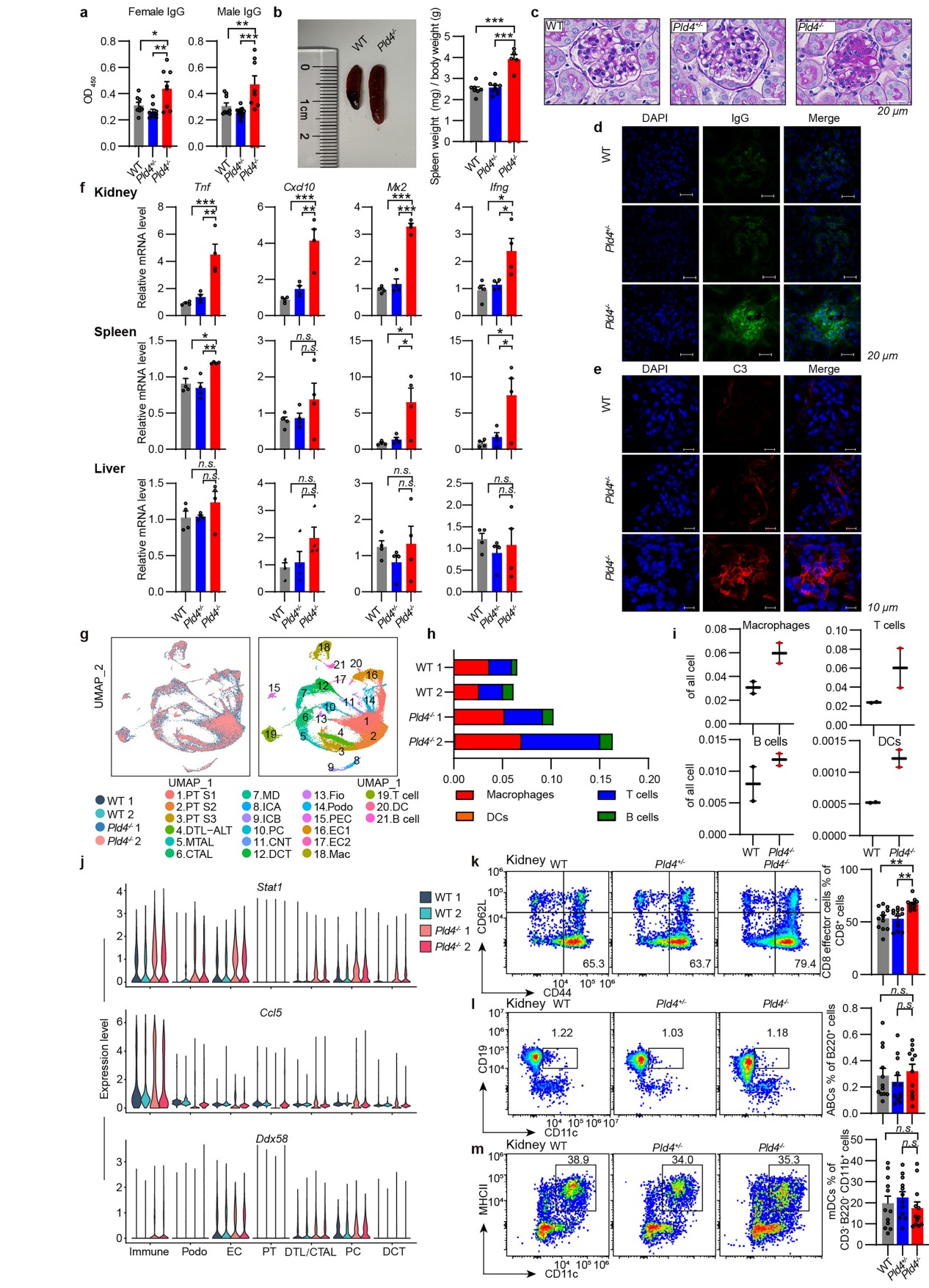

**Extended Data Fig. 7** | See next page for caption.

**Extended Data Fig. 7 | Pld4-deficient mice manifests autoimmunity and nephritic phenotypes. a**, Plasma IgG levels of WT and $Pld4^{-/-}$ mice. WT: $Pld4^{+/-}$: $Pld4^{-/-}$ = 8: 9: 8 in female and 9: 12: 8 in male. **b**, The differences in spleen length and weight between WT and $Pld4^{-/-}$ mice. WT: $Pld4^{+/-}$: $Pld4^{-/-}$ = 7: 9: 5. **c-e**, Representative the PAS (**c**), IgG (**d**) and C3 (**e**) staining of renal glomeruli in WT and $Pld4^{-/-}$ mice. Scale bar: 20 μm in (**c-d**), 10 μm in (**e**). **f**, Inflammatory genes expression in kidney, spleen and liver of WT and $Pld4^{-/-}$ mice. WT: $Pld4^{+/-}$: $Pld4^{-/-}$ = 4: 4: 4. **g-j**, Representative the scRNA-Seq results in the kidney of WT and $Pld4^{-/-}$ mice. **g**, UMAP plots show the cell type differences. **h,i**, The proportion of total or various types of renal immune cells. **j**, Violin plot of the type I interferon pathway genes in major renal cell populations. Podo: podocytes; EC: endothelial cells; PT: proximal convoluted tubule; DTL/CTAL: descending limb of loop of Henle/thick ascending cortical limb; PC: principal cells; DCT: distal convoluted tubule. **k-m**, Representative the kidney flow cytometry plot and quantification of WT and $Pld4^{-/-}$ mice: CD8 Effector T cells ($CD45^+CD3^+CD8^+CD62L^-CD44^+$) in (**k**), ABCs ($CD45^+CD3^-CD21^-CD23^-B220^+CD19^{high}CD11c^+$) in (**l**), mDCs ($CD45^+CD3^-B220^-CD11b^+CD11c^+MHC\text{-}II^+$) in (**m**). WT: $Pld4^{+/-}$: $Pld4^{-/-}$ = 12: 12: 12. Data are mean ± s.e.m. These results show a representative of at least three independent experiments in (**c, d, e, f, k, l, m**), two independent experiments in (**b**). Experiment in (**a**) was performed once with n > 8. *n.s.*, no significant difference; *$P$ < 0.05; **$P$ < 0.01; ***$P$ < 0.001. One-way ANOVA with Tukey's post hoc analysis was used in (**a, b, f, k, l, m**).

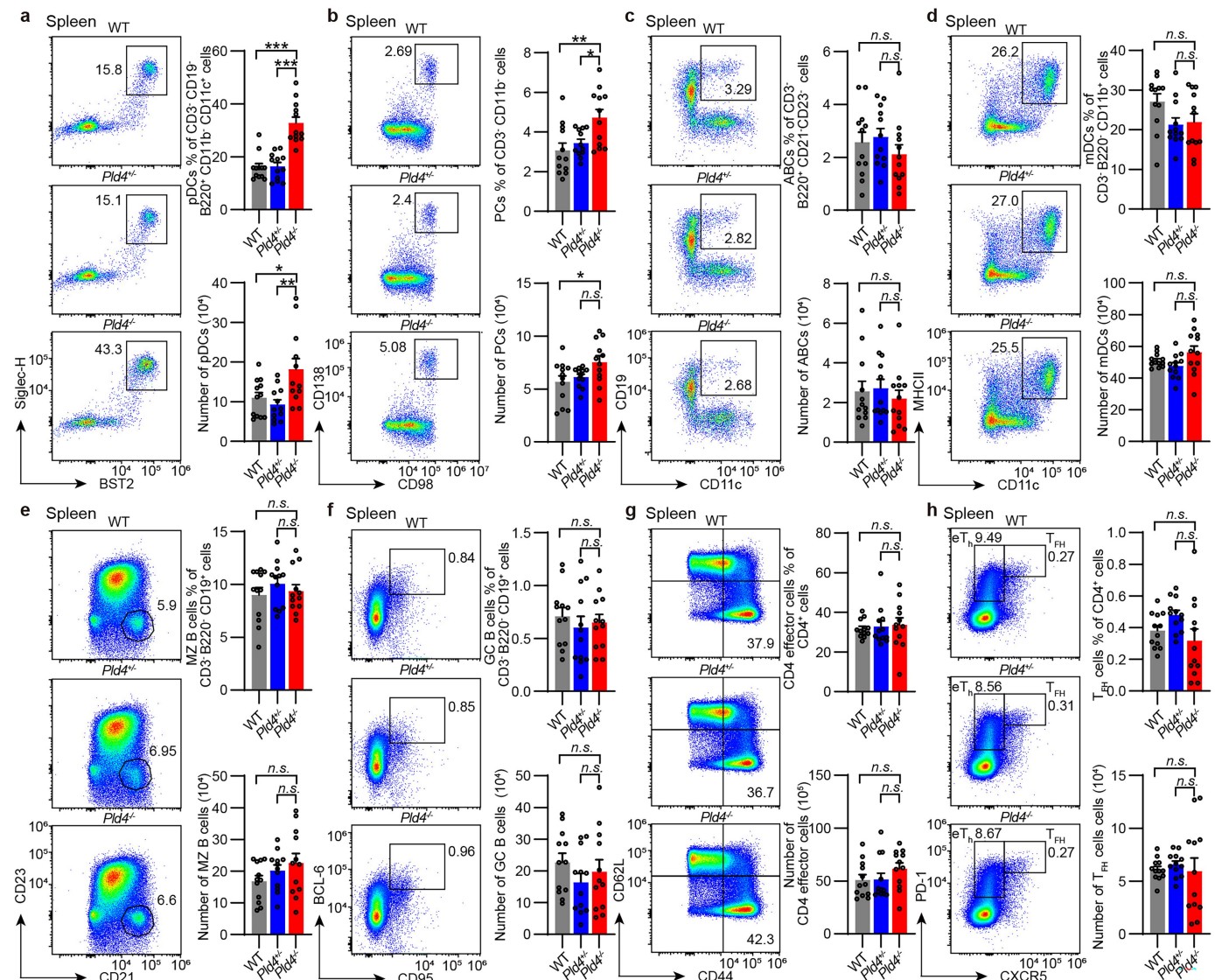

**Extended Data Fig. 8 | Flow cytometric phenotyping of splenic cells from WT and Pld4-deficient mice. a-h,** Representative the spleen flow cytometry plots and quantification of WT and Pld4-deficient mice: pDCs (CD45$^+$CD3$^-$CD19$^-$CD11b$^-$B220$^+$CD11c$^+$Siglec-H$^+$BST2$^+$) in (**a**), PCs (CD45$^+$CD3$^-$CD11b$^-$B220$^{low/-}$CD138$^+$CD98$^+$) in (**b**), ABCs (CD45$^+$CD3$^-$CD21$^-$CD23$^-$B220$^+$CD19$^{high}$CD11c$^+$) in (**c**), mDCs (CD45$^+$CD3$^-$B220$^-$CD11b$^+$CD11c$^+$MHC-II$^+$) in (**d**), MZ B cells (CD45$^+$CD3$^-$ CD19$^+$B220$^+$CD23$^-$CD21$^+$) in (**e**), GC B cells(CD45$^+$CD3$^-$CD19$^+$B220$^+$CD95$^+$BCL-6$^+$) in (**f**), CD4 Effector T cells (CD45$^+$CD3$^+$CD4$^+$CD62L$^-$CD44$^+$) in (**g**), and T$_{FH}$ (CD45$^+$CD3$^+$CD4$^+$PD-1$^+$CXCR5$^+$) in (**h**). WT, n = 12; Pld4$^{+/-}$, n = 12; Pld4$^{-/-}$, n = 12. Data are mean ± s.e.m. These results show a representative of three independent experiments in (**a-h**). *n.s.*, no significant difference; *$P$ < 0.05; **$P$ < 0.01; ***$P$ < 0.001. One-way ANOVA with Tukey's post hoc analysis was used in (**a-h**).

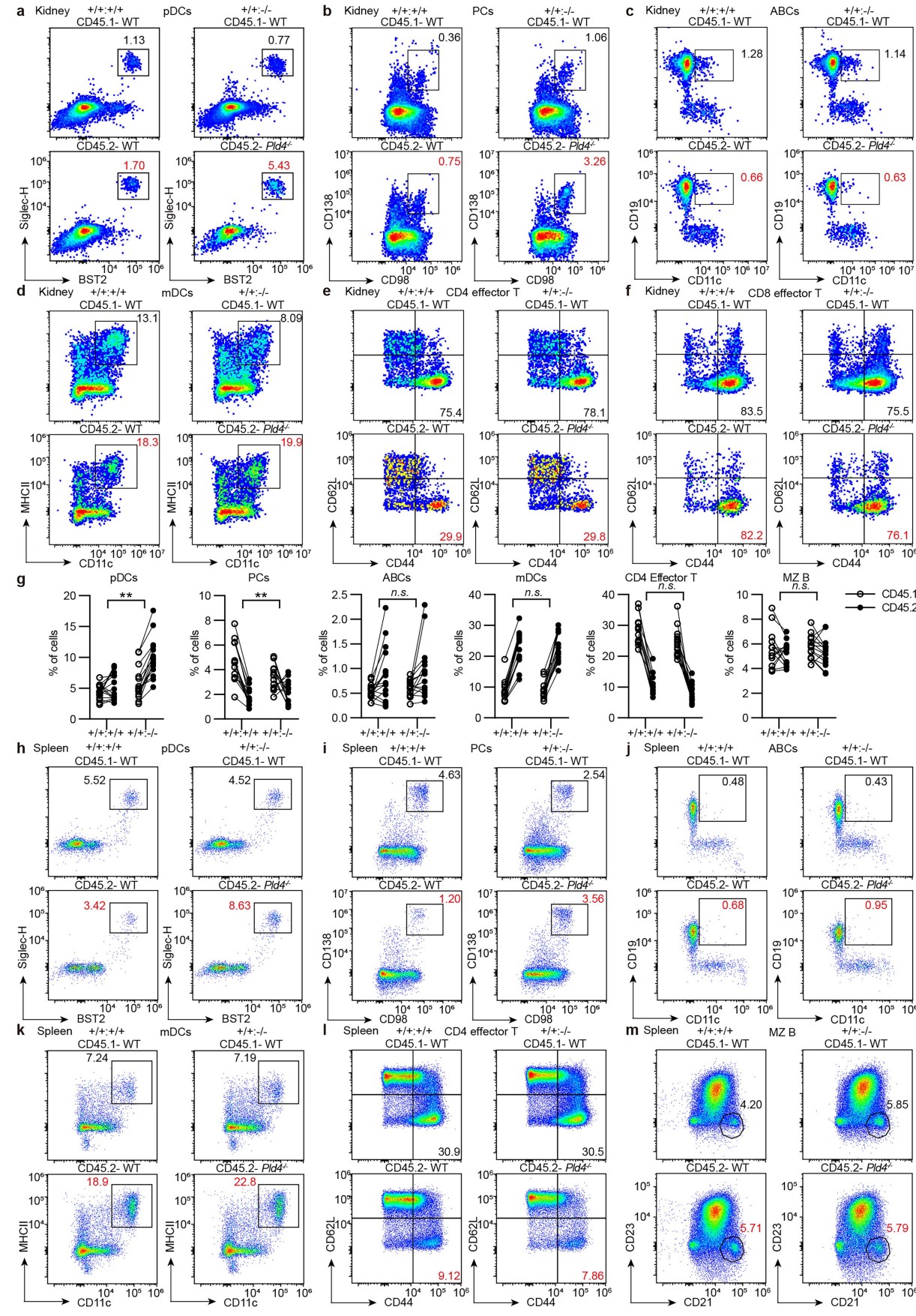

**Extended Data Fig. 9** | See next page for caption.

**Extended Data Fig. 9 | Flow cytometric phenotyping of immune cells in mixed bone marrow chimeric mice. a-f**, Representative the kidney flow cytometry plot and quantification of pDCs (CD45$^+$CD3$^-$CD19$^-$CD11b$^-$B220$^+$CD11c$^+$Siglec-H$^+$BST2$^+$, **a**), PCs (CD45$^+$CD3$^-$CD11b$^-$B220$^{low/-}$CD138$^+$CD98$^+$, **b**), ABCs (CD45$^+$CD3$^-$CD21$^-$CD23$^-$B220$^+$CD19$^{high}$CD11c$^+$, **c**), mDCs (CD45$^+$CD3$^-$B220$^-$CD11b$^+$CD11c$^+$MHC-II$^+$, **d**), CD4 Effector T cells (CD45$^+$CD3$^+$CD4$^+$CD62L$^-$CD44$^+$, **e**), CD8 Effector T cells (CD45$^+$CD3$^+$CD8$^+$CD62L$^-$CD44$^+$, **f**), in mixed bone marrow chimeric mice reconstituted with 1:1 ratio of CD45.1-WT: CD45.2-WT (+/+:+/+) or CD45.1-WT: CD45.2-*Pld4*$^{-/-}$ (+/+:−/−) bone marrow. **g**, Flow cytometry detected the immune cell phenotyping in the spleen of chimeric mice after 8 weeks of transplantation. **h-m**, Representative the spleen flow cytometry plot and quantification of pDCs (CD45$^+$CD3$^-$CD19$^-$CD11b$^-$B220$^+$CD11c$^+$Siglec-H$^+$BST2$^+$, **h**), PCs (CD45$^+$CD3$^-$CD11b$^-$B220$^{low/-}$CD138$^+$CD98$^+$, **i**), ABCs (CD45$^+$CD3$^-$CD21$^-$CD23$^-$B220$^+$CD19$^{high}$CD11c$^+$, **j**), mDCs (CD45$^+$CD3$^-$B220$^-$CD11b$^+$CD11c$^+$MHC-II$^+$, **k**), CD4 Effector T cells (CD45$^+$CD3$^+$CD4$^+$CD62L$^-$CD44$^+$, **l**), MZ B cells (CD45$^+$CD3$^-$CD19$^+$B220$^+$CD23$^-$CD21$^+$, **m**), in mixed bone marrow chimeric mice reconstituted with 1:1 ratio of CD45.1-WT: CD45.2-WT (+/+:+/+) or CD45.1-WT: CD45.2-*Pld4*$^{-/-}$ (+/+:−/−) bone marrow. +/+:+/+, n = 15; +/+:−/−, n = 15. These results show a summary of two independent experiments in (**g**) and a representative of two independent experiments in (**a-f, h-m**).

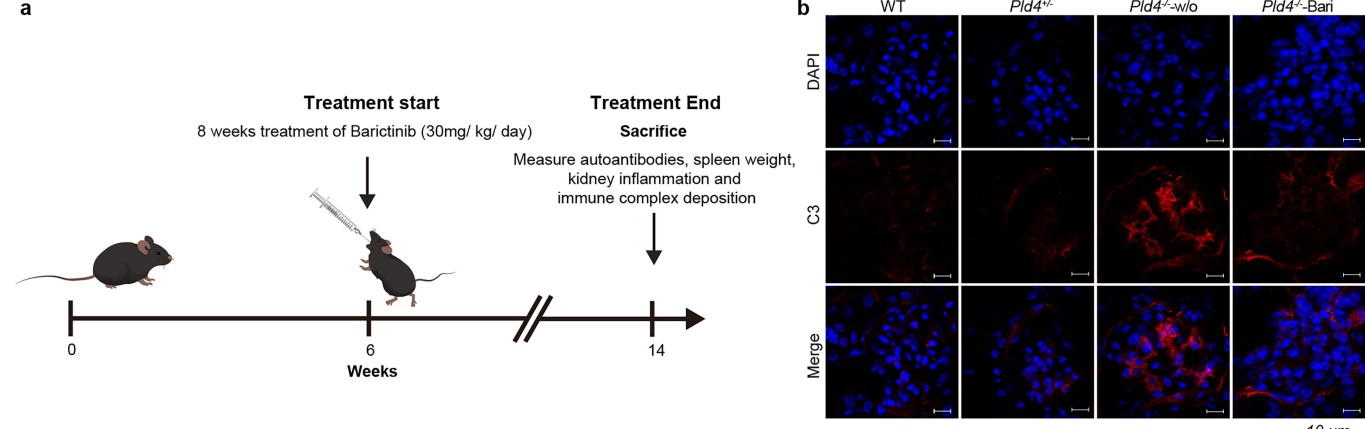

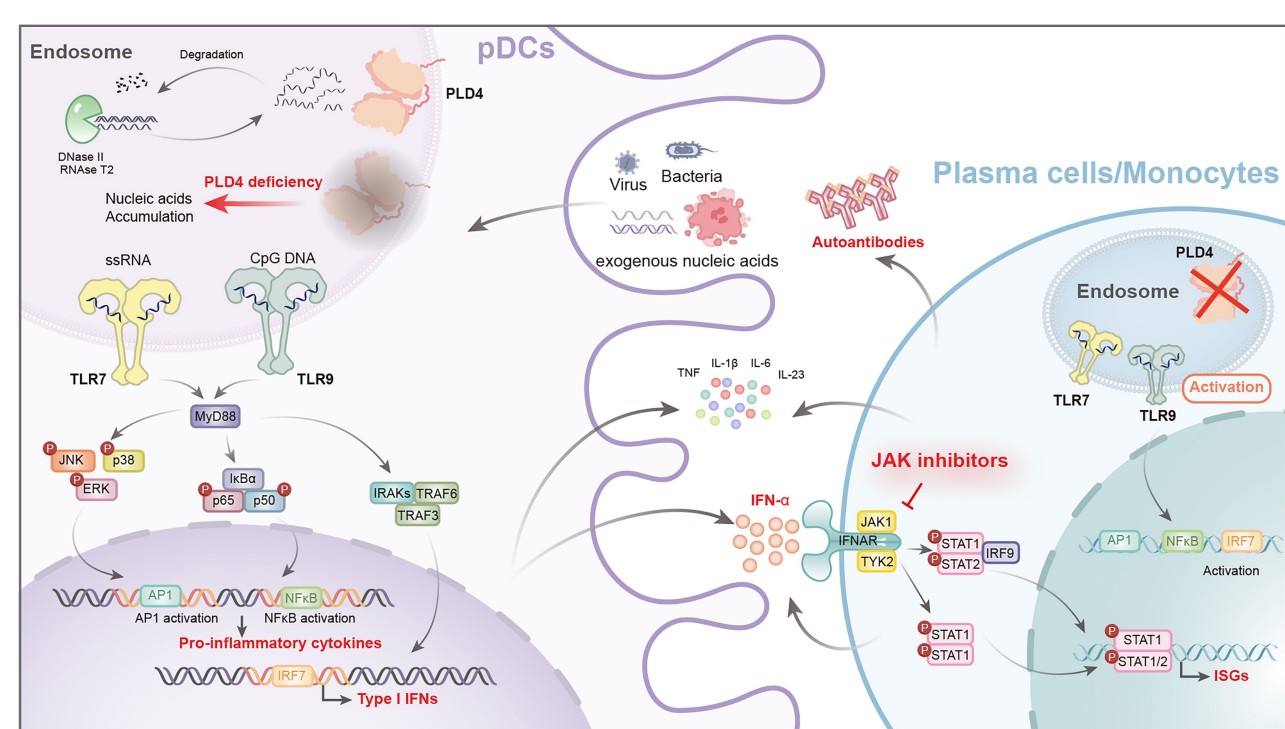

**Extended Data Fig. 10 | Baricitinib treatment strategy and schematic model of PLD4 deficiency mutations causing SLE. a**, Baricitinib treatment strategy and sampling time of Pld4-deficient mice. **b**, C3 staining of kidney glomeruli in Pld4-deficient mice after Baricitinib treatment. Scale bar: 10 μm. **c**, Plasmacytoid dendritic cells (pDCs) sense exogenous or endogenous nucleic acids, moderately activating TLR7/9 and downstream signalling pathways dominated by type I interferon. This results in the production of IFN-α and other pro-inflammatory cytokines like IL-1β, IL-6 and TNF, which in turn activate monocytes and B cells to produce pro-inflammatory cytokines and autoantibodies, facilitating the clearance of apoptotic cells or external pathogens to maintain homeostasis. In the absence of PLD4, pDCs stimulated by exogenous or endogenous nucleic acids fail to degrade these nucleic acids, leading to persistent activation of TLR7/9 and downstream inflammatory signalling pathways. Moreover, the absence of PLD4 in monocytes and B cells results in sustained activation of downstream inflammatory signalling pathways, including type I interferon, exacerbating the release of pro-inflammatory cytokines and the production and deposition of autoantibodies. Given the significant upregulation of the type I interferon signalling pathway following PLD4 deficiency, the JAK inhibitor Baricitinib can inhibit the inflammatory pathway activation caused by PLD4 deficiency, thereby rescuing the autoimmune phenotypes in mice. These results show a representative of two independent experiments in (**b**).

# Reporting Summary

## Statistics

For all statistical analyses, confirm that the following items are present in the figure legend, table legend, main text, or Methods section.

| n/a | Confirmed | |
|---|---|---|
| ☐ | ☒ | The exact sample size (*n*) for each experimental group/condition, given as a discrete number and unit of measurement |
| ☐ | ☒ | A statement on whether measurements were taken from distinct samples or whether the same sample was measured repeatedly |
| ☐ | ☒ | The statistical test(s) used AND whether they are one- or two-sided<br>*Only common tests should be described solely by name; describe more complex techniques in the Methods section.* |
| ☒ | ☐ | A description of all covariates tested |
| ☐ | ☒ | A description of any assumptions or corrections, such as tests of normality and adjustment for multiple comparisons |
| ☐ | ☒ | A full description of the statistical parameters including central tendency (e.g. means) or other basic estimates (e.g. regression coefficient) AND variation (e.g. standard deviation) or associated estimates of uncertainty (e.g. confidence intervals) |
| ☐ | ☒ | For null hypothesis testing, the test statistic (e.g. *F*, *t*, *r*) with confidence intervals, effect sizes, degrees of freedom and *P* value noted<br>*Give P values as exact values whenever suitable.* |
| ☒ | ☐ | For Bayesian analysis, information on the choice of priors and Markov chain Monte Carlo settings |
| ☒ | ☐ | For hierarchical and complex designs, identification of the appropriate level for tests and full reporting of outcomes |
| ☒ | ☐ | Estimates of effect sizes (e.g. Cohen's *d*, Pearson's *r*), indicating how they were calculated |

*Our web collection on statistics for biologists contains articles on many of the points above.*

## Software and code

Policy information about availability of computer code

| Data collection | Flow cytometry and intracellular cytokine staining data were acquired on CytoFLEX Flow Cytometer (Beckman Coulter) and Aurora CS Cell Sorter (Cytek Biosciences). qPCR data were acquired on ROCHE 480II.Immunoblot and exonuclease activity images were scanned by FluorChem E (ProteinSimple). Chemiluminiscence data were collected by the multimode plate reader (Bio Tek). Immunofluorescence images were taken by LSM 710 Confocal Microscope (Zeiss) and LSM 800 Confocal Microscope (Zeiss). RNA sequencing and Whole exome sequencing were carried out on an Illumina Novaseq system. Single cell RNA sequencing were carried out on 10x Genomics Chromium machine for single-cell capture and cDNA preparation and on an Illumina Novaseq system for sequencing. |
|---|---|
| Data analysis | FlowJo V10.8.1(Flow cytometry and intracellular cytokine staining analysis); GraphPad Prism 8 and Microsoft Excel (Graphs, statistics); Image J (image analysis); Cell Ranger V3.0.1 and Seurat R package (single cell RNA sequencing analysis); cytofkit_0.99.0 (CyTOF data analysis); R 3.5.2 (data analysis and graph); DESeq2 R package (RNA sequencing analysis); ANNOVAR (mutation effect predictions). |

For manuscripts utilizing custom algorithms or software that are central to the research but not yet described in published literature, software must be made available to editors and reviewers. We strongly encourage code deposition in a community repository (e.g. GitHub). See the Nature Portfolio guidelines for submitting code & software for further information.

## Data

All manuscripts must include a data availability statement. This statement should provide the following information, where applicable:

- Accession codes, unique identifiers, or web links for publicly available datasets
- A description of any restrictions on data availability
- For clinical datasets or third party data, please ensure that the statement adheres to our policy

The raw RNA-sequence data reported in this paper have been deposited in the Genome Sequence Archive in National Genomics Data Center, China National Center for Bioinformation / Beijing Institute of Genomics, Chinese Academy of Sciences (GSA-Human: HRA012523) at https://ngdc.cncb.ac.cn/gsa-human. Gel source data are shown in Supplementary Fig. 1. All data are available in the main text or the Supplementary Appendix from the corresponding authors upon reasonable request.

## Research involving human participants, their data, or biological material

Policy information about studies with human participants or human data. See also policy information about sex, gender (identity/presentation), and sexual orientation and race, ethnicity and racism.

| | |
|---|---|
| Reporting on sex and gender | In this study, both sex and gender were considered. Sex was determined based on self-reporting by participants during data collection. This research involved two biological males and three biological females, with detailed information provided in Supplementary Table. 1. The human samples for this study were blood samples to isolate peripheral blood mononuclear cells (PBMCs). |
| Reporting on race, ethnicity, or other socially relevant groupings | All participants in this study were of Asian descent, specifically from China. No other racial or ethnic groups were included, and therefore, no comparison across different racial or ethnic categories was performed. |
| Population characteristics | The patients characteristics have been described in Supplementary Table 1. |
| Recruitment | Patient samples were obtained from patients with SLE but without clear genetic diagnosis. And multiple age and gender matched healthy controls used in the experiment were randomly selected. |
| Ethics oversight | All patients enrolled in the study were evaluated under a protocol approved by the Institutional Review Boards evaluated at Jinling Hospital(2022DZKY-061-01). |

Note that full information on the approval of the study protocol must also be provided in the manuscript.

# Field-specific reporting

Please select the one below that is the best fit for your research. If you are not sure, read the appropriate sections before making your selection.

☒ Life sciences ☐ Behavioural & social sciences ☐ Ecological, evolutionary & environmental sciences

For a reference copy of the document with all sections, see nature.com/documents/nr-reporting-summary-flat.pdf

# Life sciences study design

All studies must disclose on these points even when the disclosure is negative.

| | |
|---|---|
| Sample size | No statistical method was used to calculate sample size. Sample sizes with mice were determined by the availability of animals with the correct genotypes or based on numbers used in previous publications where comparable sample sizes produced statistically significant results. |
| Data exclusions | No data was excluded from analysis. |
| Replication | RNA-Seq in Fig.2a-b, Fig.3d, ED Fig. 2a-b was performed using 2 biological replicates in each sample.<br>Single cell RNA sequencing analysis of patient sample and healthy controls in Fig.2d-f, ED Fig. 3 and ED Fig. 4a,b,e were performed once.<br>Single cell RNA sequencing analysis of mice kidney sample in Fig.4c, and ED Fig. 7g-j were performed once with n=2.<br>CyTOF analysis of patient sample and healthy controls in Fig.2g and ED Fig. 5g were performed once.<br>Fig.2c were performed twice.<br>Fig.3 a, b, f, g, h show a representative of at least three independent experiments, Fig.3 c show a representative of two independent experiments and Fig.3 e shows a summary of two independent experiments.<br>Fig.4 b, d, e, f, g show a representative of at least three independent experiments, Fig.4 a show a representative of two independent experiments and Fig.4 h,i shows a summary of two independent experiments.<br>Fig.5 a, b, c, d, e, f show a representative of two independent experiments.<br>ED Fig. 2 c, d, e were performed three times.<br>ED Fig. 4 d was performed twice.<br>ED Fig. 5 b, c, d, e, f were performed twice.<br>ED Fig. 6 a, b, h, i, k, l were performed three times, ED Fig. 6 d, e were performed twice and ED Fig. 6 c, f, g were performed once. |

ED Fig. 7 c, d, e, f, k, l, m were performed at least three times, ED Fig. 7 b was performed twice, and ED Fig. 7 a was performed once with n>8.
ED Fig. 8 a-h were performed three times.
ED Fig. 9 a-m were performed twice.
ED Fig. 10 b was performed twice.

Randomization | Multiple age and gender matched healthy controls used in the experiment were randomly selected. Patient samples were taken multiple times and used for independent biological replicate experiments.
For animals experiment, they were age and sex matched with each other in each experiment (littermates).

Blinding | Genotypes were blinded in the kidney histology pathology analysis. Other data collection and analyses were not performed blindly to the conditions of the experiment.

# Reporting for specific materials, systems and methods

We require information from authors about some types of materials, experimental systems and methods used in many studies. Here, indicate whether each material, system or method listed is relevant to your study. If you are not sure if a list item applies to your research, read the appropriate section before selecting a response.

## Materials & experimental systems

| n/a | Involved in the study |
|---|---|
| ☐ | ☒ Antibodies |
| ☐ | ☒ Eukaryotic cell lines |
| ☒ | ☐ Palaeontology and archaeology |
| ☐ | ☒ Animals and other organisms |
| ☒ | ☐ Clinical data |
| ☒ | ☐ Dual use research of concern |
| ☒ | ☐ Plants |

## Methods

| n/a | Involved in the study |
|---|---|
| ☒ | ☐ ChIP-seq |
| ☐ | ☒ Flow cytometry |
| ☒ | ☐ MRI-based neuroimaging |

## Antibodies

Antibodies used | 1.Western Blot:
NF-κB p65 (Cell signaling technology, 8242,1:1000 ), Phospho-NF-kappaB p65 (Cell signaling technology, 3033,1:1000 ), p44/42 MAPK (Cell signaling technology, 4696,1:1000 ), Phospho-p44/42 MAPK (Cell signaling technology, 4370,1:1000 ), STAT1 (Cell signaling technology, 14994, 1:1000 ), Phospho-STAT1 (Cell signaling technology, 9167,1:1000 ), β-actin(Cell signaling technology, 4970,1:1000 ), GAPDH (Cell signaling technology, 2118,1:1000 ), Phospho-Stat2 (Tyr690) (Cell signaling technology, 88410,1:1000),PLD4 (Thermo Fisher Scientific, PA5-98680),Stat2(Cell signaling technology, 72604,1:1000),Toll-like Receptor 7 (Cell signaling technology, 5632,1:1000 ), Phospho-STING (Ser366) (Cell signaling technology, 50907), STING  (Cell signaling technology, 13647),HRP Conjugated Anti-DYKDDDDK Tag(FLAG) (HUABIO, 0912-3,1:5000 )

2.Flow cytometry, intracellular cytokine staining and Immunofluorescence:
CD3-APC-H7 (BD Biosciences, 560176), CD4-FITC (BD Biosciences, 555346), CD8-APC (BD Biosciences, 561952), CD14-PE-CY7 (BD Biosciences, 557742), CD19-BB700 (BD Biosciences, 566396), PE Mouse anti-Human IFN-α (BD Biosciences, 560097), BD Pharmingen™ p38 MAPK (pT180/pY182) PE (BD Biosciences, 612565), BD Phosflow™ BV421 Anti-Human NF-κB p65 (pS529) (BD Biosciences, 565446), Alexa Fluor 647 anti-ERK1/2 Phospho (Biolegend, 369504),  Anti-C3 antibody (Abcam, ab11862), FITC anti-human CD21 (Biolegend, 354910), FITC anti-human CD24 (Biolegend, 311104), PerCP/Cyanine5.5 anti-mouse/rat/human CD27 (Biolegend, 124214), Brilliant Violet 605™ anti-human CD38 (Biolegend, 303532), Brilliant Violet 421™ anti-human IgG Fc (Biolegend, 410704), APC anti-human HLA-DR (Biolegend, 307610), Brilliant Violet 570™ anti-mouse CD11c (Biolegend, 117331) V500 Mouse anti-Mouse CD45.2(104) (BD Biosciences, 562129), Brilliant Violet 650 anti-mouse CD45.1 (Biolegend,110736), BV421 Hamster Anti-Mouse CD3e (BD Biosciences, 562600), FITC Rat Anti-Mouse CD4 (BD Biosciences, 553046), APC-Cy7 Rat Anti-Mouse CD8a (BD Biosciences, 557654), APC Rat Anti-Mouse CD19 (BD Biosciences, 550992), PE Rat Anti-Mouse CD138 (BD Biosciences, 553714), BV605 CD317 (BD Biosciences, 747606), PE-Cy7 Rat Anti-Mouse CD45R/B220 (BD Biosciences, 552772), BUV496 CD11b (BD Biosciences, 749864), BV750 F4/80 (BD Biosciences, 747295), RB780 Ly-6C (BD Biosciences, 755871), BUV395 I-A, I-E (BD Biosciences, 569244), BUV805 CD44 (BD Biosciences, 741921), BUV563 CD62L (BD Biosciences, 741230), BV786 Rat Anti-Mouse CD25 (BD Biosciences, 564023), BV650 Rat Anti-Mouse IL-17A (BD Biosciences, 564170), BUV737 Rat Anti-Mouse CD21/CD35 (BD Biosciences, 612810), RB545 CD23 (BD Biosciences, 756344), BV480 CD95 (BD Biosciences, 746755), BUV615 CD49b (BD Biosciences, 751052), BV711 CD279 (BD Biosciences, 744547), PE-CF594 Rat Anti-Mouse CD185 (CXCR5) (BD Biosciences, 562856), R718 Ly-6G (BD Biosciences, 567039), PE-CYN5 FOXP3 (Invitrogen, 15-5773-82), and PERCPEF710 BCL-6 (Invitrogen, 46-5453-82),RB744 Rat Anti-Mouse Siglec-H (BD Biosciences, 757466) and BUV661 Rat Anti-Mouse CD98 (BD Biosciences, 752893).

Validation | All antibodies have been verified by the manufacturers
1. Western Blot:
NF-κB p65:  https://www.cellsignal.com/products/primary-antibodies/nf-kb-p65-d14e12-xp-rabbit-mab/8242
Phospho-NF-kappaB p65:  https://www.cellsignal.com/products/primary-antibodies/phospho-nf-kb-p65-ser536-93h1-rabbit-mab/3033
p44/42 MAPK: https://www.cellsignal.com/products/primary-antibodies/p44-42-mapk-erk1-2-l34f12-mouse-mab/4696
Phospho-p44/42 MAPK: https://www.cellsignal.com/products/primary-antibodies/phospho-p44-42-mapk-erk1-2-thr202-tyr204-d13-14-4e-xp-rabbit-mab/4370
STAT1:  https://www.cellsignal.com/products/primary-antibodies/stat1-d1k9y-rabbit-mab/14994

Phospho-STAT1: https://www.cellsignal.com/products/primary-antibodies/phospho-stat1-tyr701-58d6-rabbit-mab/9167
β-actin:  https://www.cellsignal.com/products/primary-antibodies/b-actin-13e5-rabbit-mab/4970
GAPDH: https://www.cellsignal.com/products/primary-antibodies/gapdh-14c10-rabbit-mab/2118
PLD4 : https://www.thermofisher.com/antibody/product/PLD4-Antibody-Polyclonal/PA5-98680
Stat2:https://www.cellsignal.cn/products/primary-antibodies/stat2-d9j7l-rabbit-mab/72604
Phospho-Stat2 :https://www.cellsignal.cn/products/primary-antibodies/phospho-stat2-tyr690-d3p2p-rabbit-mab/88410
Toll-like Receptor 7: https://www.cellsignal.com/products/primary-antibodies/toll-like-receptor-7-d7-rabbit-mab/5632
Phospho-STING : https://www.cellsignal.cn/products/primary-antibodies/phospho-sting-ser366-e9a9k-rabbit-mab/50907
STING: https://www.cellsignal.cn/products/primary-antibodies/sting-d2p2f-rabbit-mab/13647
HRP Conjugated Anti-DYKDDDDK Tag(FLAG) : https://www.huabio.com/products/dykddddk-tag-flag-hrp-antibody-polyclonal-0912-3

2.Flow cytometry, intracellular cytokine staining and Immunofluorescence:
CD3-APC-H7 : https://www.bdbiosciences.com/en-nz/products/reagents/flow-cytometry-reagents/research-reagents/single-color-antibodies-ruo/apc-h7-mouse-anti-human-cd3.560176
CD4-FITC :  https://www.bdbiosciences.com/en-nz/products/reagents/flow-cytometry-reagents/research-reagents/single-color-antibodies-ruo/fitc-mouse-anti-human-cd4.555346
CD8-APC : https://www.bdbiosciences.com/en-us/products/reagents/flow-cytometry-reagents/research-reagents/single-color-antibodies-ruo/apc-mouse-anti-human-cd8.561952
CD14-PE-CY7: https://www.bdbiosciences.com/en-us/products/reagents/flow-cytometry-reagents/research-reagents/single-color-antibodies-ruo/pe-cy-7-mouse-anti-human-cd14.557742
CD19-BB700: https://www.bdbiosciences.com/en-us/products/reagents/flow-cytometry-reagents/research-reagents/single-color-antibodies-ruo/bb700-mouse-anti-human-cd19.566396
PE Mouse anti-Human IFN-α: https://www.bdbiosciences.com/en-us/products/reagents/flow-cytometry-reagents/research-reagents/single-color-antibodies-ruo/pe-mouse-anti-human-ifn-2b.560097
p38 MAPK (pT180/pY182) PE: https://www.bdbiosciences.com/en-fi/products/reagents/flow-cytometry-reagents/research-reagents/single-color-antibodies-ruo/pe-mouse-anti-p38-mapk-pt180-py182.612565
BV421 Anti-Human NF-κB p65 (pS529): https://www.bdbiosciences.com/en-eu/products/reagents/flow-cytometry-reagents/research-reagents/single-color-antibodies-ruo/bv421-mouse-anti-human-nf-b-p65-ps529.565446
Alexa Fluor 647 anti-ERK1/2 Phospho:https://www.biolegend.com/ja-jp/products/alexa-fluor-647-anti-erk1-2-phospho-thr202-tyr204-antibody-12924
Anti-C3 antibody: https://www.abcam.com/en-us/products/primary-antibodies/c3-antibody-11h9-ab11862
FITC anti-human CD21: https://www.biolegend.com/fr-lu/products/fitc-anti-human-cd21-antibody-8189
FITC anti-human CD24: https://www.biolegend.com/fr-ch/products/fitc-anti-human-cd24-antibody-1804?GroupID=BLG5916
PerCP/Cyanine5.5 anti-mouse/rat/human CD27: https://www.biolegend.com/nl-be/products/percp-cyanine55-anti-mouse-rat-human-cd27-antibody-4396
Brilliant Violet 605™ anti-human CD38: https://www.biolegend.com/nl-be/products/brilliant-violet-605-anti-human-cd38-antibody-8154
Brilliant Violet 421™ anti-human IgG Fc: https://www.biolegend.com/en-ie/products/brilliant-violet-421-anti-human-igg-fc-11784
APC anti-human HLA-DR: https://www.biolegend.com/fr-fr/products/apc-anti-human-hla-dr-antibody-787
Brilliant Violet 570™ anti-mouse CD11c: https://www.biolegend.com/de-de/products/brilliant-violet-570-anti-mouse-cd11c-antibody-7396
Brilliant Violet 650 anti-mouse CD45.1: https://www.biolegend.com/de-de/products/brilliant-violet-650-anti-mouse-cd45-1-antibody-7644
BV421 Hamster Anti-Mouse CD3e: https://www.bdbiosciences.com/en-ca/products/reagents/flow-cytometry-reagents/research-reagents/single-color-antibodies-ruo/bv421-hamster-anti-mouse-cd3e.562600?tab=product_details
FITC Rat Anti-Mouse CD4: https://www.bdbiosciences.com/en-us/products/reagents/flow-cytometry-reagents/research-reagents/single-color-antibodies-ruo/fitc-rat-anti-mouse-cd4.553046?tab=product_details
APC-Cy7 Rat Anti-Mouse CD8a: https://www.bdbiosciences.com/en-us/products/reagents/flow-cytometry-reagents/research-reagents/single-color-antibodies-ruo/apc-cy-7-rat-anti-mouse-cd8a.557654?tab=product_details
APC Rat Anti-Mouse CD19: https://www.bdbiosciences.com/en-us/products/reagents/flow-cytometry-reagents/research-reagents/single-color-antibodies-ruo/apc-rat-anti-mouse-cd19.550992?tab=product_details
PE Rat Anti-Mouse CD138: https://www.bdbiosciences.com/en-nl/products/reagents/flow-cytometry-reagents/research-reagents/single-color-antibodies-ruo/pe-rat-anti-mouse-cd138.553714?tab=product_details
BV605 CD317 : https://www.bdbiosciences.com/en-ca/products/reagents/flow-cytometry-reagents/research-reagents/single-color-antibodies-ruo/bv605-rat-anti-mouse-cd317-bst2.747606?tab=product_details
PE-Cy7 Rat Anti-Mouse CD45R/B220: https://www.bdbiosciences.com/en-fi/products/reagents/flow-cytometry-reagents/research-reagents/single-color-antibodies-ruo/pe-cy-7-rat-anti-mouse-cd45r-b220.552772?tab=product_details
BUV496 CD11b: https://www.bdbiosciences.com/en-fi/products/reagents/flow-cytometry-reagents/research-reagents/single-color-antibodies-ruo/buv496-rat-anti-cd11b.749864?tab=product_details
BV750 F4/80: https://www.bdbiosciences.com/en-br/products/reagents/flow-cytometry-reagents/research-reagents/single-color-antibodies-ruo/bv750-rat-anti-mouse-f4-80.747295?tab=product_details
RB780 Ly-6C: https://www.bdbiosciences.com/en-us/products/reagents/flow-cytometry-reagents/research-reagents/single-color-antibodies-ruo/rb780-rat-anti-mouse-ly-6c.755871?tab=product_details
BUV395 I-A, I-E: https://www.bdbiosciences.com/en-us/products/reagents/flow-cytometry-reagents/research-reagents/single-color-antibodies-ruo/buv395-rat-anti-mouse-i-a-i-e.569244?tab=product_details
BUV805 CD44: https://www.bdbiosciences.com/en-us/products/reagents/flow-cytometry-reagents/research-reagents/single-color-antibodies-ruo/buv805-rat-anti-mouse-cd44.741921?tab=product_details
BUV563 CD62L: https://www.bdbiosciences.com/en-us/products/reagents/flow-cytometry-reagents/research-reagents/single-color-antibodies-ruo/buv563-rat-anti-mouse-cd62l.741230?tab=product_details
 BV786 Rat Anti-Mouse CD25: https://www.bdbiosciences.com/en-us/products/reagents/flow-cytometry-reagents/research-reagents/single-color-antibodies-ruo/bv786-rat-anti-mouse-cd25.564023?tab=product_details
BV650 Rat Anti-Mouse IL-17A: https://www.bdbiosciences.com/en-us/products/reagents/flow-cytometry-reagents/research-reagents/single-color-antibodies-ruo/bv650-rat-anti-mouse-il-17a.564170?tab=product_details
BUV737 Rat Anti-Mouse CD21/CD35: https://www.bdbiosciences.com/en-be/products/reagents/flow-cytometry-reagents/research-reagents/single-color-antibodies-ruo/buv737-rat-anti-mouse-cd21-cd35.612810?tab=product_details
RB545 CD23 : https://www.bdbiosciences.com/en-ca/products/reagents/flow-cytometry-reagents/research-reagents/single-color-

antibodies-ruo/RB545-Rat-Anti-Mouse-CD23.756344?tab=product_details
BV480 CD95: https://www.bdbiosciences.com/en-us/products/reagents/flow-cytometry-reagents/research-reagents/single-color-antibodies-ruo/bv480-hamster-anti-mouse-cd95-fas.746755?tab=product_details
BUV615 CD49b: https://www.bdbiosciences.com/en-us/products/reagents/flow-cytometry-reagents/research-reagents/single-color-antibodies-ruo/buv615-hamster-anti-mouse-cd49b.751052?tab=product_details
BV711 CD279: https://www.bdbiosciences.com/en-us/products/reagents/flow-cytometry-reagents/research-reagents/single-color-antibodies-ruo/BV711-Hamster-Anti-Mouse-CD279-(PD-1).744547?tab=product_details
PE-CF594 Rat Anti-Mouse CD185 (CXCR5) : https://www.bdbiosciences.com/en-pt/products/reagents/flow-cytometry-reagents/research-reagents/single-color-antibodies-ruo/pe-cf594-rat-anti-mouse-cd185-cxcr5.562856?tab=product_details
R718 Ly-6G: https://www.bdbiosciences.com/en-ca/products/reagents/flow-cytometry-reagents/research-reagents/single-color-antibodies-ruo/r718-rat-anti-mouse-ly-6g.567039?tab=product_details
PERCPEF710 BCL-6:https://www.thermofisher.com/antibody/product/BCL6-Antibody-clone-BCL-DWN-Monoclonal/46-5453-82
RB744 Rat Anti-Mouse Siglec-H:https://www.bdbiosciences.com/en-us/products/reagents/flow-cytometry-reagents/research-reagents/single-color-antibodies-ruo/rb744-rat-anti-mouse-siglec-h.757466?tab=product_details
BUV661 Rat Anti-Mouse CD98:https://www.bdbiosciences.com/en-no/products/reagents/flow-cytometry-reagents/research-reagents/single-color-antibodies-ruo/BUV661-Rat-Anti-Mouse-CD98.752893?tab=product_details

# Eukaryotic cell lines

Policy information about cell lines and Sex and Gender in Research

| Cell line source(s) | HEK293T and THP-1 cell line were from the American Type Culture Collection. |
| Authentication | Cell line from ATCC has been authenticated by ATCC. |
| Mycoplasma contamination | The cell lines have been tested to be negative for mycoplasma contamination. |
| Commonly misidentified lines (See ICLAC register) | No commonly misidentified cell lines were used. |

# Animals and other research organisms

Policy information about studies involving animals; ARRIVE guidelines recommended for reporting animal research, and Sex and Gender in Research

| Laboratory animals | Pld4 KO mice, on a C57BL/6 background, were purchased from the Shanghai Model Organisms used at 6 weeks of age. To generate mixed bone marrow chimeras, bone marrow cells from WT mice and Pld4-/- mice were flushed from mouse tibias and femurs and transplanted to lethally irradiated (7 Gly) B6-CD45.1(Ptprc-p.K302E) mice. |
| Wild animals | The study did not involve any wild animals. |
| Reporting on sex | In this study, we used both male and female mice. No significant differences in phenotype were observed between the two sexes. |
| Field-collected samples | The study did not involve samples collected from the field. |
| Ethics oversight | All mouse experiments were performed according to the guidelines of the Animal Care and Use Committees at the Zhejiang University School of Medicine. |

Note that full information on the approval of the study protocol must also be provided in the manuscript.

# Plants

| Seed stocks | N/A |
| Novel plant genotypes | N/A |
| Authentication | N/A |

# Flow Cytometry

## Plots

Confirm that:

☒ The axis labels state the marker and fluorochrome used (e.g. CD4-FITC).

☒ The axis scales are clearly visible. Include numbers along axes only for bottom left plot of group (a 'group' is an analysis of identical markers).

☒ All plots are contour plots with outliers or pseudocolor plots.

☒ A numerical value for number of cells or percentage (with statistics) is provided.

## Methodology

**Sample preparation**

We used EDTA-anticoagulated peripheral whole blood from patients and health controls. PBMCs from patients and healthy donors were separated by lymphocyte separation medium (LSM) according to the manufacturer's instructions.

Single-cell suspensions from mouse spleens were prepared by gently triturating the tissue with the plunger end of a syringe and filtering the resultant mixture through 40 μm filters with 2% FBS in PBS (2% FBS-PBS). The filtration was then centrifuged at 350 g for 5 minutes. The cell pellet was resuspended in 2 mL of RBC lysis buffer and incubated at 25 °C for 3 minutes. Following this, the suspension was diluted with 10 mL 2% FBS-PBS. After a second centrifugation at 350 g for 5 minutes, the spleen cells were resuspended in PBS.

For mouse kidney single-cell suspensions preparation, half of the kidney was minced into 1-2 mm³ pieces using surgical curved scissors in a dish and transferred to a 15 mL centrifuge tube. To this, 2 mL of digestion solution (1 mg/mL collagen IV, 200μg/mL DNase I) was added, and the tissue was incubated at 37 °C for 30 minutes. Digestion was then halted by the addition of 10% FBS-PBS, and filtered the mixture through 40 μm filters. The mixture was subsequently centrifuged, and resuspended in 2 mL of RBC lysis buffer following the protocol of spleen single-cell suspension.

**Instrument**

All events were acquired on CytoFLEX Flow Cytometer (Beckman Coulter) and Aurora CS Cell Sorter (Cytek Biosciences).

**Software**

All events were analyzed with FlowJo V10.8.1.

**Cell population abundance**

1.0E6 of PBMCs or 2.0E6 of mouse spleen or kidney cells were used for each test. Coutess II FL (Thermo Fisher) and 0.4% Typan Blue were used to determine the viability and number of cells. The viability of PBMCs and spleen cells in each sample was above 90%. The viability of kidney cells in each sample was above 80%.

**Gating strategy**

Surface markers CD3, CD4, CD8, CD14 and CD19 were used to gate total T cells, CD4+T cells, CD8+T cells, monocytes and total B cells respectively. Other target populations were further determined by specific antibodies, which were able to distinguish from negative populations.

☒ Tick this box to confirm that a figure exemplifying the gating strategy is provided in the Supplementary Information.

