## [Peer Review File · Nature]

Loss-of-Function Mutations in *PLD4* Lead to Systemic Lupus Erythematosus

Corresponding Author: Professor Zhihong Liu

Version 0:

Reviewer comments:

Referee #1

(Remarks to the Author)

This study by Wang et al., identifies five patients with systemic lupus erythematosus (SLE) carrying compound heterozygous *PLD4* variants that cause loss of exonuclease function. GWAS studies have identified *PLD4* as a lupus susceptibility gene and *PLD4* deficiency in mice has been shown to result in the development of autoimmunity through increased TLR7 signaling, and in a Balb/c background, also increased TLR9 signaling. Wang et al's findings are stringly supportive that biallelic *PLD4* loss-of-function (LoF) variants cause - or significantly contribute to - SLE in humans. Given that all variants are rare, biallelic and LoF in vitro, and that there are studies from the Nemazee lab showing *PLD4* deficiency causes lupus-like disease in mice that involves TLR7 and TLR9, it is reasonable not to require further proof of variant pathogenicity by introduction of the specific variants into mice, nor further proof that TLR7 and TLR9 are involved in disease pathogenesis. The authors provide some evidence that patients' PBMCs exhibit heightened sensitivity to TLR9 ligand (CpG) stimulation, although the fold difference is less marked than unstimulated cells, thus caution is needed when interpreting these data. Overall, these findings strongly support a pathogenic role of *PLD4* deficiency in human SLE.

Main points:

1. The magnitude of differences in TNF and type I IFN-related RNAs are higher in unstimulated cells compared with CpG stimulation, suggesting that the differences may not be due so much to the nature of the stimulus, but simply to their hyperactivated status. For example, there is a 3-4-fold increase in TNF mRNA in unstimulated cells compared with a 1.5-fold difference in cells after CpG stimulation. The authors could state their conclusions with some caution.
2. Given the existing controversies of TLR9 role in SLE with data suggesting that TLR9 also plays protective roles (PMID: 36151396), it would be useful for the reader if the authors mention the different - and for some phenotypes, opposite - results observed in Balb/c (PMID: 37555846) vs C57BL/6 mice that lack *PLD4*. It is only in the Balb/c mouse background where TLR9 clearly contributes to lupus-like autoimmunity, as opposed to myeloid cell driven autoinflammation.
3. The cell subset characterization by scRNAseq in Fig. 4 is not very useful (besides the labels not being clear); measurements of total B and T cells are not very meaningful or informative. Given the recent advances in characterization of cell subsets thought to be pathogenic in SLE (i.e. aged associated B cells, plasmacytoid DCs) and the intriguing contribution of TLR9, it would be useful if the authors characterized and quantified the different B cell and DC subsets show if the changes are cell autonomous (using mixed bone marrow chimeras) or dependent on the inflammatory milieu.
4. The authors show response to baricitinib in B6 mice lacking *PLD4*. In the B6 background TLR9 may not be significantly contributing to disease given Nemazee's results in their 2018 and 2021 papers, in which TLR9 deficiency in fact exacerbates some disease features including thrombocytopenia. Thus, a word of caution is needed: the results seen in B6 mice may not necessarily be extrapolated to patients. This should at least be discussed.

Referee #2

(Remarks to the Author)

This is an interesting paper which expands the list of monogenic lupus cases. It is well documented and clearly presented.

Previous monogenic cases in the same pathways have reported mutations of TLR7 and its downstream molecule UNCB1. The PDL4 mutations which are reported herein involve a molecule which is "upstream" of TLR7/9 and involves the defective digestion of DNA/RNA. The PDLA4 mutations therefore can be classified along with DNASEs defects which also account for defective degradation of nucleic acids.

The novelty of this manuscript is with the presentation of 5 patients with PDL4 mutations all of whom had SLE a couple in young age and a couple at an older age. PDLA4 deficient mice developing autoimmunity had been described before and the use baricitinib has been in lupus clinical trials. One way to increase the novelty of this paper to construct knock in mice with the patient-identified PDL4 mutations. The author's present evidence that TLR9 in this setting behaves like TLR7. The role of TLR9 has not been settled and it looks like that it is setting dependent.

In a number of places the writing can be improved.

In lines 180-200 they refer to figure 4 as figure 3.

Referee #3

(Remarks to the Author)

This manuscript reports on five patients with both early and adult-onset SLE who carry biallelic mutations in PLD4, a putative new cause of monogenic SLE. The

recent discoveries of new genes causal for Mendelian forms of lupus are critically important for understanding the underlying mechanisms of tolerance breakdown in Human.

The work includes experiments in primary cells from patients, mouse models and cell lines, and the findings are globally compelling.

I have some suggestions / questions regarding the current work.

-The genetic data are well detailed with position, in silico scoring, and all together these data are convincing. Additional structural data with the position of the variants may be informative if the variants cluster in a specific region.

-Regarding clinical data, it interesting to observe that all patients presented with a lupus nephritis. Similar to DNASE1L3 deficiency (associated with extracellular nuclease defects), some patients had urticaria. Was the diagnosis consistent with hypocomplementemic urticaria vasculitis (Mc Duffie syndrome)? Were there any signs of brain calcification, paresis, leukodystrophy? Analyzing ANCA systematically would be valuable.

-IgG level is dramatically reduced in patients 1 and 2 and increase in Pt4/5. How were B cell compartments in these patients and globally in KO Mice? (number and phenotype)

-PLD4 is said to be primarily expressed in endosomes in the introduction, while other reports mention it is also expressed in ER / golgi / Cytoplasm. Have the authors analysed the subcellular expression of PDL4? Importantly, the inflammation of Pld4 KO mice was also reported as secondary to intracellular sensing via Sting. Indeed, the IFN-I signature is persistent in Unc93b13d/3dPld3-/-Pld4-/- mice (Gavin et al. 2021). Exploring the cGAS/STING pathway in cell lines would add to the understanding of immune dysregulation of PDL4 deficiency.

-The assays on nuclease activity are particularly convincing on DNA material and to a milder extend to RNA. This could be discussed in the text.

-How many times were these experiments repeated?

-The terms patient "group" is inappropriate if analyzing only one patient (result section, Pt1 Figure 3C).

-The overexpression of TLR7 in PBMCs of Pt1 may also occur in non PLD4 lupus patients.

-Pt1 and Pt2 have been extensively explored and it should always be mentioned in the main and extended Figure legends which patients are involved, avoiding generalizing to all patients

-The Figure 2D is difficult to read due to overlapping colors among cell types. Was the overexpression of TLR9 in scRNASeq experiment confirmed at protein level? Is it observed in lupus patients without mutation?

-In the extended table 1, Hypertension for Pt 3 is mentioned among the immunologic disorders. It would fit better to kidney disorders.

Version 1:

Reviewer comments:

Referee #1

(Remarks to the Author)

The authors have addressed the concerns regarding text modifications/edits for accuracy and completeness.

I however have concerns about the newly added data that are either not robust or unexpected. The issues may be technical but for conclusions to be drawn, the authors need better gating strategies (including additional markers), increased mouse numbers to reach statistical significance, and a repeat mixed chimera experiment given the unexpected results.

Concerns:

1. New Fig. 4e: For their ABC gating, given that they have included all B220+ cells, it would be important to check that they are excluding pDCs, and plot CD11c against CD19 instead of CXCR5; ABCs are CD19^{high} and clearly separated from pDCs that are CD19^{low}. They may well be enumerating pDCs, which may explain why ABCs are not appear increased in their mixed chimeras.
2. In new Fig. 4f there does not appear to be a clear pDC population. If the authors want to conclude pDCs are increased, they cannot do it based on the staining below. They can use a specific pDC marker such as Siglec-H, together with CD11c, which should identify a discrete population. If the authors want to claim a statistically significant effect in pDCs as the mechanism underpinning PLD4 pathogenicity, it needs to be shown robustly. They should also show all events instead of a contour, to make sure there are enough events to be robust.
3. In new Fig. 4g there does not appear to be a convincing plasma cell population. The authors should use a second plasma cell marker like CD98 or TACI instead of CD44 and show how their PC gate confidently identifies plasma cells in a secondary lymphoid tissue from the same mice side by side.
4. It is very surprising that PLD4 does not act in a B cell intrinsic manner for the observed increase in ABCs (if statistically significant) or plasma cells, given the increased TLR7 signalling and the fact that PLD4 is expressed very highly in B cells according to Immgen RNAseq. The results in their new mixed chimera experiment are all over the place: ABCs are not increased, and neither are plasma cells. There may have been a genotyping problem in the donor mouse. This mixed chimera experiment needs to be repeated for the authors to conclude that PLD4 only acts intrinsically in pDCs. The authors also need to show the FACS plots for this experiment, including a convincing gating for pDCs using a specific marker like Siglec-H.
5. In straight mice, the PDC increase is not statistically significant (Extended data Fig. 8a). Given that this effect of PLD4 on PDCs is proposed to underpin the pathogenic mechanism of PLD4 deficiency in their study, they need to make this finding more robust by increasing the power to detect the difference, which may show whether there is a real difference or not. They also need to show total pDC numbers.
6. In Extended data Fig. 8e, the plasma cell gating does not appear reliable. The diagonal seen in WT mice indicates autofluorescence. The authors should use a dump channel for autofluorescent and myeloid cells, and then gate PCs as CD138 vs B220 (CD138⁺ cells should be B220 low or negative) and CD138 vs CD98 or TACI.

Referee #3

(Remarks to the Author)

The authors have made a significant revision and addressed all raised questions, including dissecting the cGAS-STING pathways and adding jSLE non genetic controls.

Version 2:

Reviewer comments:

Referee #1

(Remarks to the Author)

The authors have now performed high quality stains that allow unequivocal identification and robust quantification of pDCs and plasma cells. They have also repeated the mixed chimeras with increased mouse numbers and improved stains, and they now also see a B cell / plasma cell -intrinsic effect, which makes sense given the expression of PLD4 in B cells and the TLR7 dependence of the phenotype.

It is unclear why the abstract only mentions the kidney as the site with "cell-intrinsic expansion of plasmacytoid dendritic cells and plasma cells". Was this not seen in the spleen? Given that they see robust splenic phenotypes throughout, it is not clear why they authors do not show the spleen results in the mixed bone marrow chimera experiments. For completion and as a minor revision, I suggest they include the spleen data in the figures.

Reply to the Referee #1

Referee #1 (Remarks to the Author):

This study by Wang et al., identifies five patients with systemic lupus erythematosus (SLE) carrying compound heterozygous PLD4 variants that cause loss of exonuclease function. GWAS studies have identified PLD4 as a lupus susceptibility gene and PLD4 deficiency in mice has been shown to result in the development of autoimmunity through increased TLR7 signaling, and in a Balb/c background, also increased TLR9 signaling. Wang et al's findings are stringly supportive that biallelic PLD4 loss-of-function (LoF) variants cause - or significantly contribute to - SLE in humans. Given that all variants are rare, biallelic and LoF in vitro, and that there are studies from the Nemazee lab showing PLD4 deficiency causes lupus-like disease in mice that involves TLR7 and TLR9, it is reasonable not to require further proof of variant pathogenicity by introduction of the specific variants into mice, nor further proof that TLR7 and TLR9 are involved in disease pathogenesis. The authors provide some evidence that patients' PBMCs exhibit heightened sensitivity to TLR9 ligand (CpG) stimulation, although the fold difference is less marked than unstimulated cells, thus caution is needed when interpreting these data. Overall, these findings strongly support a pathogenic role of PLD4 deficiency in human SLE.

Reply: Thank this reviewer for dedicating the time to evaluating our study. We appreciate the positive impressions, constructive feedback and valuable suggestions. We have carefully addressed each comment below.

Main points:

1. The magnitude of differences in TNF and type I IFN-related RNAs are higher in unstimulated cells compared with CpG stimulation, suggesting that the differences may not be due so much to the nature of the stimulus, but simply to their hyperactivated status. For example, there is a 3-4-fold increase in TNF mRNA in unstimulated cells compared with a 1.5-fold difference in cells after CpG stimulation. The authors could state their conclusions with some caution.

Reply: We thank the reviewer for this critical observation. To clarify, blood samples from the same patients (Patient P1 or patient P2) identified two inflammatory phenotypes with different baseline inflammatory activity (**Fig. R1a, upper**). qPCR analysis of two patients (P1 and P2) demonstrated that PBMCs isolated during periods of low baseline inflammation (P1.1, P2.1) exhibited higher response to CpG-DNA stimulation compared to the PBMCs with elevated baseline inflammation (P1.2, P2.2), such as *TNF*, *ISG15*, and *RSAD2* (**Fig. R1a, lower and Fig. R1b**), suggesting potential cellular exhaustion. But the PBMCs in patient P1 and P2 indeed show no significant sensitivity change compared to healthy controls.

Fig. R1, a, qPCR detected the upregulated expression of genes related to NF- κ B and type I interferon signaling pathways in the PBMCs from PLD4 deficiency patients, healthy controls and SLE patients without *PLD4* mutations under untreated (**upper**) and CpG-DNA stimulation (**lower**). **b**, Fold induction of genes in (**a**) following CpG-DNA stimulation. HC: Health controls, n = 6; SLE: SLE patients without *PLD4* mutations, n = 6. P1.1 and P2.1: PBMCs with low baseline inflammation in patients P1 and P2; P1.2 and P2.2: PBMCs with elevated baseline inflammation in patients P1 and P2. Data are mean \pm s.e.m.

To avoid the heterogeneity of PBMCs between different patients and further detect the sensitivity change in PLD4-expressed cells (PLD4 is highly expressed in dendritic cells, monocytes and B cells), we isolated monocytes from patient P1 (low baseline inflammation) and stimulated with CpG-DNA. The results showed that the inflammatory genes in monocytes derived from P1 were not significantly different from those in healthy controls at basal level. When stimulated with CpG-DNA, P1-derived monocytes showed higher sensitivity in some inflammatory genes, like *TNF*, *IL6*, and *IFNA2* compared to healthy controls (**new Extended Data Fig. 6f and 6g**).

new Extended Data Fig. 6 f,g, qPCR detected the upregulated expression of genes related to NF- κ B and type I interferon signaling pathways in the monocytes from PLD4 deficiency patient P1 and healthy controls under untreated (**f**, **upper**) and CpG-DNA stimulation (**f**, **lower**). **g**, Fold induction of genes in (**f**) following CpG-DNA stimulation. P1: PBMCs with low baseline inflammation. Data are mean \pm s.e.m.

Based on these data, we agree to remove the description of PBMCs sensitivity to CpG-DNA and revised the original interpretation to “*Additionally, RNA-Seq analysis of PBMCs from patients P1 and P2 revealed significantly elevated type I interferon pathway gene expression in patients compared to healthy controls, both at basal level and upon CpG-DNA stimulation (Fig. 3d).*” on Page 6, line 158 to 160.

Besides, we added “*Notably, upon CpG-DNA stimulation of isolated monocytes from patient P1, several genes elevated fold upregulation relative to healthy controls (Extended Data Fig. 6f, 6g).*” on Page 6, line 167 to 169.

We appreciate the reviewer’s emphasis on caution, and these revisions ensure our descriptions align strictly with the data.

2. Given the existing controversies of TLR9 role in SLE with data suggesting that TLR9 also plays protective roles (PMID: 36151396), it would be useful for the reader if the authors mention the different - and for some phenotypes, opposite - results observed in Balb/c (PMID: 37555846) vs C57BL/6 mice that lack PLD4. It is only in the Balb/c mouse background where TLR9 clearly contributes to lupus-like autoimmunity, as opposed to myeloid cell driven autoinflammation.

Reply: We thank the reviewer for this suggestion, and we agree that addressing this request to expand the mechanistic explanation of PLD4 deficiency, especially clarifying the role of TLR9 in this context, is necessary to prevent readers from oversimplifying the mechanism of PLD4 deficiency and ignoring other critical pathways involved, such as TLR7 and cGAS-STING signaling pathways.

Endosomal TLR7 and TLR9 recognize endogenous RNA and DNA, initiating downstream immune signaling pathways¹. The role of TLR7 has been well-documented, capable of inducing autoimmune or lupus-like phenotypes in mice and humans²⁻⁴. In contrast, TLR9 exhibits more complex functions. Earlier knockout studies in MRL/lpr mice demonstrated that TLR9 counteracts TLR7 by suppressing disease progression^{5,6}, with this lupus-inhibitory mechanism attributed to enhanced TLR7 activation following TLR9 deletion. Some evidence also indicates that abnormal TLR9 modification may promote SLE progression⁷. TLR9 displays cell type-specific functionality: its deficiency in B cells exacerbates nephritis, while B cell-specific Tlr9 overexpression alleviates renal pathology⁸. In pDCs, TLR9 drives excessive type I interferon production and directly regulates T/B cell activation⁹. Notably, recent studies have revealed that TLR9 exhibits dual intrinsic functions: MyD88-dependent pro-inflammatory effects and MyD88-independent protective effects (either ligand-binding dependent or independent), which offers a molecular explanation for understanding its context-dependent complexity¹⁰.

Regarding PLD4 deficiency, previous studies have shown that knockout of TLR9 in the BALB/c background can completely rescue the autoimmune phenotype caused by PLD4 deficiency¹¹, suggesting that the phenotype in this background is mediated by TLR9. In the C57BL/6 background, *Tlr9^{CpG11/CpG11} Pld3^{-/-} Pld4^{-/-}* can rescue the survival of *Pld3^{-/-} Pld4^{-/-}* mice, but the mice still have severe hepatosplenomegaly, erythrocytopenia, and increased monocytes infiltration, suggesting that other pathways are involved in the disease process¹². Further experiments also found that *Unc93b1^{3d/3d} Pld3^{-/-} Pld4^{-/-}* can effectively alleviate phenotypes such as splenomegaly and thrombocytopenia through TLR7/9 signaling blockade, and the activation of the type I interferon signaling pathway that still exists can be completely rescued by knocking out STING. These results suggest that in the C57BL/6 background, in addition to TLR9, TLR7 (and other endosomal TLR signaling, like TLR13) and cGAS-STING signaling pathways contributed significantly to the development of the diseases following Pld3/Pld4 deficiency¹².

Despite distinct genetic backgrounds driving TLR9-mediated autoimmune phenotypes

in BALB/c mice and severe IFN- γ -dependent autoinflammatory phenotypes in C57BL/6 mice, TLR9 predominantly exerts TLR9-MyD88-mediated proinflammatory functions under PLD4 deficiency-induced substrate accumulation. This is evidenced by a complete phenotypic rescue in BALB/c PLD4 deficiency mice after TLR9-deficient¹¹, survival restoration in C57BL/6 *Pld3*^{-/-}*Pld4*^{-/-} mice upon TLR9 deletion, reduced inflammatory cytokine expression, and alleviation of survival rates, body weight loss, and B-cell population defects through anti-TLR9 antibody treatment following *Pld3/Pld4* deficiency¹².

Based on this, we added the following sentences to address this issue:

*“Recent study identified MyD88-independent protective roles and MyD88-dependent proinflammatory role of TLR9, which offers a molecular explanation for understanding its context-dependent complexity. In murine models of PLD4 deficiency, TLR9 exhibits context-dependent roles across genetic backgrounds. Within the C57BL/6 strain, TLR9-mediated autoinflammation cooperates with TLR7 and cGAS-STING signaling pathways to drive disease pathogenesis in *Pld3/Pld4* deficiency mice. Besides, in BALB/c mice, TLR9-driven autoimmunity following PLD4 deficiency is the cause of disease in this background. These findings, combined with our CpG-DNA stimulation experiments in PLD4-deficient cell lines and patient-derived cells, suggest that substrate accumulation caused by PLD4 deficiency shifts the balance between TLR9 protective effects and TLR9 proinflammatory activity toward the latter.”* on Page 10, line 278 to line 289.

3. The cell subset characterization by scRNAseq in Fig. 4 is not very useful (besides the labels not being clear); measurements of total B and T cells are not very meaningful or informative.

Reply: We thank the reviewer for this suggestion. We have moved (**Fig. 4a, 4b in previous version**) to the (**new Extended Data Fig. 7g and 7h**). Additionally, we observed that the colors in the UMAP plot could lead to confusion, so we have aligned the cell subset names and their corresponding positions in new figures.

new Extended Data Fig. 7g, UMAP plots display the differences in cell type between WT and *Pld4*-deficient mice kidneys. **h**, The proportion of total or various types of kidney infiltrating immune cells in WT and *Pld4*-deficient mice.

Given the recent advances in characterization of cell subsets thought to be pathogenic in SLE (i.e. aged associated B cells, plasmacytoid DCs) and the intriguing contribution

of TLR9, it would be useful if the authors characterized and quantified the different B cell and DC subsets show if the changes are cell autonomous (using mixed bone marrow chimeras) or dependent on the inflammatory milieu.

Reply: We appreciate Referee #1 for raising this constructive suggestion. To verify the increase in B cells and DCs observed in scRNA-Seq and to further assess whether B cell subsets (like ABCs and Plasma cells) and DC subsets (like pDCs and mDCs) are increase in the kidney of *Pld4*-deficient mice, we performed flow cytometry analysis on kidney cells from *Pld4*-deficient and WT mice. The results indicated a significant increase in total immune cells (**new Fig. 4d**), ABCs (**new Fig. 4e**), plasma cells (**new Fig. 4g**) and pDCs (**new Fig. 4f**) in *Pld4*-deficient mice compared with WT mice, while mDCs (**new Extended Data Fig. 7l**) remained unchanged.

new Fig. 4 d-g, Representative the kidney flow cytometry results of WT and *Pld4*-deficient mice: immune cell (CD45⁺) in (d), ABCs (CD3⁺CD19⁺B220⁺CXCR5⁺CD11c⁺) in (e), pDCs (CD3⁻CD19⁻CD11b⁻B220⁺MHC-II⁺CD317⁺) in (f), Plasma cells (CD138⁺CD44⁺) in (g). WT, n = 6; *Pld4*^{+/-}, n = 6; *Pld4*^{-/-}, n = 6. Data are mean ± s.e.m. These results show a representative of at least three independent experiments in (d, e, f, g). **P* < 0.05; ***P* < 0.01; ****P* < 0.001. One-way ANOVA with Tukey's post hoc analysis was used in (d, e, f, g).

Extended Data Fig. 7 k, l, Representative the kidney flow cytometry plot and quantification of WT and *Pld4*-deficient mice: Plasma cells (CD45⁺CD138⁺MHC-II⁺) in (k), mDCs (CD45⁺CD3⁻CD11b⁺CD11c⁺MHC-II⁺) in (l). WT, n = 6; *Pld4*^{+/-}, n = 6; *Pld4*^{-/-}, n = 6. Data are mean ± s.e.m. These results show a representative of at least three independent experiments in (k, l). *n.s.*, no significant difference; **P* < 0.05; ***P* < 0.01. One-way ANOVA with Tukey's post hoc analysis was used in (k, l).

Additionally, since GC B cells and MZ B cells, which are important for the development of SLE, were absent in the kidney, we assessed the proportion of immune cell subsets of splenic cells from Pld4-deficient and WT mice. The results revealed an upregulation of Plasma cells, T_{reg} cells, Monocytes, and Macrophages, while no significant differences were observed in the proportions of MZ B cells, GC B cells, ABCs, Follicular B cells, pDCs, mDCs, Th1, Th17, eT_h, T_{FH}, and other cell subsets between Pld4-deficient and WT mice (**new Extended Data Fig. 8**). These findings consistent with the results of RT-qPCR from different tissues, which indicated a higher upregulation of renal inflammation levels in the PLD4-deficient mice.

Extended Data Fig. 8

new Extended Data Fig. 8 a-m, Representative the spleen flow cytometry plots and quantification of WT and *Pld4*-deficient mice: pDCs (CD45⁺CD3⁺CD19⁺CD11b⁺B220⁺MHC-II⁺CD317⁺) in (a), mDCs (CD45⁺CD3⁺CD11b⁺CD11c⁺MHC-II⁺) in (b), ABCs (CD45⁺CD3⁺CD19⁺B220⁺CXCR5⁺CD11c⁺) in (c), GCB (CD45⁺CD3⁺CD19⁺B220⁺BCL6⁺CD95⁺) in (d), Plasma cells (CD138⁺CD44⁺) in (e) or (CD45⁺CD138⁺MHC-II⁺) in (f), Th1 cells (CD45⁺CD3⁺CD4⁺T-bet⁺) in (g), CD4 Effector cells (CD45⁺CD3⁺CD4⁺CD62L⁺CD44⁺) in (h), Th17 cells (CD45⁺CD3⁺CD4⁺IL-17A⁺) in (i), Treg cells (CD45⁺CD3⁺CD4⁺Foxp3⁺CD25⁺) in (j), eT_h (CD45⁺CD3⁺CD4⁺PD-1⁺CXCR5⁺) and T_{Fh} (CD45⁺CD3⁺CD4⁺PD-1⁺CXCR5⁺) in (k), Follicular

B cells (CD45⁺CD3⁻CD19⁺B220⁺CD23⁺CD21⁻) and MZ B cells (CD45⁺CD3⁻CD19⁺B220⁺CD23⁻CD21⁺) in **(l)**, Monocytes (CD45⁺CD3⁻CD19⁻B220⁻CD11b⁺Ly6C⁺F4/80⁻) and Macrophages (CD45⁺CD3⁻CD19⁻B220⁻CD11b⁺Ly6C⁻F4/80⁺) in **(m)**. WT, n = 5; *Pld4*^{+/-}, n = 5; *Pld4*^{-/-}, n = 5. Data are mean ± s.e.m. These results show a representative of two independent experiments in **(a-m)**. n.s., no significant difference; **P* < 0.05; ***P* < 0.01; ****P* < 0.001. One-way ANOVA with Tukey's post hoc analysis was used in **(a-m)**.

To determine which immune cells infiltrating the kidneys were cell-intrinsic, mixed bone marrow (BM) chimeras were generated by transplants of BM(1:1 ratio of B6-CD45.1 and either WT or *Pld4*-deficient (B6-CD45.2)) into lethally irradiated CD45.1 mice. 10 weeks after transplantation, autoantibody analysis revealed that chimeric mice reconstituted with *Pld4*^{-/-} BM exhibited elevated levels of anti-dsDNA and anti-dsRNA antibodies (**new Fig. 4h**). Flow cytometry results revealed that only the expansion of pDCs was cell-intrinsic, while the expansion of B cells and T cells were cell-extrinsic (**new Fig. 4i**). This finding is consistent with the highly conserved expression of *PLD4* in pDCs and the abnormal inflammatory upregulation of pDCs observed in patient PBMCs (**new Fig. 2f and 2g**), further emphasizing the critical role of pDCs in the development of SLE in *Pld4*-deficient mice.

new Fig. 4 h, i, Representative the autoantibodies and flow cytometry results of mixed bone marrow chimeric mice reconstituted with 1:1 ratio of CD45.1-WT: CD45.2-WT (+/+:+/+) or CD45.1-WT: CD45.2-*Pld4*^{-/-} (+/+:-/-) bone marrow. **h**, ELISA detected the levels of anti-dsDNA and anti-dsRNA antibodies in the plasma of chimeric mice after 10 weeks transplantation. **i**, Flow cytometry detected the immune cell phenotyping in the kidneys of chimeric mice after 10 weeks transplantation. +/+:+/+, n = 4; +/+:-/-, n = 6. Data are mean ± s.e.m. These results show a representative of two independent experiments in **(h, i)**. n.s., no significant difference; **P* < 0.05; ***P* < 0.01. Two-way ANOVA was used in **(i)**, unpaired *t*-test was used in **(h)**.

Based on these findings, we have incorporated the following statement to address this issue in the Discussion section: “*pDCs, a unique cell population central to antiviral responses through nucleic acid sensing and robust type I interferon production*^{9,13}. *PLD4* exhibit evolutionarily conserved high expression in *pDCs*¹⁴ (versus *PLD3*), implicating its non-redundant role in *pDCs* nucleic acid homeostasis. *PLD4*-deficient patients uniformly develop lupus nephritis, with *scRNA-Seq* and *CyTOF* analyses suggesting *pDCs* as the predominant cellular drivers of upregulated type I interferon and TLR signaling in patients’ PBMCs. Mirroring human pathology, *Pld4*-deficient mice show preferential renal involvement, while mixed bone marrow chimeras demonstrate *pDCs* as the only cell-intrinsically expanded immune population in

kidneys. This cross-species convergence underscores pDCs' central regulatory role in PLD4-deficiency-mediated immune dysregulation.” on Page 10, line 291 to Page 11, line 300.

4. The authors show response to baricitinib in B6 mice lacking PLD4. In the B6 background TLR9 may not be significantly contributing to disease given Nemazee's results in their 2018 and 2021 papers, in which TLR9 deficiency in fact exacerbates some disease features including thrombocytopenia. Thus, a word of caution is needed: the results seen in B6 mice may not necessarily be extrapolated to patients. This should at least be discussed.

Reply: We thank the reviewer for raising this critical point regarding the translational caution and we acknowledge the need for caution when extrapolating results across species and genetic backgrounds.

In addition to TLR9, TLR7 (and other endosomal TLR signaling) and cGAS-STING pathway significantly contribute to the disease features after Pld3/Pld4 deficiency according to Nemazee's studies¹². Our experiments in THP-1 *PLD4* knockout cell lines also revealed significant activation of the cGAS-STING pathway^{12,15}. These findings suggest that immune dysregulation following PLD4 deficiency in the C57BL/6 background involves multiple signaling pathways.

Activation of TLR7 and TLR9 in pDCs drives robust type I interferon production⁹, whereas constitutive STING activation underlies the type I interferonopathy - STING-associated vasculopathy with onset in infancy (SAVI)¹⁶⁻¹⁸. Importantly, regardless of the upstream triggers, the use of the JAK inhibitor Baricitinib in this study was based on the abnormal activation of the type I interferon pathway observed in both patients, cell line and mice. Our PLD4-deficient patients' PBMCs displayed constitutive IFN-I activation, suggests that JAK inhibition may benefit patients with IFN-high signature, regardless of the magnitude of TLR9 contribution.

The clinical trials of JAK inhibitors in SLE patients may provide indirect evidence^{19,20}, but targeted studies in PLD4-deficient patients are essential. Based on this, we agree that caution is warranted when extrapolating mouse model data to human disease, especially since human SLE patients have greater heterogeneity, and as such, we have added the Discussion section on *Page 11, line 319 to 321* to address this issue:

“Conserved type I interferon signaling in PLD4-deficient models supports the translational applicability of JAK inhibitor, though interspecies discrepancies necessitate clinical validation in PLD4 deficiency patients.”

Reply to the Referee #2

Referee #2 (Remarks to the Author):

This is an interesting paper which expands the list of monogenic lupus cases. It is well documented and clearly presented.

Previous monogenic cases in the same pathways have reported mutations of TLR7 and its downstream molecule UNCB1. The PDL4 mutations which are reported herein involve a molecule which is “upstream” of TLR7/9 and involves the defective digestion of DNA/RNA. The PDLA4 mutations therefore can be classified along with DNASEs defects which also account for defective degradation of nucleic acids.

Reply: We sincerely thank this reviewer for the time and thoughtful evaluation of our work, as well as for recognizing its contributions to broadening the genetic and phenotypic spectrum of monogenic lupus. We have carefully addressed each comment below.

The novelty of this manuscript is with the presentation of 5 patients with PDL4 mutations all of whom had SLE a couple in young age and a couple at an older age. PDLA4 deficient mice developing autoimmunity had been described before and the use baricitinib has been in lupus clinical trials. One way to increase the novelty of this paper to construct knock in mice with the patient-identified PDL4 mutations. The author’s present evidence that TLR9 in this setting behaves like TLR7. The role of TLR9 has not been settled and it looks like that it is setting dependent.

Reply: We appreciate the reviewer’s insightful suggestion to generate knock-in (KI) mice harboring patient-identified *PLD4* mutations. While we acknowledge that KI mice could refine our understanding of mutation-specific mechanisms, the current genetic, functional, *in vivo* and *in vitro* evidence already provides strong support for our conclusions in this manuscript that biallelic *PLD4* loss-of-function mutations as causative in human lupus pathogenesis.

All identified mutations are exceedingly rare (<0.001% in gnomAD) and biallelic, with *in vitro* assays demonstrating loss of exonuclease activity (**new Fig. 3a, 3b, new Extended Data Fig. 6a and 6b**). Structural analyses further revealed that these variants span multiple functional domains (**new Extended Data Fig. 1b to 1e**), yet all converge on disrupted enzymatic activity. This functional homogeneity strongly supports a shared mechanism of pathogenicity—impaired nucleic acid degradation—rather than variant-specific effects. Besides, our *Pld4* deficiency mice recapitulates the key SLE phenotypes observed in patients: elevated type I IFN signatures (**human: new Fig. 2a, 2b, 2c new Extended Data Fig. 2a, 2c, 2d, 3b/ mice: new Fig. 4c, new Extended Data Fig. 7f, 7j**), lupus nephritis(**human: new Fig. 1a, new Supplementary table. 1/ mice: new Fig. 4a to 4g, new Extended Data Fig. 7**), the critical role of pDCs (**human: new Fig. 2f, 2g/ mice: new Fig. 4h, 4i**) and the responsiveness to JAK inhibitor (**human: new Fig. 5f/ mice: new Fig. 5a to 5e**). These findings strongly support conserved pathogenic mechanisms between patients and KO mice, and the phenotypic

concordance between patients and KO mice validates the utility of KO mice for mechanistic and therapeutic exploration.

Combined functional evidence of *PLD4* biallelic loss-of-function mutations and cross-species phenotypic concordance demonstrate that impaired nucleic acid degradation – rather than mutation-specific effects – drives pathogenesis, with existing KO models providing sufficient mechanistic validation.

Moreover, the novelty of our work not only in identifying human *PLD4* mutations as causative for lupus, thereby expanding the spectrum of monogenic lupus, but also in elucidating the cellular basis of immune dysregulation following *PLD4* deficiency. Through experimental investigation of both *PLD4*-deficient patients and murine models, we demonstrate the critical roles of pDCs in mediating immune dysregulation upon *PLD4* deficiency. Most importantly, our investigation of *PLD4*-deficient models has revealed its most prominent molecular signature: hyperactivation of the type I interferon signaling pathway. Based on this, we choose the JAK inhibitor Baricitinib to inhibit inflammation by targeting this pathway. Notably, *PLD4*-deficient mice exhibited favorable responses to Baricitinib. This finding suggests that Baricitinib - not currently a first-line clinical option for SLE - may hold therapeutic potential for patients harboring *PLD4* deficiency.

In summary, we believe that our study not only identifies *PLD4* as a novel disease-causing gene of human lupus, but also establishes the pivotal role of pDCs in driving immune dysregulation following *PLD4* deficiency. Furthermore, preclinical validation in animal models highlights Baricitinib as a promising potential targeted therapy for lupus patients with *PLD4* deficiency.

In a number of places the writing can be improved.

Reply: We appreciate the reviewer's feedback. Upon thorough re-examination of the original text, we identified several areas requiring refinement: some terminology in the manuscript is ambiguous and the grammatical errors. For instance, using “patients” when referring specifically to patients P1 and P2 might lead the reader to assume that it refers to all patients.

We have now carefully checked the revised manuscript to ensure consistency between figures and descriptions and thoroughly reviewed the grammar. We trust these revisions substantially improve manuscript rigor and readability and we hope these revisions can satisfy the concerns raised.

In lines 180-200 they refer to figure 4 as figure 3.

Reply: We thank the reviewer for the valuable comments, we have corrected the reference to **new Figure 5** on *Page 9, line 246 to 262*.

Reply to the Referee #3

Referee #3 (Remarks to the Author):

This manuscript reports on five patients with both early and adult-onset SLE who carry biallelic mutations in *PLD4*, a putative new cause of monogenic SLE. The

recent discoveries of new genes causal for Mendelian forms of lupus are critically important for understanding the underlying mechanisms of tolerance breakdown in Human.

The work includes experiments in primary cells from patients, mouse models and cell lines, and the findings are globally compelling.

Reply: We sincerely thank this reviewer for the time and effort invested in reviewing our manuscript. Your encouraging remarks and insightful comments are greatly valued, and we have responded to each comment in the following.

I have some suggestions / questions regarding the current work.

-The genetic data are well detailed with position, in silico scoring, and all together these data are convincing. Additional structural data with the position of the variants may be informative if the variants cluster in a specific region.

Reply: We thank the reviewer for raising this important point. While the *PLD4* variants identified in patients do not spatially cluster within a single structural domain (**new Extended Data Fig. 1b**), but our structural analysis of *PLD4* variants revealed functional effects through distinct mechanisms.

D189E, R201Q and Y248C mutations were found to disrupt hydrogen-bonding with neighboring residues (**new Extended Data Fig. 1c**). P181 is located within the substrate-binding pocket, and the P181L may impair ssDNA interaction by disrupting the conformational rigidity of this critical region (**new Extended Data Fig. 1d**).

Notably, A323 and G457 are very close in spatial structure and the G457D mutation shortens the distance between A323 and D457. In addition, a new hydrogen bond is formed between D457 mutant and G441 (**new Extended Data Fig. 1e**). This aberrant interaction may correlate with the most severe defects of A323V and G457D mutations in enzymatic assays (**new Fig. 3a**), highlighting the functional significance of this structural domain.

In summary, although these mutations identified in patients do not spatially cluster within a specific region, the structure predicts that they may disrupt *PLD4* function through different mechanisms.

new Extended Data Fig. 1 b, Schematic diagram of the position of the mutations site in the PLD4 protein structure (PDB: 8V08). **c-e**, Potential effects of spatial structural simulation mutations on PLD4 protein function, including the disruption of hydrogen bonding with neighboring amino acid residues (**c, e**) and altered DNA-binding capacity (**d**).

-Regarding clinical data, it interesting to observe that all patients presented with a lupus nephritis. Similar to DNASE1L3 deficiency (associated with extracellular nuclease defects), some patients had urticaria. Was the diagnosis consistent with hypocomplementemic urticaria vasculitis (Mc Duffie syndrome)? Were there any signs of brain calcification, paresis, leukodystrophy? Analyzing ANCA systematically would be valuable.

Reply: The diagnosis of HUVS is primarily based on the Schwarz criteria, which encompasses both clinical manifestations and laboratory findings²¹. In the case of Patient P2, she fulfilled two major criteria: chronic recurrent urticaria lasting more than six months and hypocomplementemia. Additionally, she met three minor criteria: arthralgias, glomerulonephritis, and a positive C1q antibody test. Based on this comprehensive assessment, Patient P2 was deemed to meet the diagnostic criteria for HUVS.

The differentiation between HUVS and SLE remains challenging due to overlapping clinical presentations and diagnostic criteria^{22,23}. HUVS can manifest independently or in conjunction with SLE. Notably, approximately 50% of patients initially diagnosed with HUVS are subsequently diagnosed with SLE during follow-up²⁴. In the case of Patient P2, she had biopsy-proven lupus nephritis and satisfied at least seven of the eleven American College of Rheumatology (ACR) classification criteria for SLE,

including arthralgia, serositis, lupus nephritis, hematologic disorder, positive antinuclear antibody (ANA), positive anti-double-stranded DNA (anti-dsDNA), and hypocomplementemia. Consequently, Patient P2 can be diagnosed as SLE and lupus nephritis as well.

Of course, there is partial overlap in clinical phenotypes between HUVS and SLE. Although the Patient P2 could be diagnosed with HUVS, the presence of autoantibodies and the kidney biopsy results led us to diagnose the patient with SLE in this paper. This underscores the importance of conducting research on genomic rare variants in SLE patients. Grouping patients according to the common genetic basis of the disease may circumvent the challenges posed by overlapping clinical phenotypes, thereby enhancing our understanding and treatment of these complex disorders.

P2 did not exhibit any clinical signs of brain calcification, paresis, or leukodystrophy. However, she declined to undergo computed tomography (CT) or magnetic resonance imaging (MRI) of the brain, thereby limiting the ability to definitively rule out these potential imaging manifestations.

A reevaluation of the ANCA profiles for the five patients was conducted. The results indicated that Patients P3 and P5 tested positive for myeloperoxidase (MPO)-ANCA, while Patients P1, P2, and P4 tested negative for ANCA profile. This updated information has been incorporated into the revised **new Supplementary table. 1**.

-IgG level is dramatically reduced in patients 1 and 2 and increase in Pt4/5. How were B cell compartments in these patients and globally in KO Mice? (number and phenotype)

Reply: IgG levels were assayed in P1/2 (Time point 1 in **new Extended Data Fig. 5a**) after prolonged treatment with steroids or multiple immunosuppressants (MMF, FK506, etc.). In these patients, the number and percentage of CD20⁺ B cells were decreased, with counts of 55/ μ l and 38/ μ l and percentages of 9.4% and 3%, respectively. Conversely, IgG levels were measured in P4/5 (Time point 1 in **new Extended Data Fig. 5a**) during periods of high disease activity and without treatment with steroids or immunosuppressants. In these patients, the number of CD20⁺ B cells was within normal range (111/ μ l and 129/ μ l, respectively), while the percentage of CD20⁺ B cells was slightly elevated (15.07% and 16.3%, respectively). During follow-up, as the disease was controlled, IgG levels normalized in P2, P4, and P5 (**new Extended Data Fig. 5a**).

We conducted a further analysis of the B cell compartments in P1 and P4 using the available blood samples. The results revealed that proportions of ABCs (CD19⁺CD11c⁺), Plasma cells (CD19⁺CD38⁺CD138⁺), Naive B cells (CD19⁺CD27⁻IgD⁺), and Memory B cells (CD19⁺CD27⁺IgD⁻) in both patients P1 and P4 during remission phases, aligned with those observed in healthy controls. In contrast, active-phase measurements of P4 exhibited profiles consistent with SLE patients control (**Extended Data Fig. 5b to 5f**), suggesting the activation of B cell function in patient P4.

Extended Data Fig. 5

new Extended Data Fig. 5 a, IgG levels of four patients at different stages. The green box represents the normal level of IgG. W/O: without treatment. T: treatment with steroids or multiple immunosuppressants. **b-f**, Representative the flow cytometry plot and quantification of IgG level (CD19⁺IgG⁺, **b**), ABCs (CD19⁺CD11c⁺, **c**), Plasma cells (CD19⁺CD38⁺CD138⁺, **d**), Naïve B cells (CD19⁺CD27⁺IgD⁺, **e**), Memory B cells (CD19⁺CD27⁺IgD⁻, **e**), Double negative B cell (CD19⁺CD27⁻IgD⁻, **e**), non-class-switched B cells (CD19⁺CD27⁺IgD⁺, **e**) and CD21^{low} B cells (CD19⁺CD21^{low}, **f**) in patients P1, P4, health controls and SLE patients controls. Data are mean \pm s.e.m. HC: Health controls; P1 R: patient P1 remission phase; P4 R/F: patient P4 remission phase /flare phase; PC: SLE patients control without *PLD4* mutations. Blue circles in patients P1 or P4: remission phase in patients P1 or P4; Red circles in patient P4: flare phase in patient P4. These results show a representative of two independent experiments in (**b**, **c**, **d**, **e**, **f**). * $P < 0.05$; ** $P < 0.01$; *** $P < 0.001$. unpaired *t*-test was used in (**b**, **c**, **d**, **e**, **f**).

We also examined the B cell compartments and IgG levels in *Pld4*^{-/-} mice. The results

show IgG concentrations in plasma (new Extended Data Fig. 7a), along with proportions of ABCs, and plasma cells, were markedly elevated in the kidney of *Pld4*^{-/-} mice (new Fig. 4e and 4g).

new Extended Data Fig. 7 a, shows ELISA detected IgG levels in the plasma of *Pld4*-deficient and WT mice. Female: WT, n = 8; *Pld4*^{+/+}, n = 9; *Pld4*^{-/-}, n = 8; Male: WT, n = 9; *Pld4*^{+/+}, n = 12; *Pld4*^{-/-}, n = 8. Data are mean ± s.e.m. Experiment in (a) was performed once with n > 8. **P* < 0.05; ***P* < 0.01; ****P* < 0.001. One-way ANOVA with Tukey's post hoc analysis was used in (a).

new Fig. 4 e, g, Representative the kidney flow cytometry results of WT and *Pld4*-deficient mice: ABCs (CD45⁺CD3⁻CD19⁺B220⁺CXCR5⁻CD11c⁺) (e), Plasma cells (CD138⁺ CD44⁺) (g). WT, n = 6; *Pld4*^{+/+}, n = 6; *Pld4*^{-/-}, n = 6. Data are mean ± s.e.m. These results show a representative of at least three independent experiments in (e, g). **P* < 0.05; ***P* < 0.01; ****P* < 0.001. One-way ANOVA with Tukey's post hoc analysis was used in (e, g).

Therefore, kidney-biopsy from the five patients were subjected to CD138 immunostaining for plasma cells. Varying degrees of plasma cell infiltration were consistently observed. The infiltration is predominantly localized to the interstitium and

periglomerular regions. No intraglomerular plasma cell infiltration was detected throughout all specimens (**Fig. R2**).

Fig. R2, Numerous CD138⁺ plasma cells (brown) in the periglomerular (left) and interstitium (right) regions of kidney biopsy from the patient P2 (400 \times).

In summary, long-term therapeutic management in patients P1 and P2 achieved IgG levels comparable to or lower than healthy controls, with flow cytometry analysis revealing normal IgG production and B cell compartments in patient P1. In contrast, patients P4 and P5 during active disease demonstrated markedly elevated IgG levels, which returned to normal ranges after treatment. Flow cytometry analysis of patient P4 showed B cell activation patterns resembling those of active SLE controls during disease flares, with normalization of most populations following therapeutic intervention.

-PLD4 is said to be primarily expressed in endosomes in the introduction, while other reports mention it is also expressed in ER / golgi / Cytoplasm. Have the authors analysed the subcellular expression of PDL4? Importantly, the inflammation of Pld4 KO mice was also reported as secondary to intracellular sensing via Sting. Indeed, the IFN-I signature is persistent in *Unc93b13d/3dPld3^{-/-}Pld4^{-/-}* mice (Gavin *et al.* 2021). Exploring the cGAS/STING pathway in cell lines would add to the understanding of immune dysregulation of PDL4 deficiency.

Reply: We acknowledge the constructive feedback from the reviewer and note that while earlier studies reported co-localization of PLD4 with Calnexin (marked by ER) and Golgin97 (marked by Golgi)²⁵, subsequent investigations have established PLD4 and its paralog PLD3 as endolysosomal exonucleases^{1,12,15,26-32}. Fazzari *et al.*²⁶ reported that PLD3 exhibits lower abundance in early endosomes (marked by EEA1) and higher abundance in late endosomes and lysosomes (marked by LAMP1), with its loss-of-function increasing Alzheimer's disease risk through impaired APP processing. Gavin *et al.*^{12,27} demonstrated that PLD3 and PLD4 are most active at an acidic pH (5–5.5) and are localized to endolysosomes, where they maintain immune homeostasis by restricting TLR7 and TLR9 activation. Bérouti *et al.*²⁹ revealed that complex RNA molecules in the endolysosomal compartment undergo sequential processing: initial cleavage by RNase T2 followed by degradation via PLD exonucleases (PLD3 and PLD4). Yuan *et al.*³¹ determined the structures of endolysosomal exonucleases PLD3 and PLD4, elucidating their catalytic mechanisms. Notably, Singh *et al.*²⁸ recently expanded their function beyond nucleic acid homeostasis, demonstrating that PLD3 and PLD4 also regulate lysosomal lipid metabolism. They found that PLD3 and PLD4 have significant co-localization with LAMP1 and can catalyze the synthesis of BMP in lysosomes. Genetic ablation of PLD3 or PLD4 in mice caused significant BMP depletion, triggering gangliosidosis and lysosomal abnormalities.

These findings underscore PLD4 and its paralog PLD3 as pivotal endolysosomal proteins coordinating both nucleic acid homeostasis and lipid metabolic processes. Consistent with these findings, our immunofluorescence analysis similarly revealed pronounced co-localization between PLD4 and LAMP2 (**Fig. R3**).

Fig. R3, Immunofluorescence shows co-localization of PLD4 and LAMP2 in HeLa cell overexpressing PLD4. Scale bar: 5 μm.

Previous investigations have demonstrated that STING-dependent signaling, particularly type I interferon responses, is activated in *Pld3/Pld4*-deficient mice models¹². Further experiments found that the activation of the type I interferon signaling pathway that still exists in *Unc93b1^{3d/3d} Pld3^{-/-} Pld4^{-/-}* can be completely rescued by knocking out STING. These findings highlight the critical role of STING in the immune dysregulation following PLD4 deficiency.

In our study, the phosphorylation of STING and inflammatory pathway were enhanced in THP-1 *PLD4* KO monoclonal cell lines. Treatment with specific inhibitors of STING (H-151 and C-176³³) substantially attenuated the type I interferon signaling activation induced by PLD4 deficiency, while demonstrating limited efficacy in suppressing the NF- κ B pathway (**new Fig. 3g and 3h**). Furthermore, PLD4 and STING double-knockout cell lines corroborated these findings. Both qPCR and western blotting analyses demonstrated the rescue of type I interferon pathway activation upon STING deletion, whereas NF- κ B signaling showed only partial restoration (**new Extended Data Fig. 6k and 6l**).

new Fig. 3 g, h, Representative the change of signaling pathways and downstream inflammatory genes after STING-specific inhibitor: H-151 and C-176 treatment in THP-1 *PLD4* monoclonal KO cell. **g**, Western blotting shows the signaling pathways change after H-151 treatment in THP-1 *PLD4* monoclonal KO cell. **h**, qPCR analysis of genes in NF- κ B and type I interferon with or without H-151 and C-176 treatment in THP-1 *PLD4* monoclonal KO cell. Data are mean \pm s.e.m. These results show a representative of at least three independent experiments in (**h**) and two independent experiments in (**g**). *n.s.*, no significant difference; * $P < 0.05$; ** $P < 0.01$. One-way ANOVA with Tukey's post hoc analysis was used in (**h**).

new Extended Data Fig. 6 k, l, Representative the change of signaling pathways and downstream inflammatory genes after STING KO in THP-1 *PLD4* monoclonal KO cell. **k**, Western blotting shows the signaling pathways change in THP-1 *PLD4* monoclonal KO cell after STING KO. **l**, qPCR analysis shows the change of genes in NF- κ B and type I interferon pathway in THP-1 *PLD4* monoclonal KO cell after STING KO. Data are mean \pm s.e.m. These results show a representative of three independent experiments in (**k**, **l**). *n.s.*, no significant difference; * $P < 0.05$; *** $P < 0.001$. One-way ANOVA with Tukey's post hoc analysis was used in (**l**).

Based on these observations, our experiments establish an important role for the cGAS-STING signaling axis in mediating the immune dysregulation resulting from *PLD4* deficiency in THP-1 cells. These findings align with previous studies showing that STING knockout ameliorates the hyperactivation of type I interferon signaling in *Unc93b1^{3d/3d} Pld3^{-/-} Pld4^{-/-}* mice.

PLD4 deficiency may activate cGAS-STING pathway through multiple mechanisms. As previously demonstrated, the primordial function of cGAS involves sensing cytosolic DNA to initiate STING-dependent autophagic clearance of cytosolic DNA^{33,34}. In *PLD4*-deficient model, cytosolic DNA transported to lysosomes via autophagy fails to undergo proper degradation, resulting in impaired autophagic flux. This blockage leads to persistent accumulation of cytosolic DNA, which consequently activates the cGAS-STING signaling pathway.

Besides, the activation of cGAS-STING pathway also likely stems from excessive accumulation of ssNAs in endolysosomes upon *PLD4* deficiency, which compromises membrane integrity, leading to cytoplasmic leakage of nucleic acids and subsequent cGAS-STING pathway activation. For instance, dysfunctional *PLD3*—a paralog of *PLD4*—has been shown to cause lysosomal accumulation of mitochondrial DNA (mtDNA), resulting in disrupted lysosome-mitochondria crosstalk, impaired lipid metabolism, and eventual mtDNA leakage into the cytoplasm, thereby activating the cGAS-STING pathway¹⁵. Singh *et al.*²⁸ recently studies also demonstrated that *PLD3* and *PLD4* also regulate lysosomal lipid metabolism. *PLD3* or *PLD4* deficiency in mice

significantly reduced BMP levels, leading to gangliosidosis and lysosomal abnormalities.

Collectively, these findings underscore the critical role of cGAS-STING signaling in mediating immune dysregulation following PLD4 deficiency.

-The assays on nuclease activity are particularly convincing on DNA material and to a milder extend to RNA. This could be discussed in the text.

Reply: We optimized the experimental conditions to minimize RNA degradation and performed the RNA exonuclease activity assay. The new results are consistent with DNA exonuclease activity assays. The P181L, D189E, and R201Q mutations exhibited partial loss of exonuclease activity, whereas the Y248C, A323V, and G457D mutations demonstrated a more pronounced reduction in exonuclease activity (**new Fig. 3a**).

new Fig. 3 a, ssNAs exonuclease activity of purified WT PLD4 and different PLD4 mutants in different times. These results show a representative of at least three independent experiments in (a).

-How many times were these experiments repeated?

Reply: We repeated the exonuclease activity experiments at least three times, and the number of repetitions for all experiments is now specified in the figure legends.

-The terms patient “group” is inappropriate if analyzing only one patient (result section, Pt1 Figure 3C).

Reply: We thank the reviewer for this important remark, we have added “P1” on *Page 5, line 141* and revised “patient group” to “Patient P1” on *Page 5, line 142*.

-The overexpression of TLR7 in PBMCs of Pt1 may also occur in non PLD4 lupus patients.

Reply: We thank the reviewer for this suggestion. We have now moved (**Fig. 3c in previous version**) to (**new Extended Data Fig. 6c**).

-Pt1 and Pt2 have been extensively explored and it should always be mentioned in the main and extended Figure legends which patients are involved, avoiding generalizing to all patients

Reply: We thank the reviewer for bringing up this point. We have revised the figure

legends related to the patient data to provide clearer details on the specific patients involved in each experiment.

-The Figure 2D is difficult to read due to overlapping colors among cell types. Was the overexpression of TLR9 in scRNASeq experiment confirmed at protein level? Is it observed in lupus patients without mutation?

Reply: Thanks for these comments, we have aligned cell subset names and their corresponding positions in new figures (**new Fig. 2d**).

new Fig. 2 d, f, Representative the scRNA-Seq results of PBMCs from patients and healthy controls. **d**, Uniform manifold approximation and projection (UMAP) plot of scRNA-Seq depicts differences of various cell types between patients and healthy controls.

Using blood samples from patient P1 for flow cytometry analysis, we examined pDCs and assessed TLR9 expression. The results revealed a significant upregulation of TLR9 in pDCs from patient P1. Additionally, similar upregulation of TLR9 was observed in pDCs from lupus patients without *PLD4* mutations (**new Extended Data Fig. 4c and 4d**), suggesting that TLR9 upregulation in pDCs may be a compensatory response, in line with the observed upregulation of TLR7³⁵.

new Extended Data Fig. 4 c, Representative pDCs gating strategies from human PBMCs. **d**, Flow cytometry plot and quantification of TLR9 expression in pDCs from patients and healthy controls. PC: SLE patients control without *PLD4* mutations. Data are mean \pm s.e.m. The results show a summary of two independent experiments in (**d**). ******* $P < 0.001$. One-way ANOVA with Tukey's post hoc analysis was used in (**d**).

-In the extended table 1, Hypertension for Pt 3 is mentioned among the immunologic

disorders. It would fit better to kidney disorders.

Reply: We thank the Reviewer for the valuable comments. We have adjusted the revised **new Supplementary table. 1** by reclassifying hypertension for Patient P3 under the category of kidney disorders.

Reference:

- 1 Lind, N. A., Rael, V. E., Pestal, K., Liu, B. & Barton, G. M. Regulation of the nucleic acid-sensing Toll-like receptors. *Nat Rev Immunol* **22**, 224-235 (2022). <https://doi.org/10.1038/s41577-021-00577-0>
- 2 Jackson, S. W. *et al.* Opposing impact of B cell–intrinsic TLR7 and TLR9 signals on autoantibody repertoire and systemic inflammation. *The Journal of Immunology* **192**, 4525-4532 (2014). <https://doi.org/10.4049/jimmunol.1400098>
- 3 Brown, G. J. *et al.* TLR7 gain-of-function genetic variation causes human lupus. *Nature* **605**, 349-356 (2022). <https://doi.org/10.1038/s41586-022-04642-z>
- 4 Fairhurst, A. M. *et al.* Yaa autoimmune phenotypes are conferred by overexpression of TLR7. *Eur J Immunol* **38**, 1971-1978 (2008). <https://doi.org/10.1002/eji.200838138>
- 5 Christensen, S. R. *et al.* Toll-like receptor 7 and TLR9 dictate autoantibody specificity and have opposing inflammatory and regulatory roles in a murine model of lupus. *Immunity* **25**, 417-428 (2006). <https://doi.org/10.1016/j.immuni.2006.07.013>
- 6 Nickerson, K. M. *et al.* TLR9 regulates TLR7-and MyD88-dependent autoantibody production and disease in a murine model of lupus. *The journal of Immunology* **184**, 1840-1848 (2010). <https://doi.org/10.4049/jimmunol.0902592>
- 7 Ni, H. *et al.* Cyclical palmitoylation regulates TLR9 signalling and systemic autoimmunity in mice. *Nature communications* **15**, 1 (2024). <https://doi.org/10.1038/s41467-023-43650-z>
- 8 Tilstra, J. S. *et al.* B cell–intrinsic TLR9 expression is protective in murine lupus. *The Journal of clinical investigation* **130**, 3172-3187 (2020). <https://doi.org/10.1172/JCI132328>
- 9 Colonna, M., Trinchieri, G. & Liu, Y. J. Plasmacytoid dendritic cells in immunity. *Nat Immunol* **5**, 1219-1226 (2004). <https://doi.org/10.1038/ni1141>
- 10 Leibler, C. *et al.* Genetic dissection of TLR9 reveals complex regulatory and cryptic proinflammatory roles in mouse lupus. *Nat Immunol* **23**, 1457-1469 (2022). <https://doi.org/10.1038/s41590-022-01310-2>
- 11 Gavin, A. L. *et al.* Disease in the Pld4^{thss/thss} Model of Murine Lupus Requires TLR9. *Immunohorizons* **7**, 577-586 (2023). <https://doi.org/10.4049/immunohorizons.2300058>
- 12 Gavin, A. L. *et al.* Cleavage of DNA and RNA by PLD3 and PLD4 limits autoinflammatory triggering by multiple sensors. *Nature communications* **12**, 5874 (2021). <https://doi.org/10.1038/s41467-021-26150-w>
- 13 Panda, S. K., Kolbeck, R. & Sanjuan, M. A. Plasmacytoid dendritic cells in autoimmunity. *Current opinion in immunology* **44**, 20-25 (2017). <https://doi.org/10.1016/j.coi.2016.10.006>
- 14 Yasaka, K. *et al.* Phospholipase D4 as a signature of toll-like receptor 7 or 9 signaling is expressed on blastic T-bet⁺ B cells in systemic lupus erythematosus. *Arthritis Research & Therapy* **25**, 200 (2023). <https://doi.org/10.1186/s13075->

023-03186-5

- 15 Van Acker, Z. P. *et al.* Phospholipase D3 degrades mitochondrial DNA to regulate nucleotide signaling and APP metabolism. *Nature communications* **14**, 2847 (2023). <https://doi.org/10.1038/s41467-023-38501-w>
- 16 Liu, Y. *et al.* Activated STING in a vascular and pulmonary syndrome. *The New England journal of medicine* **371**, 507-518 (2014). <https://doi.org/10.1056/NEJMoa1312625>
- 17 Fremond, M. L. *et al.* Overview of STING-Associated Vasculopathy with Onset in Infancy (SAVI) Among 21 Patients. *J Allergy Clin Immunol Pract* **9**, 803-818 e811 (2021). <https://doi.org/10.1016/j.jaip.2020.11.007>
- 18 Jeremiah, N. *et al.* Inherited STING-activating mutation underlies a familial inflammatory syndrome with lupus-like manifestations. *J Clin Invest* **124**, 5516-5520 (2014). <https://doi.org/10.1172/JCI79100>
- 19 Petri, M. *et al.* Baricitinib for systemic lupus erythematosus: a double-blind, randomised, placebo-controlled, phase 3 trial (SLE-BRAVE-II). *Lancet* **401**, 1011-1019 (2023). [https://doi.org/10.1016/S0140-6736\(22\)02546-6](https://doi.org/10.1016/S0140-6736(22)02546-6)
- 20 Morand, E. F. *et al.* Baricitinib for systemic lupus erythematosus: a double-blind, randomised, placebo-controlled, phase 3 trial (SLE-BRAVE-I). *Lancet* **401**, 1001-1010 (2023). [https://doi.org/10.1016/S0140-6736\(22\)02607-1](https://doi.org/10.1016/S0140-6736(22)02607-1)
- 21 Schwartz, H., McDuffie, F., Black, L., Schroeter, A. & Conn, D. in *Mayo Clinic Proceedings*. 231-238.
- 22 Trendelenburg, M. *et al.* Hypocomplementemic urticarial vasculitis or systemic lupus erythematosus? *American journal of kidney diseases* **34**, 745-751 (1999). [https://doi.org/10.1016/S0272-6386\(99\)70402-6](https://doi.org/10.1016/S0272-6386(99)70402-6)
- 23 Mehta, J. P. *et al.* Hypocomplementemic urticarial vasculitis syndrome masquerading as systemic lupus erythematosus: a case report. *Glomerular Diseases* **2**, 189-193 (2022). <https://doi.org/10.1159/000525942>
- 24 Dincy, C. *et al.* Clinicopathologic profile of normocomplementemic and hypocomplementemic urticarial vasculitis: a study from South India. *Journal of the European Academy of Dermatology and Venereology* **22**, 789-794 (2008). <https://doi.org/10.1111/j.1468-3083.2007.02641.x>
- 25 Yoshikawa, F. *et al.* Phospholipase D family member 4, a transmembrane glycoprotein with no phospholipase D activity, expression in spleen and early postnatal microglia. *PloS one* **5**, e13932 (2010). <https://doi.org/10.1371/journal.pone.0013932>
- 26 Fazzari, P. *et al.* PLD3 gene and processing of APP. *Nature* **541**, E1-E2 (2017). <https://doi.org/10.1038/nature21030>
- 27 Gavin, A. L. *et al.* PLD3 and PLD4 are single-stranded acid exonucleases that regulate endosomal nucleic-acid sensing. *Nat Immunol* **19**, 942-953 (2018). <https://doi.org/10.1038/s41590-018-0179-y>
- 28 Singh, S. *et al.* PLD3 and PLD4 synthesize S,S-BMP, a key phospholipid enabling lipid degradation in lysosomes. *Cell* **187**, 6820-6834 e6824 (2024). <https://doi.org/10.1016/j.cell.2024.09.036>
- 29 Bérouti, M. *et al.* Lysosomal endonuclease RNase T2 and PLD exonucleases

- cooperatively generate RNA ligands for TLR7 activation. *Immunity* **57**, 1482-1496. e1488 (2024). <https://doi.org/10.1016/j.immuni.2024.04.010>
- 30 Gonzalez, A. C. *et al.* Unconventional trafficking of mammalian phospholipase D3 to lysosomes. *Cell Rep* **22**, 1040-1053 (2018). <https://doi.org/10.1016/j.celrep.2017.12.100>
- 31 Yuan, M. *et al.* Structural and mechanistic insights into disease-associated endolysosomal exonucleases PLD3 and PLD4. *Structure* **32**, 766-779 e767 (2024). <https://doi.org/10.1016/j.str.2024.02.019>
- 32 Otani, Y. *et al.* PLD4 is involved in phagocytosis of microglia: expression and localization changes of PLD4 are correlated with activation state of microglia. *PloS one* **6**, e27544 (2011). <https://doi.org/10.1371/journal.pone.0027544>
- 33 Decout, A., Katz, J. D., Venkatraman, S. & Ablasser, A. The cGAS–STING pathway as a therapeutic target in inflammatory diseases. *Nature Reviews Immunology* **21**, 548-569 (2021). <https://doi.org/10.1038/s41577-021-00524-z>
- 34 Gui, X. *et al.* Autophagy induction via STING trafficking is a primordial function of the cGAS pathway. *Nature* **567**, 262-266 (2019). <https://doi.org/10.1038/s41586-019-1006-9>
- 35 Mortezaigholi, S. *et al.* Evaluation of TLR9 expression on PBMCs and CpG ODN-TLR9 ligation on IFN- α production in SLE patients. *Immunopharmacology and Immunotoxicology* **39**, 11-18 (2017). <https://doi.org/10.1080/08923973.2016.1263859>

Reply to the Referee #1

Referee #1 (Remarks to the Author):

The authors have addressed the concerns regarding text modifications/edits for accuracy and completeness.

I however have concerns about the newly added data that are either not robust or unexpected. The issues may be technical but for conclusions to be drawn, the authors need better gating strategies (including additional markers), increased mouse numbers to reach statistical significance, and a repeat mixed chimera experiment given the unexpected results.

Reply: We sincerely thank this reviewer for the time and effort invested in reviewing our manuscript. Your insightful comments are greatly valued; accordingly, we expanded mouse numbers, used a new gating strategy, repeated immunophenotyping of kidney/spleen cells in WT and *Pld4*-deficient mice, and repeated mixed chimera experiment.

These revised results confirmed significant expansion of pDCs and PCs in kidney and spleen, whereas the ABCs remained unchanged under the new gating strategy. Critically, mixed bone marrow chimeric mice revealed that the expansion of pDCs and PCs was cell-intrinsic, while the expansion of T cells was cell-extrinsic.

We have carefully addressed each comment below.

Concerns:

1. New Fig. 4e: For their ABC gating, given that they have included all B220⁺ cells, it would be important to check that they are excluding pDCs, and plot CD11c against CD19 instead of CXCR5; ABCs are CD19^{high} and clearly separated from pDCs that are CD19^{low}. They may well be enumerating pDCs, which may explain why ABCs are not appear increased in their mixed chimeras.

Reply: We thank the reviewer for this critical observation. Re-analysis of our ABCs gating strategy using specific pDC markers (Siglec-H and BST2) revealed inadvertent inclusion of pDCs in the original ABCs populations (**Fig. R1a**). Although previous gating strategies (CD45⁺CD3⁻CD19⁺B220⁺CXCR5⁻CD11c⁺) have eliminated most pDCs through CD19⁺, minor residual pDCs persist at variable levels across different mice. After using the new gating strategy (CD45⁺CD3⁻CD21⁻CD23⁻B220⁺CD19^{high}CD11c⁺), the results demonstrated no detectable pDCs populations (**Fig. R1b**).

Fig. R1 a,b, Representative the flow cytometry plot of different ABCs gating strategy.

However, the new flow cytometry results showed no statistically differences in ABCs between WT and *Pld4*^{-/-} mice in either kidney or spleen (**new Extended Data Fig. 7I and 8c**), whether assessed by frequency or total number in spleen. We believe these results accurately reflect ABCs changes in *Pld4*-deficient mice versus WT mice according to the reliability of the new ABCs gating strategy and the consistent ABCs changes across tissues.

new Extended Data Fig. 7 I, Representative the flow cytometry plot and quantification of ABCs (CD45⁺CD3⁻CD21⁻CD23⁻B220⁺CD19^{high}CD11c⁺) in WT and *Pld4*-deficient mice kidney. WT, n = 12; *Pld4*^{+/-}, n = 12; *Pld4*^{-/-}, n = 12. Data are mean ± s.e.m. These results show a representative of at least three independent experiments in (I). *n.s.*, no significant difference. One-way ANOVA with Tukey's post hoc analysis was used in (I).

new Extended Data Fig. 8 c, Representative the flow cytometry plot and quantification of ABCs ($CD45^+CD3^-CD21^-CD23^-B220^+CD19^{high}CD11c^+$) in WT and *Pld4*-deficient mice spleen. WT, $n = 12$; *Pld4*^{+/-}, $n = 12$; *Pld4*^{-/-}, $n = 12$. Data are mean \pm s.e.m. These results show a representative of at least three independent experiments in (c). *n.s.*, no significant difference. One-way ANOVA with Tukey's post hoc analysis was used in (c).

2. In new Fig. 4f there does not appear to be a clear pDC population. If the authors want to conclude pDCs are increased, they cannot do it based on the staining below. They can use a specific pDC marker such as Siglec-H, together with CD11c, which should identify a discrete population. If the authors want to claim a statistically significant effect in pDCs as the mechanism underpinning PLD4 pathogenicity, it needs to be shown robustly.

They should also show all events instead of a contour, to make sure there are enough events to be robust.

Reply: We thank the reviewer for raising this constructive suggestion. Upon using pDC-specific markers (Siglec-H) for gating of splenic and renal immune cells, discrete pDC populations can be clearly identified (**Fig. R2**).

Fig. R2 Representative the flow cytometry plot of pDCs ($CD45^+CD3^-CD19^-CD11b^-B220^+CD11c^+Siglec-H^+BST2^+$) gating strategy.

Consistent with our previous observations, renal pDCs exhibited a significant increase in *Pld4*^{-/-} mice compared to WT mice (new Fig. 4e). Critically, pDCs in spleens of *Pld4*^{-/-} mice elevated significantly following implementation of the new gating strategy (new Extended Data Fig. 8a).

new Fig. 4 e, Representative the flow cytometry plot and quantification of pDCs (CD45⁺CD3⁻CD19⁻CD11b⁻B220⁺CD11c⁺Siglec-H⁺BST2⁺) in WT and *Pld4*-deficient mice kidney. WT, n = 12; *Pld4*^{+/-}, n = 12; *Pld4*^{-/-}, n = 12. Data are mean ± s.e.m. These results show a representative of at least three independent experiments in (e). ***P* < 0.01; ****P* < 0.001. One-way ANOVA with Tukey's post hoc analysis was used in (e).

new Extended Data Fig. 8 a, Representative the flow cytometry plot and quantification of pDCs (CD45⁺CD3⁻CD19⁻CD11b⁻B220⁺CD11c⁺Siglec-H⁺BST2⁺) in WT and *Pld4*-deficient mice spleen. WT, n = 12; *Pld4*^{+/-}, n = 12; *Pld4*^{-/-}, n = 12. Data are mean ± s.e.m. These results show a representative of at least three independent experiments in (a). **P* < 0.05; ***P* < 0.01; ****P* < 0.001. One-way ANOVA with Tukey's post hoc analysis was used in (a).

3. In new Fig. 4g there does not appear to be a convincing plasma cell population. The authors should use a second plasma cell marker like CD98 or TACI instead of CD44 and show how their PC gate confidently identifies plasma cells in a secondary lymphoid tissue from the same mice side by side.

Reply: We thank the reviewer for this suggestion. Using a secondary plasma cell marker (CD98) combined with CD138 and exclusion of T cells/myeloid cells enabled identification of discrete plasma cell populations in both kidney and spleen (Fig. R3a, b).

Fig. R3 a,b, Representative the flow cytometry plot of PCs (CD45⁺CD3⁻CD11b⁻B220^{low/-}CD138⁺CD98⁺) in WT and *Pld4*-deficient mice. **a**, Representative the flow cytometry plot of PCs gating strategy in WT and *Pld4*-deficient mice spleen. **b**, Representative the flow cytometry plots of PCs in the spleen and kidney of the same mice.

Using the new gating strategy, renal and splenic PCs remained significantly expanded in *Pld4*-deficient mice compared to WT mice, consistent with prior findings (**new Fig. 4f and new Extended Data Fig. 8 b**). Comparison of spleen and kidney samples revealed significantly greater expansion of PCs in kidney versus spleen of *Pld4*^{-/-} mice.

new Fig. 4 f, Representative the flow cytometry plot and quantification of PCs (CD45⁺CD3⁻CD11b⁻B220^{low}-CD138⁺CD98⁺) in WT and *Pld4*-deficient mice kidney. These results show a representative of three independent experiments in (f). WT, n = 12; *Pld4*^{+/-}, n = 12; *Pld4*^{-/-}, n = 12. Data are mean ± s.e.m. ****P* < 0.001. One-way ANOVA with Tukey's post hoc analysis was used in (f).

new Extended Data Fig. 8 b, Representative the flow cytometry plot and quantification of PCs (CD45⁺CD3⁻CD11b⁻B220^{low}-CD138⁺CD98⁺) in WT and *Pld4*-deficient mice spleen. WT, n = 12; *Pld4*^{+/-}, n = 12; *Pld4*^{-/-}, n = 12. Data are mean ± s.e.m. These results show a representative of three independent experiments in (b). *n.s.*, no significant difference; **P* < 0.05; ***P* < 0.01. One-way ANOVA with Tukey's post hoc analysis was used in (b).

4. It is very surprising that PLD4 does not act in a B cell intrinsic manner for the observed increase in ABCs (if statistically significant) or plasma cells, given the increased TLR7 signalling and the fact that PLD4 is expressed very highly in B cells according to Immgen RNAseq. The results in their new mixed chimera experiment are all over the place: ABCs are not increased, and neither are plasma cells. There may have been a genotyping problem in the donor mouse. This mixed chimera experiment needs to be repeated for the authors to conclude that PLD4 only acts intrinsically in pDCs. The authors also need to show the FACS plots for this experiment, including a convincing gating for pDCs using a specific marker like Siglec-H.

Reply: We thank the reviewer for prompting validation of cell-intrinsic effects. To confirm whether only pDCs were cell intrinsic expansion, we increased the number of mice, re-confirmed the donor mouse genotype and used the new gating strategy, and repeated the experiment.

The results showed that the levels of anti-dsDNA and anti-dsRNA antibodies in the chimeric mice reconstituted with *Pld4*^{-/-} BM were significantly elevated (**new Fig. 4h**), which was consistent with the previous results.

We next performed flow cytometry experiments using new markers and new gating strategies. The results showed that the proportion of pDCs and PCs in the chimeric mice reconstituted with *Pld4*^{-/-} BM was significantly increased compared to reconstituted with WT BM, suggesting that the expansion of pDCs and PCs was cell-intrinsic, while the expansion of T cells was cell-extrinsic (**new Fig. 4i** and **new Extended Data Fig.**

9).

new Fig. 4 h, i, Representative the autoantibodies and flow cytometry results of mixed bone marrow chimeric mice reconstituted with 1:1 ratio of CD45.1-WT: CD45.2-WT (+/+:+/+) or CD45.1-WT: CD45.2-Pld4^{-/-} (+/+:-/-) bone marrow. **h**, ELISA detected the levels of anti-dsDNA and anti-dsRNA antibodies in the plasma of chimeric mice after 8 weeks transplantation. +/+:+/+, n = 20; +/+:-/-, n = 26. Data are mean \pm s.e.m. **i**, Flow cytometry detected the immune cell phenotyping in the kidneys of chimeric mice after 8 weeks transplantation. +/+:+/+, n = 15; +/+:-/-, n = 15. These results show a summary of two independent experiments in (**h, i**). *n.s.*, no significant difference; **P* < 0.05; ***P* < 0.01; ****P* < 0.001. Two-way ANOVA was used in (**i**), unpaired *t*-test was used in (**h**).

new Extended Data Fig. 9, a-f, Representative the kidney flow cytometry plot and quantification of pDCs (CD45⁺CD3⁻CD19⁻CD11b⁻B220⁺CD11c⁺Siglec-H⁺BST2⁺, **a**), PCs (CD45⁺CD3⁻CD11b⁻CD138⁺CD98⁺, **b**), ABCs (CD45⁺CD3⁻CD21⁻CD23⁻B220⁺CD19^{high}CD11c⁺, **c**), mDCs (CD45⁺CD3⁻B220⁻CD11b⁺CD11c⁺MHC-II⁺, **d**), CD4 Effector T cells (CD45⁺CD3⁺CD4⁺CD62L⁻CD44⁺, **e**), CD8 Effector T cells (CD45⁺CD3⁺CD8⁺CD62L⁻CD44⁺, **f**), in mixed bone marrow chimeric mice reconstituted with 1:1 ratio of CD45.1-WT: CD45.2-WT (+/+;+/+) or CD45.1-WT: CD45.2-*Pld4*^{-/-} (+/+;-/-) bone marrow. +/+;+/+, n = 15; +/+;-/-, n = 15. These results show a representative of two independent experiments in (**a-f**).

5. In straight mice, the PDC increase is not statistically significant (Extended data Fig. 8a). Given that this effect of PLD4 on PDCs is proposed to underpin the pathogenic mechanism of PLD4 deficiency in their study, they need to make this finding more robust by increasing the power to detect the difference, which may show whether there is a real difference or not.

They also need to show total pDC numbers.

Reply: We thank the reviewer for this suggestion. We increased the number of mice in the new experiment and used a new gating strategy. The results showed both the proportion and total number of splenic pDCs in the *Pld4*^{-/-} mice increased significantly compared with the WT mice (**new Extended Data Fig. 8 a**).

new Extended Data Fig. 8 a, Representative the flow cytometry plot and quantification of pDCs ($CD45^+CD3^-CD19^+CD11b^-B220^+CD11c^+Siglec-H^+BST2^+$) in WT and *Pld4*-deficient mice spleen. WT, $n = 12$; *Pld4*^{+/-}, $n = 12$; *Pld4*^{-/-}, $n = 12$. Data are mean \pm s.e.m. These results show a representative of at least three independent experiments in (a). * $P < 0.05$; ** $P < 0.01$; *** $P < 0.001$. One-way ANOVA with Tukey's post hoc analysis was used in (a).

6. In Extended data Fig. 8e, the plasma cell gating does not appear reliable. The diagonal seen in WT mice indicates autofluorescence. The authors should use a dump channel for autofluorescent and myeloid cells, and then gate PCs as CD138 vs B220 ($CD138^+$ cells should be B220 low or negative) and CD138 vs CD98 or TACI.

Reply: We appreciate the reviewer for raising this issue. To mitigate autofluorescence, T cells and myeloid cells were excluded in revised analyses and $B220^{low/-}$ PCs were subsequently gated (**Fig. R3 a**). This new gating strategy eliminated autofluorescence while clearly identifying discrete PCs populations in both spleen and kidney (**Fig. R3 a**).

Fig. R3 a, Representative the flow cytometry plot of PCs ($CD45^+CD3^-CD11b^-B220^{low/-}CD138^+CD98^+$) in WT and *Pld4*-deficient mice.

Reply to the Referee #1

Referee #1 (Remarks to the Author):

The authors have now performed high quality stains that allow unequivocal identification and robust quantification of pDCs and plasma cells. They have also repeated the mixed chimeras with increased mouse numbers and improved stains, and they now also see a B cell / plasma cell -intrinsic effect, which makes sense given the expression of PLD4 in B cells and the TLR7 dependence of the phenotype.

It is unclear why the abstract only mentions the kidney as the site with "cell-intrinsic expansion of plasmacytoid dendritic cells and plasma cells". Was this not seen in the spleen? Given that they see robust splenic phenotypes throughout, it is not clear why they authors do not show the spleen results in the mixed bone marrow chimera experiments. For completion and as a minor revision, I suggest they include the spleen data in the figures.

Reply: We thank the reviewer for this suggestion. A marked lupus nephritis phenotype was observed in our patients, and renal involvement represented the most severe manifestation in Pld4-deficient mice, accompanied by distinct renal pathology. Consistent renal involvement was confirmed as a shared feature in both PLD4-deficient patients and mice through scRNA-Seq, flow cytometry, and histological staining. Therefore, our mixed bone marrow chimeric experiments primarily focused on the immunophenotyping of kidney cells.

Addressing the reviewer's request for enhanced data completeness, we have incorporated spleen results of mixed bone marrow chimeras into the revised manuscript. The data reveal that pDCs and PCs within the spleen also exhibit cell-intrinsic effect, although to a lesser extent than observed in the kidney (**new Extended Data Fig. 9g to 9m**). This difference is presumably attributable to the unique immunological microenvironment of the kidney, potentially involving the relative paucity of suppressive cell populations, compared to spleen.

Collectively, our findings demonstrate that pDCs and PCs exhibit aberrant activation or expansion in both PLD4-deficient patients and mice, with cell-intrinsic expansion in both kidney and spleen of mice models.

Based on these data, we added "*Similarly, splenic analyses showed equivalent results, with pDCs and PCs also exhibiting cell-intrinsic effect (Extended Data Fig. 9g to 9m).*" on Page 9, line 243 to 244.

Extended Data Fig. 9 g, Flow cytometry detected the immune cell phenotyping in the spleen of chimeric mice after 8 weeks of transplantation. **h-m**, Representative the spleen flow cytometry plot and quantification of pDCs ($CD45^+CD3^-CD19^+CD11b^-B220^+CD11c^+Siglec-H^+BST2^+$, **h**), PCs ($CD45^+CD3^-CD11b^-B220^{low/-}CD138^+CD98^+$, **i**), ABCs ($CD45^+CD3^-CD21^-CD23^+B220^+CD19^{high}CD11c^+$, **j**), mDCs ($CD45^+CD3^-B220^-CD11b^+CD11c^+MHC-II^+$, **k**), CD4 Effector T cells ($CD45^+CD3^+CD4^+CD62L^-CD44^+$, **l**), MZ B cells ($CD45^+CD3^-CD19^+B220^+CD23^-CD21^+$, **m**), in mixed bone marrow chimeric mice reconstituted with 1:1 ratio of CD45.1-WT: CD45.2-WT (+/+;+/+) or CD45.1-WT: CD45.2-*Pld4*^{-/-} (+/+;-/-) bone marrow. +/+;+/+, n = 15; +/+;-/-, n = 15. These results show a summary of two independent experiments in (**g**) and a representative of two independent experiments in (**h-m**).